# A Practical, Progressively-Expressive GNN

**Lingxiao Zhao**
Carnegie Mellon University
lingxiao@cmu.edu

**Louis Härtel**
RWTH Aachen University
haertel@informatik.rwth-aachen.de

**Neil Shah**
Snap Inc.
nshah@snap.com

**Leman Akoglu**
Carnegie Mellon University
lakoglu@andrew.cmu.edu

## Abstract

Message passing neural networks (MPNNs) have become a dominant flavor of graph neural networks (GNNs) in recent years. Yet, MPNNs come with notable limitations; namely, they are at most as powerful as the 1-dimensional Weisfeiler-Leman (1-WL) test in distinguishing graphs in a graph isomorphism testing framework. To this end, researchers have drawn inspiration from the $k$-WL hierarchy to develop more expressive GNNs. However, current $k$-WL-equivalent GNNs are not practical for even small values of $k$, as $k$-WL becomes combinatorially more complex as $k$ grows. At the same time, several works have found great empirical success in graph learning tasks without highly expressive models, implying that chasing expressiveness with a *"coarse-grained ruler"* of expressivity like $k$-WL is often unneeded in practical tasks. To truly understand the expressiveness-complexity tradeoff, one desires a more *"fine-grained ruler,"* which can more gradually increase expressiveness. Our work puts forth such a proposal: Namely, we first propose the $(k, c)(\leq)$-SETWL hierarchy with greatly reduced complexity from $k$-WL, achieved by moving from $k$-tuples of nodes to sets with $\leq k$ nodes defined over $\leq c$ connected components in the induced original graph. We show favorable theoretical results for this model in relation to $k$-WL, and concretize it via $(k, c)(\leq)$-SETGNN, which is as expressive as $(k, c)(\leq)$-SETWL. Our model is *practical* and *progressively-expressive*, increasing in power with $k$ and $c$. We demonstrate effectiveness on several benchmark datasets, achieving several state-of-the-art results with runtime and memory usage applicable to practical graphs. We open source our implementation at https://github.com/LingxiaoShawn/KCSetGNN.

## 1 Introduction

In recent years, graph neural networks (GNNs) have gained considerable attention [58, 60] for their ability to tackle various node-level [35, 56], link-level [51, 66] and graph-level [9, 40] learning tasks, given their ability to learn rich representations for complex graph-structured data. The common template for designing GNNs follows the message passing paradigm; these so-called message-passing neural networks (MPNNs) are built by stacking layers which encompass feature transformation and aggregation operations over the input graph [25, 39]. Despite their advantages, MPNNs have several limitations including oversmoothing [12, 47, 67], oversquashing [2, 55], inability to distinguish node identities [63] and positions [62], and *expressive power* [61].

Since Xu et al.'s [61] seminal work showed that MPNNs are at most as powerful as the first-order Weisfeiler-Leman (1-WL) test in the graph isomorphism (GI) testing framework, there have been several follow-up works on improving the understanding of GNN expressiveness [3, 15]. In re-

36th Conference on Neural Information Processing Systems (NeurIPS 2022).

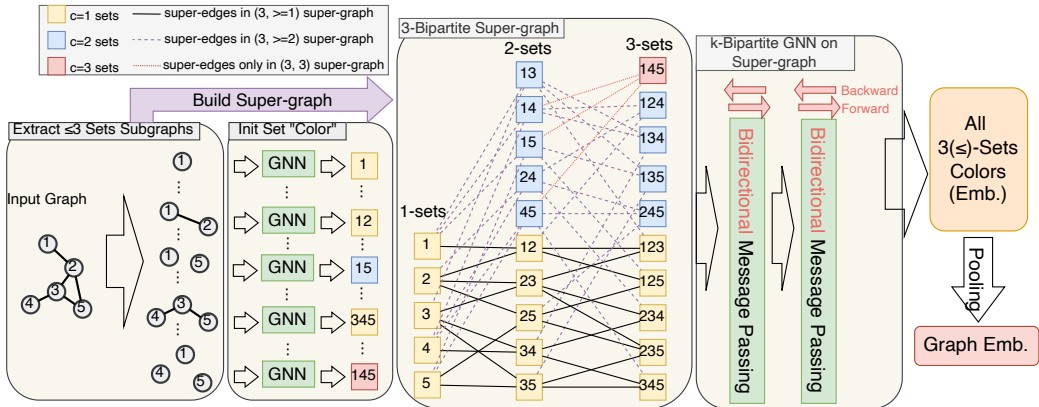

Figure 1: Main steps of $(k, c)(\leq)$-SETGNN. Given a graph and $(k, c)$, we build the $(k, c)$-bipartite super-graph (in middle) containing sets with up to $k$ nodes and $c$ connected components in the induced subgraph, on which a base GNN assigns initial "colors". Bidirectional bipartite GNN layers with frw.-bckw. message passing learn set embeddings, pooled into graph embedding. The size of super-graph, and accordingly its expressiveness, grows progressively with increasing $k$ and $c$. The 2-bipartite message passing generalizes normal GNN, edge GNN, and line graph (see Appendix.A.12).

sponse, the community proposed many GNN models to overcome such limitations [4, 52, 68]. Several of these aim to design powerful higher-order GNNs which are increasingly expressive [9, 33, 41, 43] by inspiration from the $k$-WL hierarchy [54].

A key limitation of such higher-order models that reach beyond 1-WL is their poor scalability; in fact, these models can only be applied to small graphs with small $k$ in practice [40, 41, 43] due to combinatorial growth in complexity. On the other hand lies an open question on whether practical graph learning tasks indeed need such complex, and extremely expressive GNN models. Historically, Babai et al. [5] showed that almost all graphs on $n$ nodes can be distinguished by 1-WL. In other contexts like node classification, researchers have encountered superior generalization performance with graph-augmented multi-layer perceptron (GA-MLP) models [37, 50] compared to MPNNs, despite the former being strictly less expressive than the latter [13]. Considering that increased model complexity has negative implications in terms of overfitting and generalization [29], it is worth re-evaluating continued efforts to pursue maximally expressive GNNs in practice. Ideally, one could study these models' generalization performance on various real-world tasks by increasing $k$ (expressiveness) in the $k$-WL hierarchy. Yet, given the impracticality of this too-coarse "ruler" of expressiveness, which becomes infeasible beyond $k>3$, one desires a more fine-grained "ruler", that is, a new hierarchy whose expressiveness grows more gradually. Such a hierarchy could enable us to gradually build more expressive models which admit improved scaling, and avoid unnecessary leaps in model complexity for tasks which do not require them, guarding against overfitting.

**Present Work.** In this paper, we propose such a hierarchy, and an associated *practical progressively-expressive GNN model*, called $(k, c)(\leq)$-SETGNN, whose expressiveness can be modulated through $k$ and $c$. In a nutshell, we take inspiration from $k$-WL, yet achieve practicality through three design ideas which simplify key bottlenecks of scaling $k$-WL: **First**, we move away from $k$-tuples of the original $k$-WL to $k$-multisets (unordered), and then to $k(\leq)$-sets. We demonstrate that these steps drastically reduce the number nodes in the $k$-WL graph while retaining significant expressiveness. (Sec.s 3.2−3.3) **Second**, by considering the underlying sparsity of an input graph, we reduce scope to $k, c(\leq)$-sets that consist of $\leq k$ nodes whose induced subgraph is comprised of $\leq c \leq k$ connected components. This also yields massive reduction in the number of nodes while improving practicality; i.e. small values of $c$ on sparse graphs can allow one to increase $k$ well beyond 3 in practice. (Sec. 3.4) **Third**, we design a super-graph architecture that consists of a sequence of $k−1$ *bipartite* graphs over which we learn embeddings for our $k, c(\leq)$-sets, using bidirectional message passing. These embeddings can be pooled to yield a final graph embedding.. (Sec.s 4.1−4.2) We also speed up initializing "colors" for the $k, c(\leq)$-sets, for $c > 1$, substantially reducing computation and memory. (Sec. 4.3) Fig. 1 overviews of our proposed framework. Experimentally, our $(k, c)(\leq)$-SETGNN outperforms existing state-of-the-art GNNs on simulated expressiveness as well as real-world graph-level tasks, achieving new bests on substructure counting and ZINC-12K, respectively. We show that generalization performance reflects increasing expressiveness by $k$ and $c$. Our proposed scalable designs allow us to train models with e.g. $k=10$ or $c=3$ with practical running time and memory requirements.

## 2 Related Work

**MPNN limitations.** Given the understanding that MPNNs have expressiveness upper-bounded by 1-WL [61], several researchers investigated what else MPNNs *cannot* learn. To this end, Chen et al. [15] showed that MPNNs cannot count induced connected substructures of 3 or more nodes, while along with [15], Arvind et al. [3] showed that MPNNs can only count star-shaped patterns. Loukas [38] further proved several results regarding decision problems on graphs (e.g. subgraph verification, cycle detection), finding that MPNNs cannot solve these problems unless strict conditions of depth and width are met. Moreover, Garg et al. [22] showed that many standard MPNNs cannot compute properties such as longest or shortest cycles, diameters or existence of cliques.

**Improving expressivity.** Several works aim to improve expressiveness limitations of MPNNs. One approach is to inject features into the MPNN aggregation, motivated by Loukas [38] who showed that MPNNs can be universal approximators when nodes are sufficiently distinguishable. Sato et al. [53] show that injecting random features can better enable MPNNs to solve problems like minimum dominating set and maximum matching. You et al. [63] inject cycle counts as node features, motivated by the limitation of MPNNs not being able to count cycles. Others proposed utilizing subgraph isomorphism counting to empower MPNNs with substructure information they cannot learn [9, 10]. Earlier work by You et al. [62] adds positional features to distinguish node which naïve MPNN embeddings would not. Recently, several works also propose utilizing subgraph aggregation; Zhang et al. [65], Zhao et al. [68] and Bevilacqua et al. [8] propose subgraph variants of the WL test which are no less powerful than 3-WL. Recently a following up work [21] shows that rooted subgraph based extension with 1-WL as kernel is bounded by 3-WL. The community is yet exploring several distinct avenues in overcoming limitations: Murphy et al. [46] propose a relational pooling mechanism which sums over all permutations of a permutation-sensitive function to achieve above-1-WL expressiveness. Balcilar et al. [6] generalizes spatial and spectral MPNNs and shows that instances of spectral MPNNs are more powerful than 1-WL. Azizian et al. [4] and Geerts et al. [24] unify expressivity and approximation ability results for existing GNNs.

$k$**-WL-inspired GNNs.** The $k$-WL test captures higher-order interactions in graph data by considering all $k$-tuples, i.e., size $k$ ordered multisets defined over the set of nodes. While highly expressive, it does not scale to practical graphs beyond a very small $k$ ($k = 3$ pushes the practical limit even for small graphs). However, several works propose designs which can achieve $k$-WL in theory and strive to make them practical. Maron et al. [41] proposes a general class of invariant graph networks $k$-IGNs having exact $k$-WL expressivity [23], while being not scalable. Morris et al. have several work on $k$-WL and its variants like $k$-LWL [42], $k$-GNN [43], and $\delta$-$k$-WL-GNN [44]. Both $k$-LWL and $k$-GNN use a variant of $k$-WL that only considers $k$-sets and are strictly weaker than $k$-WL but much more practical. In our paper we claim that $k(\leq)$-sets should be used with additional designs to keep the best expressivity while remaining practical. The $\delta$-$k$-WL-GNN works with $k$-tuples but sparsifies the connections among tuples by considering locality. Qian et al. [48] extends the k-tuple to subgraph network but has same scalability bottleneck as k-WL. A recent concurrent work SpeqNet [45] proposes to reduce number of tuples by restricting the number of connected components of tuples' induced subgraphs. The idea is independently explored in our paper in Sec.3.4. All four variants proposed by Morris et al. do not have realizations beyond $k > 3$ in their experiments while we manage to reach $k = 10$. Interestingly, a recent work [34] links graph transformer with $k$-IGN that having better (or equal) expressivity, however is still not scalable.

## 3 A practical progressively-expressive isomorphism test: $(k, c)(\leq)$-SETWL

We first motivate our method $(k, c)(\leq)$-SETGNN from the GI testing perspective by introducing $(k, c)(\leq)$-SETWL. We first introduce notation and background of $k$-WL (Sec. 3.1). Next, we show how to increase the practicality of $k$-WL without reducing much of its expressivity, by removing node ordering (Sec. 3.2), node repetitions (Sec. 3.3), and leveraging graph sparsity (Sec. 3.4). We then give the complexity analysis (Sec. 3.5). We close the section with extending the idea to $k$-FWL which is as expressive as $(k+1)$-WL (Sec. 3.6). All proofs are included in Appendix A.4.

**Notation:** Let $G = (V(G), E(G), l_G)$ be an undirected, colored graph with nodes $V(G)$, edges $E(G)$, and a color-labeling function $l_G : V(G) \rightarrow C$ where $C = \{c_1, ..., c_d\}$ denotes a set of $d$ distinct colors. Let $[n] = \{1, 2, 3, ..., n\}$. Let $(\cdot)$ denote a *tuple* (ordered multiset), $\{\!\{\cdot\}\!\}$ denote a *multiset* (set which allows repetition), and $\{\cdot\}$ denote a *set*. We define $\overrightarrow{\boldsymbol{v}} = (v_1, ..., v_k)$ as a $k$-tuple, $\tilde{\boldsymbol{v}} = \{\!\{v_1, ..., v_k\}\!\}$ as a $k$-multiset, and $\hat{\boldsymbol{v}} = \{v_1, ..., v_k\}$ as a $k$-set. Let $\overrightarrow{\boldsymbol{v}}[x/i] =$

$(v_1, ..., v_{i-1}, x, v_{i+1}, ..., v_k)$ denote replacing the $i$-th element in $\overrightarrow{v}$ with $x$, and analogously for $\tilde{v}[x/i]$ and $\hat{v}[x/i]$ (assume mutlisets and sets are represented with the canonical ordering). When $v_i \in V(G)$, let $G[\overrightarrow{v}], G[\tilde{v}], G[\hat{v}]$ denote the induced subgraph on $G$ with nodes inside $\overrightarrow{v}, \tilde{v}, \hat{v}$ respectively (keeping repetitions). An isomorphism between two graphs $G$ and $G'$ is a bijective mapping $p : V(G) \to V(G')$ which satisfies $\forall u, v \in V(G), (u, v) \in E(G) \iff (p(u), p(v)) \in E(G')$ and $\forall u \in V(G), l_G(u) = l_{G'}(p(u))$. Two graphs $G, G'$ are isomorphic if there exists an isomorphism between them, which we denote as $G \cong G'$.

## 3.1 Preliminaries: the $k$-Weisfeiler-Leman ($k$-WL) Graph Isomorphism Test

The 1-dimensional Weisfeiler-Leman (1-WL) test, also known as color refinement [49] algorithm, is a widely celebrated approach to (approximately) test GI. Although extremely fast and effective for most graphs (1-WL can provide canonical forms for all but $n^{-1/7}$ fraction of $n$-vertex graphs [5]), it fails to distinguish members of many important graph families, such as regular graphs. The more powerful $k$-WL algorithm first appeared in [57], and extends coloring of vertices (or 1-tuples) to that of $k$-tuples. $k$-WL is progressively expressive with increasing $k$, and can distinguish any finite set of graphs given a sufficiently large $k$. $k$-WL has many interesting connections to logic, games, and linear equations [11, 28]. Another variant of $k$-WL is called $k$-FWL (Folklore WL), such that $(k + 1)$-WL is equally expressive as $k$-FWL [11, 26]. Our work focuses on $k$-WL.

$k$-WL iteratively recolors all $n^k$ $k$-tuples defined on a graph $G$ with $n$ nodes. At iteration 0, each $k$-tuple $\overrightarrow{v} = (v_1, ..., v_k) \in V(G)^k$ is initialized with a color as its *atomic type* $\boldsymbol{at_k}(G, \overrightarrow{v})$. Assume $G$ has $d$ colors, then $\boldsymbol{at_k}(G, \overrightarrow{v}) \in \{0, 1\}^{2\binom{k}{2}+kd}$ is an ordered vector encoding. The first $\binom{k}{2}$ entries indicate whether $v_i = v_j, \forall i, j, 1 \leq i < j \leq k$, which is the node repetition information. The second $\binom{k}{2}$ entries indicate whether $(v_i, v_j) \in E(G)$. The last $kd$ entries one-hot encode the initial color of $v_i, \forall i, 1 \leq i \leq k$. Importantly, $\boldsymbol{at_k}(G, \overrightarrow{v}) = \boldsymbol{at_k}(G', \overrightarrow{v'})$ if and only if $v_i \mapsto v_i'$ is an isomorphism from $G[\overrightarrow{v}]$ to $G'[\overrightarrow{v'}]$. Let $\boldsymbol{wl}_k^{(t)}(G, \overrightarrow{v})$ denote the color of $k$-tuple $\overrightarrow{v}$ on graph $G$ at $t$-th iteration of $k$-WL, where colors are initialized with $\boldsymbol{wl}_k^{(0)}(G, \overrightarrow{v}) = \boldsymbol{at_k}(G, \overrightarrow{v})$.

At the $t$-th iteration, $k$-WL updates the color of each $\overrightarrow{v} \in V(G)^k$ according to

$$\boldsymbol{wl}_k^{(t+1)}(G, \overrightarrow{v}) = \text{HASH}\left(\boldsymbol{wl}_k^{(t)}(G, \overrightarrow{v}), \left\{\!\!\left\{ \boldsymbol{wl}_k^{(t)}(G, \overrightarrow{v}[x/1]) \Big| x \in V(G) \right\}\!\!\right\}, ..., \left\{\!\!\left\{ \boldsymbol{wl}_k^{(t)}(G, \overrightarrow{v}[x/k]) \Big| x \in V(G) \right\}\!\!\right\}\right) \quad (1)$$

Let $\boldsymbol{gwl}_k^{(t)}(G)$ denote the encoding of $G$ at $t$-th iteration of $k$-WL. Then,

$$\boldsymbol{gwl}_k^{(t)}(G) = \text{HASH}\left(\left\{\!\!\left\{ \boldsymbol{wl}_k^{(t)}(G, \overrightarrow{v}) \Big| \overrightarrow{v} \in V(G)^k \right\}\!\!\right\}\right) \quad (2)$$

Two $k$-tuples $\overrightarrow{v}$ and $\overrightarrow{u}$ are connected by an edge if $|\{i \in [k] | v_i = u_i\}| = k - 1$, or informally if they share $(k-1)$ entries. Then, $k$-WL defines a super-graph $S_{k\text{-wl}}(G)$ with its nodes being all $k$-tuples in $G$, and edges defined as above. Eq. (1) defines the rule of color refinement on $S_{k\text{-wl}}(G)$. Intuitively, $k$-WL is akin to (but more powerful as it orders subgroups of neighbors) running 1-WL algorithm on the supergraph $S_{k\text{-wl}}(G)$. As $t \to \infty$, the color $\boldsymbol{wl}_k^{(t+1)}(G, \overrightarrow{v})$ converges to a stable value, denoted as $\boldsymbol{wl}_k^{(\infty)}(G, \overrightarrow{v})$ and the corresponding stable graph color denoted as $\boldsymbol{gwl}_k^{(\infty)}(G)$. For two non-isomorphic graphs $G, G'$, $k$-WL can successfully distinguish them if $\boldsymbol{gwl}_k^{(\infty)}(G) \neq \boldsymbol{gwl}_k^{(\infty)}(G')$. The expressivity of $\boldsymbol{wl}_k^{(t)}$ can be exactly characterized by first-order logic with counting quantifiers. Let $\mathbf{C}_k^t$ denote all first-order formulas with at most $k$ variables and $t$-depth counting quantifiers, then $\boldsymbol{wl}_k^{(t)}(G, \overrightarrow{v}) = \boldsymbol{wl}_k^{(t)}(G', \overrightarrow{v'}) \iff \forall \phi \in \mathbf{C}_k^t, \phi(G, \overrightarrow{v}) = \phi(G', \overrightarrow{v'})$. Additionally, there is a $t$-step bijective pebble game that are equivalent to $t$-iteration $k$-WL in expressivity. See Appendix.A.2 for the pebble game characterization of $k$-WL.

Despite its power, $k$-WL uses all $n^k$ tuples and has $O(kn^k)$ complexity at each iteration.

## 3.2 From $k$-WL to $k$-MULTISETWL: Removing Ordering

Our first proposed adaptation to $k$-WL is to *remove ordering* in each $k$-tuple, i.e. changing $k$-tuples to $k$-multisets. This greatly reduces the number of supernodes to consider by $O(k!)$ times.

Let $\tilde{v} = \{\!\{v_1, ..., v_k\}\!\}$ be the corresponding multiset of tuple $\overrightarrow{v} = (v_1, ..., v_k)$. We introduce a canonical order function on $G$, $o_G : V(G) \to [n]$, and a corresponding indexing function $o_G^{-1}(\tilde{v}, i)$,

which returns the $i$-th element of $v_1, ..., v_k$ sorted according to $o_G$. Let $\boldsymbol{mwl}_k^{(t)}(G, \tilde{\boldsymbol{v}})$ denote the color of the $k$-multiset $\tilde{\boldsymbol{v}}$ at $t$-th iteration of $k$-MULTISETWL, formally defined next.

At $t = 0$, we initialize the color of $\tilde{\boldsymbol{v}}$ as $\boldsymbol{mwl}_k^{(0)}(G, \tilde{\boldsymbol{v}}) = \text{HASH}(\{\!\{\boldsymbol{at}_k(G, p(\tilde{\boldsymbol{v}}))|p \in \text{perm[k]}\}\!\})$ where perm[k][1] denotes the set of all permutation mappings of $k$ elements. It can be shown that $\boldsymbol{mwl}_k^{(0)}(G, \tilde{\boldsymbol{v}}) = \boldsymbol{mwl}_k^{(0)}(G', \tilde{\boldsymbol{v}}')$ if and only if $G[\tilde{\boldsymbol{v}}]$ and $G'[\tilde{\boldsymbol{v}}']$ are isomorphic.

At $t$-th iteration, $k$-MULTISETWL updates the color of every $k$-multiset by

$$\boldsymbol{mwl}_k^{(t+1)}(G, \tilde{\boldsymbol{v}}) = \text{HASH}\Big( \boldsymbol{mwl}_k^{(t)}(G, \tilde{\boldsymbol{v}}), \Big\{\!\!\Big\{ \{\!\{\boldsymbol{mwl}_k^{(t)}(G, \tilde{\boldsymbol{v}}[x/o_G^{-1}(\tilde{\boldsymbol{v}}, 1)])|x \in V(G)\}\!\}, \quad (3)$$

$$..., \{\!\{\boldsymbol{mwl}_k^{(t)}(G, \tilde{\boldsymbol{v}}[x/o_G^{-1}(\tilde{\boldsymbol{v}}, k)])|x \in V(G)\}\!\} \Big\}\!\!\Big\} \Big)$$

Where $\tilde{\boldsymbol{v}}[x/o_G^{-1}(\tilde{\boldsymbol{v}}, i)])$ denotes replacing the $i$-th (ordered by $o_G$) element of the multiset with $x \in V(G)$. Let $S_{k\text{-mwl}}(G)$ denote the super-graph defined by $k$-MULTISETWL. Similar to Eq. (2), the graph level encoding is $\boldsymbol{gmwl}_k^{(t)}(G) = \text{HASH}(\{\!\{\boldsymbol{mwl}_k^{(t)}(G, \tilde{\boldsymbol{v}})|\forall \tilde{\boldsymbol{v}} \in V(S_{k\text{-mwl}}(G))\}\!\})$.

Interestingly, although $k$-MULTISETWL has significantly fewer number of node groups than $k$-WL, we show it is no less powerful than $k$-1-WL in terms of distinguishing graphs, while being upper bounded by $k$-WL in distingushing both node groups and graphs.

**Theorem 1.** *Let $k \geq 1$ and $\boldsymbol{wl}_k^{(t)}(G, \tilde{\boldsymbol{v}}) := \{\!\{\boldsymbol{wl}_k^{(t)}(G, p(\tilde{\boldsymbol{v}}))|p \in perm[k]\}\!\}$. For all $t \in \mathbb{N}$ and all graphs $G, G'$: $k$-MULTISETWL is upper bounded by $k$-WL in distinguishing multisets $G, \tilde{\boldsymbol{v}}$ and $G', \tilde{\boldsymbol{v}}'$ at $t$-th iteration, i.e. $\boldsymbol{wl}_k^{(t)}(G, \tilde{\boldsymbol{v}}) = \boldsymbol{wl}_k^{(t)}(G', \tilde{\boldsymbol{v}}') \Longrightarrow \boldsymbol{mwl}_k^{(t)}(G, \tilde{\boldsymbol{v}}) = \boldsymbol{mwl}_k^{(t)}(G', \tilde{\boldsymbol{v}}')$.*

**Theorem 2.** *$k$-MULTISETWL is no less powerful than $(k\text{-}1)$-WL in distinguishing graphs: for any $k \geq 3$ there exists graphs that can be distinguished by $k$-MULTISETWL but not by $(k\text{-}1)$-WL.*

Theorem 2 is proved by using a variant of a series of CFI [11] graphs which cannot be distinguished by $k$-WL. This theorem shows that $k$-MULTISETWL is indeed very powerful and finding counter examples of $k$-WL distinguishable graphs that cannot be distinguished by $k$-MULTISETWL is very hard. Hence we conjecture that $k$-MULTISETWL may have the same expressivity as $k$-WL in distinguishing undirected graphs, with an attempt of proving it in Appendix.A.4.10. Additionally, the theorem also implies that $k$-MULTISETWL is strictly more powerful than $(k-1)$-MULTISETWL.

We next give a pebble game characterization of $k$-MULTISETWL, which is named doubly bijective $k$-pebble game presented in Appendix.A.3. The game is used in the proof of Theorem 2.

**Theorem 3.** *$k$-MULTISETWL has the same expressivity as the doubly bijective $k$-pebble game.*

### 3.3   From $k$-MULTISETWL to $k(\leq)$-SETWL: Removing Repetition

Next, we propose further *removing repetition* inside any $k$-multiset, i.e. transforming $k$-multisets to $k(\leq)$-sets. We assume elements of $k$-multiset $\tilde{\boldsymbol{v}}$ and $k(\leq)$-set $\hat{\boldsymbol{v}}$ in $G$ are sorted based on the ordering function $o_G$, and omit $o_G$ for clarity. Let $s(\cdot)$ transform a multiset to set by removing repeats, and let $r(\cdot)$ return a tuple with the number of repeats for each distinct element in a multiset. Specifically, let $\hat{\boldsymbol{v}} = s(\tilde{\boldsymbol{v}})$ and $\hat{\boldsymbol{n}} = r(\tilde{\boldsymbol{v}})$, then $m := |\hat{\boldsymbol{v}}| = |\hat{\boldsymbol{n}}| \in [k]$ denotes the number of distinct elements in $k$-multiset $\tilde{\boldsymbol{v}}$, and $\forall i \in [m]$, $\hat{\boldsymbol{v}}_i$ is the $i$-th distinct element with $\hat{\boldsymbol{n}}_i$ repetitions. Clearly there is an injective mapping between $\tilde{\boldsymbol{v}}$ and $(\hat{\boldsymbol{v}}, \hat{\boldsymbol{n}})$; let $f$ be the inverse mapping such that $\tilde{\boldsymbol{v}} = f(s(\tilde{\boldsymbol{v}}), r(\tilde{\boldsymbol{v}}))$. Equivalently, each $m$-set $\hat{\boldsymbol{v}}$ can be mapped with a multiset of $k$-multisets: $\hat{\boldsymbol{v}} \leftrightarrow \{\!\{\tilde{\boldsymbol{v}} = f(\hat{\boldsymbol{v}}, \hat{\boldsymbol{n}}) \mid \sum_{i=1}^{m} \hat{\boldsymbol{n}}_i = k, \forall i \ \hat{\boldsymbol{n}}_i \geq 1\}\!\}$. Based on this relationship, we extend the connection among $k$-multisets to $k(\leq)$-sets: given $m_1, m_2 \in [k]$, a $m_1$-set $\hat{\boldsymbol{v}}$ is connected to a $m_2$-set $\hat{\boldsymbol{u}}$ if and only if $\exists \hat{\boldsymbol{n}}_v, \hat{\boldsymbol{n}}_u, f(\hat{\boldsymbol{v}}, \hat{\boldsymbol{n}}_v)$ is connected with $f(\hat{\boldsymbol{u}}, \hat{\boldsymbol{n}}_u)$ in $k$-MULTISETWL. Let $S_{k\text{-swl}}(G)$ denote the defined super-graph on $G$ by $k(\leq)$-SETWL. It can be shown that this is equivalent to either (1) $(|m_1 - m_2| = 1) \wedge (|\hat{\boldsymbol{v}} \cap \hat{\boldsymbol{u}}| = \min(m_1, m_2))$ or (2) $(m_1 = m_2) \wedge (|\hat{\boldsymbol{v}} \cap \hat{\boldsymbol{u}}| = m_1 - 1)$ is true. Notice that $S_{k\text{-swl}}(G)$ contains a sequence of $k-1$ bipartite graphs with each reflecting the connections among the $(m-1)$-sets and the $m$-sets. It also contains $k-1$ subgraphs, i.e. the connections among the $m$-sets for $m = 2, ..., k$. Later on we will show that these $k-1$ subgraphs can be ignored without affecting $k(\leq)$-SETWL.

---

[1] This function should also consider repeated elements; we omit this for clarity of presentation.

Let $\boldsymbol{swl}_k^{(t)}(G, \hat{\boldsymbol{v}})$ denote the color of $m$-set $\hat{\boldsymbol{v}}$ at $t$-th iteration of $k(\leq)$-SETWL. Now we formally define $k(\leq)$-SETWL. At $t = 0$, we initialize the color of a $m$-set $\hat{\boldsymbol{v}}$ ($m \in [k]$) as:

$$\boldsymbol{swl}_k^{(0)}(G, \hat{\boldsymbol{v}}) = \text{HASH}(\{\!\!\{ \boldsymbol{mwl}_k^{(0)}(G, f(\hat{\boldsymbol{v}}, \hat{\boldsymbol{n}})) \mid \hat{\boldsymbol{n}}_1 + ... + \hat{\boldsymbol{n}}_m = k, \forall i \; \hat{\boldsymbol{n}}_i \geq 1 \}\!\!\}) \tag{4}$$

Clearly $\boldsymbol{swl}_k^{(0)}(G, \hat{\boldsymbol{v}}) = \boldsymbol{swl}_k^{(0)}(G', \hat{\boldsymbol{v}}')$ if and only if $G[\hat{\boldsymbol{v}}]$ and $G'[\hat{\boldsymbol{v}}']$ are isomorphic. At $t$-th iteration, $k(\leq)$-SETWL updates the color of every $m$-set $\hat{\boldsymbol{v}}$ by

$$\boldsymbol{swl}_k^{(t+1)}(G, \hat{\boldsymbol{v}}) = \text{HASH}\Big( \boldsymbol{swl}_k^{(t)}(G, \hat{\boldsymbol{v}}), \{\!\!\{ \boldsymbol{swl}_k^{(t)}(G, \hat{\boldsymbol{v}} \cup \{x\}) \mid x \in V(G) \setminus \hat{\boldsymbol{v}} \}\!\!\}, \{\!\!\{ \boldsymbol{swl}_k^{(t)}(G, \hat{\boldsymbol{v}} \setminus x) \mid x \in \hat{\boldsymbol{v}} \}\!\!\},$$

$$\Big\{\!\!\Big\{ \{\!\!\{ \boldsymbol{swl}_k^{(t)}(G, \hat{\boldsymbol{v}}[x/o_G^{-1}(\hat{\boldsymbol{v}}, 1)]) \mid x \in V(G) \setminus \hat{\boldsymbol{v}} \}\!\!\}, ..., \{\!\!\{ \boldsymbol{swl}_k^{(t)}(G, \hat{\boldsymbol{v}}[x/o_G^{-1}(\hat{\boldsymbol{v}}, m)]) \mid x \in V(G) \setminus \hat{\boldsymbol{v}} \}\!\!\} \Big\}\!\!\Big\} \Big) \tag{5}$$

Notice that when $m = 1$ and $m = k$, the third and second part of the hashing input is an empty multiset, respectively. Similar to Eq. (2), we formulate the graph level encoding as $\boldsymbol{gswl}_k^{(t)}(G) = \text{HASH}(\{\!\!\{ \boldsymbol{swl}_k^{(t)}(G, \hat{\boldsymbol{v}}) \mid \forall \hat{\boldsymbol{v}} \in V(S_{k\text{-swl}}(G)) \}\!\!\})$.

To characterize the relationship of their expressivity, we first extend $k$-MULTISETWL on sets by defining the color of a $m$-set $\hat{\boldsymbol{v}}$ on $k$-MULTISETWL as $\boldsymbol{mwl}_k^{(t)}(G, \hat{\boldsymbol{v}}) := \{\!\!\{ \boldsymbol{mwl}_k^{(t)}(G, f(\hat{\boldsymbol{v}}, \hat{\boldsymbol{n}})) \mid \sum_{i=1}^m \hat{\boldsymbol{v}}_i = k, \forall i \; \hat{\boldsymbol{n}}_i \geq 1 \}\!\!\}$. We prove that $k$-MULTISETWL is at least as expressive as $k(\leq)$-SETWL in terms of separating node sets and graphs.

**Theorem 4.** *Let $k \geq 1$, then $\forall t \in \mathbb{N}$ and all graphs $G, G'$: $\boldsymbol{mwl}_k^{(t)}(G, \hat{\boldsymbol{v}}) = \boldsymbol{mwl}_k^{(t)}(G', \hat{\boldsymbol{v}}') \Longrightarrow \boldsymbol{swl}_k^{(t)}(G, \hat{\boldsymbol{v}}) = \boldsymbol{swl}_k^{(t)}(G', \hat{\boldsymbol{v}}')$.*

We also conjecture that $k$-MULTISETWL and $k(\leq)$-SETWL could be equally expressive, and leave it to future work. As a single $m$-set corresponds to $\binom{k-1}{m-1}$ $k$-multisets, moving from $k$-MULTISETWL to $k(\leq)$-SETWL further reduces the computational cost greatly.

### 3.4 From $k(\leq)$-SETWL to $(k, c)(\leq)$-SETWL: Accounting for Sparsity

Notice that for two arbitrary graphs $G$ and $G'$ with equal number of nodes, the number of $k(\leq)$-sets and the connections among all $k(\leq)$-sets in $k(\leq)$-SETWL are exactly the same, regardless of whether they are dense or sparse. We next propose to *account for the sparsity* of a graph $G$ to further reduce the complexity of $k(\leq)$-SETWL. As the graph structure is encoded inside every $m$-set, when the graph becomes sparser, there would be more sparse $m$-sets with a potentially large number of disconnected components. Based on the hypothesis that the induced subgraph over a set (of nodes) with fewer disconnected components naturally contains more structural information, we propose to restrict the $k(\leq)$-sets to be $(k, c)(\leq)$-sets: all sets with at most $k$ nodes and at most $c$ connected components in its induced subgraph. Let $S_{k,c\text{-swl}}(G)$ denote the super-graph defined by $(k, c)(\leq)$-SETWL, then $S_{k,c\text{-swl}}(G) = S_{k\text{-swl}}(G)[\{\hat{\boldsymbol{v}} | \#\text{components}(G[\hat{\boldsymbol{v}}]) \leq c\}]$, which is the induced subgraph on the super-graph defined by $k(\leq)$-SETWL. Fortunately, $S_{k,c\text{-swl}}(G)$ can be efficiently and recursively constructed based on $S_{(k-1,c)\text{-swl}}(G)$, and we include the construction algorithm in the Appendix A.5. $(k, c)(\leq)$-SETWL can be defined similarly to $k(\leq)$-SETWL (Eq. (4) and Eq. (5)), however while removing all colors of sets that do not exist on $S_{k,c\text{-swl}}(G)$.

$(k, c)(\leq)$-SETWL is progressively expressive with increasing $k$ and $c$, and when $c = k$, $(k, c)(\leq)$-SETWL becomes the same as $k(\leq)$-SETWL, as all $k(\leq)$-sets are then considered. Let $\boldsymbol{swl}_{k,c}^{(t)}(G, \hat{\boldsymbol{v}})$ denote the color of a $(k, c)(\leq)$-set $\hat{\boldsymbol{v}}$ on $t$-th iteration of $(k, c)(\leq)$-SETWL, then $\boldsymbol{gswl}_{k,c}^{(t)}(G) = \text{HASH}(\{\!\!\{ \boldsymbol{swl}_{k,c}^{(t)}(G, \hat{\boldsymbol{v}}) | \forall \hat{\boldsymbol{v}} \in S_{k,c\text{-swl}}(G) \}\!\!\})$.

**Theorem 5.** *Let $k \geq 1$, then $\forall t \in \mathbb{N}$ and all graphs $G, G'$:*
- *(1) when $1 \leq c_1 < c_2 \leq k$, if $G, G'$ cannot be distinguished by $(k, c_2)(\leq)$-SETWL, they cannot be distinguished by $(k, c_1)(\leq)$-SETWL*
- *(2) when $k_1 < k_2$, $\forall c \leq k_1$, if $G, G'$ cannot be distinguished by $(k_2, c)(\leq)$-SETWL, they cannot be distinguished by $(k_1, c)(\leq)$-SETWL*

### 3.5 Complexity Analysis

All color refinement algorithms described above run on a super-graph; thus, their complexity at each iteration is linear to the number of supernodes and number of edges of the super-graph. Instead

of using big$\mathcal{O}$ notation that ignores constant factors, we compare the exact number of supernodes and edges. Let $G$ be the input graph with $n$ nodes and average degree $d$. For $k$-WL, there are $n^k$ supernodes and each has $n*k$ number of neighbors, hence $S_{k\text{-wl}}(G)$ has $n^k$ supernodes and $kn^{k+1}/2$ edges. For $m \in [k]$, there are $\binom{n}{m}$ $m$-sets and each connects to $m$ number of $(m-1)$-sets. So $S_{k\text{-swl}}$ has $\sum_{i=1}^{k} \binom{n}{i} \leq \binom{n}{k}\frac{n-k+1}{n-2k+1}$ supernodes and $\sum_{i=2}^{k} i\binom{n}{i} = n\sum_{i=1}^{k-1}\binom{n-1}{i} \leq n\binom{n-1}{k-1}\frac{n-k+1}{n-2k+2}$ edges (derivation in Appendix). Here we ignore edges within $m$-sets for any $m \in [k]$ as they can be reconstructed from the bipartite connections among $(m-1)$-sets and $m$-sets, described in detail in Sec. 4.1. Consider e.g., $n = 30, k = 5$; we get $\frac{|V(S_{k\text{-wl}}(G))|}{|V(S_{k\text{-swl}}(G))|} = 139$, $\frac{|E(S_{k\text{-wl}}(G))|}{|E(S_{k\text{-swl}}(G))|} = 2182$. Directly analyzing the savings by restricting number of components is not possible without assuming a graph family. In Sec. 5.3 we measured the scalability for $(k,c)(\leq)$-SETGNN with different number of components directly on sparse graphs, where $(k,c)(\leq)$-SETWL shares similar scalability.

### 3.6 Set version of $k$-FWL

$k$-FWL is a stronger GI algorithm and it has the same expressivity as $k+1$-WL [26], in this section we also demonstrate how to extend the set to $k$-FWL to get $k(\leq)$-SETFWL. Let $\boldsymbol{fwl}_k^{(t)}(G, \overrightarrow{\boldsymbol{v}})$ denote the color of $k$-tuple $\overrightarrow{\boldsymbol{v}}$ at $t$-th iteration of $k$-FWL. Then $k$-FWL is initialized the same as the $k$-WL, i.e. $\boldsymbol{fwl}_k^{(0)}(G, \overrightarrow{\boldsymbol{v}}) = \boldsymbol{wl}_k^{(0)}(G, \overrightarrow{\boldsymbol{v}})$. At $t$-th iteration, $k$-FWL updates colors with

$$\boldsymbol{fwl}_k^{(t+1)}(G,\overrightarrow{\boldsymbol{v}}) = \text{HASH}\left(\boldsymbol{fwl}_k^{(t)}(G,\overrightarrow{\boldsymbol{v}}), \left\{\!\!\left\{ \left(\boldsymbol{fwl}_k^{(t)}(G,\overrightarrow{\boldsymbol{v}}[x/1]),...,\boldsymbol{fwl}_k^{(t)}(G,\overrightarrow{\boldsymbol{v}}[x/k])\right) \middle| x \in V(G) \right\}\!\!\right\}\right) \quad (6)$$

Let $\boldsymbol{sfwl}_k^{(t)}(G,\hat{\boldsymbol{v}})$ denote the color of $m$-set $\hat{\boldsymbol{v}}$ at $t$-th iteration of $k(\leq)$-SETFWL. Then at $t$-th iteration it updates with

$$\boldsymbol{sfwl}_k^{(t+1)}(G,\hat{\boldsymbol{v}}) = \text{HASH}\left(\boldsymbol{sfwl}_k^{(t)}(G,\hat{\boldsymbol{v}}), \left\{\!\!\left\{ \left\{\!\!\left\{ \boldsymbol{sfwl}_k^{(t)}(G,\hat{\boldsymbol{v}}[x/1]),...,\boldsymbol{sfwl}_k^{(t)}(G,\hat{\boldsymbol{v}}[x/m]) \right\}\!\!\right\} \middle| x \in V(G) \right\}\!\!\right\}\right) \quad (7)$$

The $k(\leq)$-SETFWL should have better expressivity than $k(\leq)$-SETWL. We show in the next section that $k(\leq)$-SETWL can be further improved with less computation through an intermediate step while this is nontrivial for $k(\leq)$-SETFWL. We leave it to future work of studying $k(\leq)$-SETFWL.

## 4 A practical progressively-expressive GNN: $(k,c)(\leq)$-SETGNN

In this section we transform $(k,c)(\leq)$-SETWL to a GNN model by replacing the HASH function in Eq. (5) with a combination of MLP and DeepSet [64], given they are universal function approximators for vectors and sets, respectively [31, 64]. After the transformation, we propose two additional improvements (Sec. 4.2 and Sec. 4.3) to further improve scalability. We work on vector-attributed graphs. Let $G = (V(G), E(G), X)$ be an undirected graph with node features $\mathbf{x}_i \in \mathbb{R}^d, \forall i \in V(G)$.

### 4.1 From $(k,c)(\leq)$-SetWL to $(k,c)(\leq)$-SetGNN

$(k,c)(\leq)$-SETWL defines a super-graph $S_{k,c\text{-swl}}$, which aligns with Eq. (5). We first rewrite Eq. (5) to reflect its connection to $S_{k,c\text{-swl}}$. For a supernode $\hat{\boldsymbol{v}}$ in $S_{k,c\text{-swl}}$, let $\mathcal{N}_{\text{left}}^G(\hat{\boldsymbol{v}}) = \{\hat{\boldsymbol{u}} \mid \hat{\boldsymbol{u}} \in S_{k,c\text{-swl}}, \hat{\boldsymbol{u}} \leftrightarrow \hat{\boldsymbol{v}} \text{ and } |\hat{\boldsymbol{u}}| = |\hat{\boldsymbol{v}}| - 1\}$, and $\mathcal{N}_{\text{right}}^G(\hat{\boldsymbol{v}}) = \{\hat{\boldsymbol{u}} \mid \hat{\boldsymbol{u}} \in S_{k,c\text{-swl}}, \hat{\boldsymbol{u}} \leftrightarrow \hat{\boldsymbol{v}} \text{ and } |\hat{\boldsymbol{u}}| = |\hat{\boldsymbol{v}}| + 1\}$. Then we can rewrite Eq. (5) for $(k,c)(\leq)$-SETWL as

$$\boldsymbol{swl}_{k,c}^{(t+1)}(G,\hat{\boldsymbol{v}}) = \left(\boldsymbol{swl}_{k,c}^{(t)}(G,\hat{\boldsymbol{v}}), \boldsymbol{swl}_{k,c}^{(t+\frac{1}{2})}(G,\hat{\boldsymbol{v}}), \{\!\!\{\boldsymbol{swl}_{k,c}^{(t)}(G,\hat{\boldsymbol{u}}) \mid \hat{\boldsymbol{u}} \in \mathcal{N}_{\text{left}}^G(\hat{\boldsymbol{v}})\}\!\!\}, \{\!\!\{\boldsymbol{swl}_{k,c}^{(t+\frac{1}{2})}(G,\hat{\boldsymbol{u}}) \mid \hat{\boldsymbol{u}} \in \mathcal{N}_{\text{left}}^G(\hat{\boldsymbol{v}})\}\!\!\}\right) \quad (8)$$

where $\boldsymbol{swl}_{k,c}^{(t+\frac{1}{2})}(G,\hat{\boldsymbol{v}}) := \{\!\!\{\boldsymbol{swl}_{k,c}^{(t)}(G,\hat{\boldsymbol{u}}) \mid \hat{\boldsymbol{u}} \in \mathcal{N}_{\text{right}}^G(\hat{\boldsymbol{v}})\}\!\!\}$. Notice that we omit HASH and apply it implicitly. Eq. (8) essentially splits the computation of Eq. (5) into two steps and avoids repeated computation via caching the explicit $t+\frac{1}{2}$ step. It also implies that the connection among $m$-sets for any $m \in [k]$ can be reconstructed from the bipartite graph among $m$-sets and $(m-1)$-sets.

Next we formulate $(k,c)(\leq)$-SETGNN formally. Let $h^{(t)}(\hat{\boldsymbol{v}}) \in \mathbb{R}^{d_t}$ denote the vector representation of supernode $\hat{\boldsymbol{v}}$ on the $t$-th iteration of $(k,c)(\leq)$-SETGNN. For any input graph $G$, it initializes representations of supernodes by

$$h^{(0)}(\hat{\boldsymbol{v}}) = \text{BaseGNN}(G[\hat{\boldsymbol{v}}]) \quad (9)$$

where the BaseGNN can be any GNN model. Theoretically the BaseGNN should be chosen to encode non-isomorphic induced subgraphs distinctly, mimicking HASH. Based on empirical tests

of several GNNs on all (11,117) possible 8-node non-isomorphic graphs [6], and given GIN [61] is simple and fast with nearly $100\%$ separation rate, we use GIN as our BaseGNN in experiments.

$(k, c)(\leq)$-SETGNN iteratively updates representations of all supernodes by

$$h^{(t+\frac{1}{2})}(\hat{\boldsymbol{v}}) = \sum_{\hat{\boldsymbol{u}} \in \mathcal{N}^G_{\text{right}}(\hat{\boldsymbol{v}})} \text{MLP}^{(t+\frac{1}{2})}(h^{(t)}(\hat{\boldsymbol{u}})) \tag{10}$$

$$h^{(t+1)}(\hat{\boldsymbol{v}}) = \text{MLP}^{(t)}\Big(h^{(t)}(\hat{\boldsymbol{v}}), h^{(t+\frac{1}{2})}(\hat{\boldsymbol{v}}), \sum_{\hat{\boldsymbol{u}} \in \mathcal{N}^G_{\text{left}}(\hat{\boldsymbol{v}})} \text{MLP}^{(t)}_A(h^{(t)}(\hat{\boldsymbol{u}})), \sum_{\hat{\boldsymbol{u}} \in \mathcal{N}^G_{\text{left}}(\hat{\boldsymbol{v}})} \text{MLP}^{(t)}_B(h^{(t+\frac{1}{2})}(\hat{\boldsymbol{u}}))\Big) \tag{11}$$

Then after $T$ iterations, we compute the graph level encoding as

$$h^{(T)}(G) = \text{POOL}(\{\!\{h^{(T)}(\hat{\boldsymbol{v}}) \mid \hat{\boldsymbol{v}} \in V(S_{k,c\text{-swl}})\}\!\}) \tag{12}$$

where POOL can be chosen as summation. We visualize the steps in Figure 1. Under mild conditions, $(k, c)(\leq)$-SETGNN has the same expressivity as $(k, c)(\leq)$-SETWL.

**Theorem 6.** *When (i) BaseGNN can distinguish any non-isomorhpic graphs with at most $k$ nodes, (ii) all MLPs have sufficient depth and width, and (iii) POOL is an injective function, then for any $t \in \mathbb{N}$, $t$-layer $(k, c)(\leq)$-SETGNN is as expressive as $(k, c)(\leq)$-SETWL at the $t$-th iteration.*

The following facts can be derived easily from Theorem 5 and Theorem 6.

**Corollary 6.1.** $(k, c)(\leq)$-SETGNN *is progressively-expressive with increasing $k$ and $c$, that is,*

*(1) when $c_1 > c_2$, $(k, c_1)(\leq)$-SETGNN is more expressive than $(k, c_2)(\leq)$-SETGNN, and*

*(2) when $k_1 > k_2$, $(k_1, c)(\leq)$-SETGNN is more expressive than $(k_2, c)(\leq)$-SETGNN.*

### 4.2 Bidirectional Sequential Message Passing

The $t$-th layer of $(k, c)(\leq)$-SETGNN (Eq. (10) and Eq. (11)) are essentially propagating information back and forth on the super-graph $S_{k,c\text{-swl}}(G)$, which is a sequence of $k-1$ bipartite graphs (see the middle of Figure 1), in *parallel* for all supernodes. We propose to change it to bidirectional blockwise *sequential* message passing, which we call $(k, c)(\leq)$-SETGNN*, defined as follows.

$$m = k-1 \text{ to } 1, \forall m\text{-set } \hat{\boldsymbol{v}}, h^{(t+\frac{1}{2})}(\hat{\boldsymbol{v}}) = \text{MLP}^{(t)}_{m,1}\Big(h^{(t)}(\hat{\boldsymbol{v}}), \sum_{\hat{\boldsymbol{u}} \in \mathcal{N}^G_{\text{right}}(\hat{\boldsymbol{v}})} \text{MLP}^{(t)}_{m,2}(h^{(t+\frac{1}{2})}(\hat{\boldsymbol{u}}))\Big) \tag{13}$$

$$m = 2 \text{ to } k, \forall m\text{-set } \hat{\boldsymbol{v}}, h^{(t+1)}(\hat{\boldsymbol{v}}) = \text{MLP}^{(t+\frac{1}{2})}_{m,1}\Big(h^{(t+\frac{1}{2})}(\hat{\boldsymbol{v}}), \sum_{\hat{\boldsymbol{u}} \in \mathcal{N}^G_{\text{left}}(\hat{\boldsymbol{v}})} \text{MLP}^{(t+\frac{1}{2})}_{m,2}(h^{(t+1)}(\hat{\boldsymbol{u}}))\Big) \tag{14}$$

Notice that $(k, c)(\leq)$-SETGNN* has lower memory usage, as $(k, c)(\leq)$-SETGNN load the complete supergraph ($k-1$ bipartites) while $(k, c)(\leq)$-SETGNN* loads 1 out of $k-1$ bipartites at a time, which is beneficial for limited-size GPUs. What is more, for a small, finite $t$ it is even more expressive than $(k, c)(\leq)$-SETGNN. We provide both implementation of parallel and sequential message passing in the official github repository, while only report the performance of $(k, c)(\leq)$-SETGNN* given its efficiency and better expressivity.

**Theorem 7.** *For any $t \in \mathbb{N}$, the $t$-layer $(k, c)(\leq)$-SETGNN* is more expressive than the $t$-layer $(k, c)(\leq)$-SETGNN. As $\lim_{t \to \infty}$, $(k, c)(\leq)$-SETGNN is as expressive as $(k, c)(\leq)$-SETGNN*.*

### 4.3 Improving Supernode Initialization

Next we describe how to improve the supernode initialization (Eq. (9)) and extensively reduce computational and memory overhead for $c > 1$ without losing any expressivity. We achieve this by the fact that a graph with $c$ components can be viewed as a set of $c$ connected components. Formally,

**Theorem 8.** *Let $G$ be a graph with $c$ connected components $C_1, ..., C_c$, and $G'$ be a graph also with $c$ connected components $C'_1, ..., C'_c$, then $G$ and $G'$ are isomorphic if and only if $\exists p : [c] \to [c]$, s.t. $\forall i \in [c]$, $C_i$ and $C'_{p(i)}$ are isomorphic.*

The theorem implies that we only need to apply Eq. (9) to all supernodes with a single component, and for any $c$-components $\hat{\boldsymbol{v}}$ with components $\hat{\boldsymbol{u}}_1, ..., \hat{\boldsymbol{u}}_c$, we can get the encoding by $h^{(0)}(\hat{\boldsymbol{v}}) = \text{DeepSet}(\{h^{(0)}(\hat{\boldsymbol{u}}_1), ..., h^{(0)}(\hat{\boldsymbol{u}}_c)\})$ without passing its induced subgraph to BaseGNN. This eliminates the heavy computation of passing a large number of induced subgraphs. We give the algorithm of building connection between $\hat{\boldsymbol{v}}$ to its single components in Appendix A.6.

Table 1: Simulation data performances. For $(k, c)(\leq)$-SETGNN*, $(k, c)$ values that achieve reported performance in parenthesis. (ACC: accuracy, MA[S]E: mean abs.[sq.] error)

| Method | EXP (ACC) | SR25 (ACC) | Counting Substructures (MAE) | | | | Graph Properties ($\log_{10}$(MSE)) | | |
|---|---|---|---|---|---|---|---|---|---|
| | | | Triangle | Tailed Tri. | Star | 4-Cycle | IsConnected | Diameter | Radius |
| GCN | 50% | 6.67% | 0.4186 | 0.3248 | 0.1798 | 0.2822 | -1.7057 | -2.4705 | -3.9316 |
| GIN | 50% | 6.67% | 0.3569 | 0.2373 | 0.0224 | 0.2185 | -1.9239 | -3.3079 | -4.7584 |
| PNA* | 50% | 6.67% | 0.3532 | 0.2648 | 0.1278 | 0.2430 | -1.9395 | -3.4382 | -4.9470 |
| PPGN | **100%** | 6.67% | 0.0089 | 0.0096 | 0.0148 | 0.0090 | -1.9804 | -3.6147 | -5.0878 |
| GIN-AK$^+$ | **100%** | 6.67% | 0.0123 | 0.0112 | 0.0150 | 0.0126 | -2.7513 | -3.9687 | -5.1846 |
| PNA*-AK$^+$ | **100%** | 6.67% | 0.0118 | 0.0138 | 0.0166 | 0.0132 | -2.6189 | -3.9011 | **-5.2026** |
| $(k, c)(\leq)$ | **100%** ($\geq 3, \geq 2$) | **100%** ($\geq 4, \geq 1$) | **0.0073** (3, 2) | **0.0075** (4, 1) | **0.0134** (3, 2) | **0.0075** (4, 1) | **-5.4667** (4, 2) | **-4.0800** (4, 1) | -5.1603 (2, 2) |

Table 2: Train and Test performances on substructure counting tasks by varying $k$ and $c$. Notice the orders of magnitude drop in Test MAE between bolded entries per task.

| | | Counting Substructures (MAE) | | | | | | | |
|---|---|---|---|---|---|---|---|---|---|
| | | Triangle | | Tailed Tri. | | Star | | 4-Cycle | |
| k | c | Train | Test | Train | Test | Train | Test | Train | Test |
| 2 | 1 | 0.9941 ± 0.2623 | **1.1409 ± 0.1224** | 1.1506 ± 0.2542 | **0.8695 ± 0.0781** | 1.5348 ± 2.0697 | **2.3454 ± 0.8198** | 1.2159 ± 0.0292 | 0.8361 ± 0.1171 |
| 3 | 1 | 0.0311 ± 0.0025 | **0.0088 ± 0.0001** | 0.0303 ± 0.0108 | **0.0085 ± 0.0018** | 0.0559 ± 0.0019 | **0.0151 ± 0.0006** | 0.1351 ± 0.0058 | **0.1893 ± 0.0030** |
| 4 | 1 | 0.0321 ± 0.0008 | 0.0151 ± 0.0074 | 0.0307 ± 0.0085 | 0.0075 ± 0.0012 | 0.0687 ± 0.0104 | 0.0339 ± 0.0009 | 0.0349 ± 0.0007 | **0.0075 ± 0.0002** |
| 5 | 1 | 0.0302 ± 0.0070 | 0.0208 ± 0.0042 | 0.0553 ± 0.0009 | 0.0189 ± 0.0024 | 0.0565 ± 0.0078 | 0.0263 ± 0.0023 | 0.0377 ± 0.0057 | 0.0175 ± 0.0036 |
| 6 | 1 | 0.0344 ± 0.0024 | 0.0247 ± 0.0085 | 0.0357 ± 0.0017 | 0.0171 ± 0.0000 | 0.0560 ± 0.0000 | 0.0168 ± 0.0022 | 0.0356 ± 0.0014 | 0.0163 ± 0.0064 |
| 2 | 2 | 0.3452 ± 0.0329 | 0.4029 ± 0.0053 | 0.2723 ± 0.0157 | 0.2898 ± 0.0055 | 0.0466 ± 0.0025 | 0.0242 ± 0.0006 | 0.2369 ± 0.0123 | 0.2512 ± 0.0029 |
| 3 | 2 | 0.0234 ± 0.0030 | 0.0073 ± 0.0009 | 0.0296 ± 0.0074 | 0.0100 ± 0.0009 | 0.0640 ± 0.0003 | 0.0134 ± 0.0006 | 0.0484 ± 0.0135 | 0.0194 ± 0.0065 |
| 4 | 2 | 0.0587 ± 0.0356 | 0.0131 ± 0.0010 | 0.0438 ± 0.0140 | 0.0094 ± 0.0002 | 0.0488 ± 0.0008 | 0.0209 ± 0.0063 | 0.0464 ± 0.0037 | 0.0110 ± 0.0020 |

## 5 Experiments

We design experiments to answer the following questions. **Q1. Performance:** How does $(k, c)(\leq)$-SETGNN* compare to SOTA expressive GNNs? **Q2. Varying $k$ and $c$:** How does the progressively increasing expressiveness reflect on generalization performance? **Q3. Computational requirements:** Is $(k, c)(\leq)$-SETGNN feasible on practical graphs w.r.t. running time and memory usage?

### 5.1 Setup

**Datasets.** To inspect the expressive power, we use four different types of simulation datasets: **1)** EXP [1] contains 600 pairs of 1&2-WL failed graphs which we split into two where graphs in each pair is assigned to two different classes; **2)** SR25 [6] has 15 strongly regular graphs (3-WL failed) with 25 nodes each, which we transform to a 15-way classification task; **3)** Substructure counting (i.e. triangle, tailed triangle, star and 4-cycle) tasks on random graph dataset [15]; **4)** Graph property regression (i.e. connectedness, diameter, radius) tasks on random graph dataset [17]. We also evaluate performance on two real world graph learning tasks: **5)** ZINC-12K [19], and **6)** QM9 [59] for molecular property regression. See Table 6 in Appendix A.7 for detailed dataset statistics.

**Baselines.** We use GCN [36], GIN [61], PNA* [17], PPGN [40], PF-GNN [18], and GNN-AK [68] as baselines on the simulation datasets. On ZINC-12K we also reference CIN [9] directly from literature. Most baselines results are taken from [68]. Finally, we compare to GINE [32] on QM9. **Hyperparameter and model configurations** are described in Appendix A.8.

### 5.2 Results

Theorem 2 shows that $k$-MULTISETWL is able to distinguish CFI(k) graphs. It also holds for $k(\leq)$-SETWL as the proof doesn't use repetitions. We implemented the construction of CFI(k) graphs for any k. Empirically we found that $(k, c)(\leq)$-SETGNN* *is able to distinguish CFI(k) for k = 3, 4, 5 (k>6 out of memory)*. This empirically verifies its theoretical expressivity.

Table 1 shows the performance results on all the simulation datasets. The low expressive models such as GCN, GIN and PNA* underperform on EXP and SR25, while 3-WL equivalent PPGN excels only on EXP. Notably, $(k, c)(\leq)$-SETGNN* achieves 100% discriminating power with any $k$ larger than 2 and 3, and any $c$ larger than 1 and 0, resp. for EXP and SR25. $(k, c)(\leq)$-SETGNN* also outperforms the baselines on all substructure counting tasks, as well as on two out of three graph property prediction tasks (except Radius) with significant gap, for relatively small values of $k$ and $c$.

In Table 2, we show the train and test MAEs on substructure counting tasks for individual values of $k$ and $c$. As expected, performances improve for increasing $k$ when $c$ is fixed, and vice versa. It is notable that orders of magnitude improvements on test error occur moving from $k$=2 to 3 for the

triangle tasks as well as the star task, while a similarly large magnitude drop is obtained at $k{=}4$ for the 4-cycle task, which is expected as triangle and 4-cycle have 3 and 4 nodes respectively. Similar observations hold for graph property tasks as well. (See Appendix A.9)

In addition to simulation datasets, we evaluate our $(k, c)(\leq)$-SETGNN$^*$ on real-world data; Table 3 shows our performance on ZINC-12K. Our method achieves a new state-of-the-art performance, with a mean absolute error (MAE) of 0.0750, using $k{=}5$ and $c{=}2$.

In Table 4 we show the test and validation MAE along with training loss for varying $k$ and $c$ for ZINC-12K. For a fixed $c$, validation MAE and training loss both follow a first decaying and later increasing trend with increasing $k$, potentially owing to the difficulty in fitting with too many sets (i.e. supernodes) and edges in the super-graph.

Similar results are shown for QM9 in Table 5. For comparison we also show GINE performances using both 4 and 6 layers, both of which are significantly lower than $(k, c)(\leq)$-SETGNN$^*$.

Table 3: SetGNN$^*$ achieves SOTA on ZINC-12K.

| Method | MAE |
|---|---|
| GatedGCN | $0.363 \pm 0.009$ |
| GCN | $0.321 \pm 0.009$ |
| PNA | $0.188 \pm 0.004$ |
| DGN | $0.168 \pm 0.003$ |
| GIN | $0.163 \pm 0.004$ |
| GINE | $0.157 \pm 0.004$ |
| HIMP | $0.151 \pm 0.006$ |
| PNA$^*$ | $0.140 \pm 0.006$ |
| GSN | $0.115 \pm 0.012$ |
| PF-GNN | $0.122 \pm 0.010$ |
| GIN-AK$^+$ | $0.080 \pm 0.001$ |
| CIN | $0.079 \pm 0.006$ |
| $(k, c)(\leq)$ | $\mathbf{0.0750 \pm 0.0027}$ |

Table 4: $(k, c)(\leq)$-SETGNN$^*$ performances on ZINC-12K by varying $(k,c)$. **Test MAE** at lowest Val. MAE, and lowest Test MAE.

| $k$ | $c$ | Train loss | Val. MAE | Test MAE |
|---|---|---|---|---|
| 2 | 1 | $0.1381 \pm 0.0240$ | $0.2429 \pm 0.0071$ | $0.2345 \pm 0.0131$ |
| 3 | 1 | $0.1172 \pm 0.0063$ | $0.2298 \pm 0.0060$ | $0.2252 \pm 0.0030$ |
| 4 | 1 | $0.0693 \pm 0.0111$ | $0.1645 \pm 0.0052$ | $0.1636 \pm 0.0052$ |
| 5 | 1 | $0.0643 \pm 0.0019$ | $0.1593 \pm 0.0051$ | $0.1447 \pm 0.0013$ |
| 6 | 1 | $0.0519 \pm 0.0064$ | $0.0994 \pm 0.0093$ | $0.0843 \pm 0.0048$ |
| 7 | 1 | $0.0543 \pm 0.0048$ | $0.0965 \pm 0.0061$ | $0.0747 \pm 0.0022$ |
| 8 | 1 | $0.0564 \pm 0.0152$ | $0.0961 \pm 0.0043$ | $0.0732 \pm 0.0037$ |
| 9 | 1 | $0.0817 \pm 0.0274$ | $0.0909 \pm 0.0094$ | $0.0824 \pm 0.0056$ |
| 10 | 1 | $0.0894 \pm 0.0266$ | $0.1060 \pm 0.0157$ | $0.0950 \pm 0.0102$ |
| 2 | 2 | $0.1783 \pm 0.0602$ | $0.2913 \pm 0.0102$ | $0.2948 \pm 0.0210$ |
| 3 | 2 | $0.0640 \pm 0.0072$ | $0.1668 \pm 0.0078$ | $0.1391 \pm 0.0102$ |
| 4 | 2 | $0.0499 \pm 0.0043$ | $0.1029 \pm 0.0033$ | $0.0836 \pm 0.0010$ |
| 5 | 2 | $0.0483 \pm 0.0017$ | $0.0899 \pm 0.0056$ | $\mathbf{0.0750 \pm 0.0027}$ |
| 6 | 2 | $0.0530 \pm 0.0064$ | $0.0927 \pm 0.0050$ | $0.0737 \pm 0.0006$ |
| 7 | 2 | $0.0547 \pm 0.0036$ | $0.0984 \pm 0.0047$ | $0.0784 \pm 0.0043$ |
| 3 | 3 | $0.0798 \pm 0.0062$ | $0.1881 \pm 0.0076$ | $0.1722 \pm 0.0086$ |
| 4 | 3 | $0.0565 \pm 0.0059$ | $0.1121 \pm 0.0066$ | $0.0869 \pm 0.0026$ |
| 5 | 3 | $0.0671 \pm 0.0156$ | $0.1091 \pm 0.0097$ | $0.0920 \pm 0.0054$ |

Table 5: $(k, c)(\leq)$-SETGNN$^*$ performances on QM9 by varying $(k,c)$. **Test MAE** at lowest Val. MAE, and lowest Test MAE. All variances are $\leq 0.002$ and thus omitted.

| $k$ | $c$ | Train loss | Val. MAE | Test MAE |
|---|---|---|---|---|
| 2 | 1 | $0.0376 \pm 0.0005$ | $0.0387 \pm 0.0007$ | $0.0389 \pm 0.0008$ |
| 3 | 1 | $0.0308 \pm 0.0010$ | $0.0386 \pm 0.0017$ | $0.0379 \pm 0.0010$ |
| 4 | 1 | $0.0338 \pm 0.0003$ | $0.0371 \pm 0.0005$ | $0.0370 \pm 0.0006$ |
| 5 | 1 | $0.0299 \pm 0.0017$ | $0.0343 \pm 0.0008$ | $0.0341 \pm 0.0009$ |
| 6 | 1 | $0.0226 \pm 0.0004$ | $0.0296 \pm 0.0007$ | $0.0293 \pm 0.0007$ |
| 7 | 1 | $0.0208 \pm 0.0005$ | $0.0289 \pm 0.0007$ | $0.0269 \pm 0.0003$ |
| 2 | 2 | $0.0367 \pm 0.0007$ | $0.0398 \pm 0.0004$ | $0.0398 \pm 0.0004$ |
| 3 | 2 | $0.0282 \pm 0.0013$ | $0.0358 \pm 0.0009$ | $0.0356 \pm 0.0007$ |
| 4 | 2 | $0.0219 \pm 0.0004$ | $0.0280 \pm 0.0008$ | $0.0278 \pm 0.0008$ |
| 5 | 2 | $0.0175 \pm 0.0003$ | $0.0267 \pm 0.0005$ | $\mathbf{0.0251 \pm 0.0006}$ |
| 3 | 3 | $0.0391 \pm 0.0107$ | $0.0428 \pm 0.0057$ | $0.0425 \pm 0.0052$ |
| 4 | 3 | $0.0219 \pm 0.0011$ | $0.0301 \pm 0.0010$ | $0.0286 \pm 0.0004$ |
| GINE ($L{=}4$) | | $0.0507 \pm 0.0014$ | $0.0478 \pm 0.0003$ | $0.0479 \pm 0.0004$ |
| GINE ($L{=}6$) | | $0.0440 \pm 0.0009$ | $0.0440 \pm 0.0009$ | $0.0451 \pm 0.0009$ |

## 5.3 Computational requirements

We next investigate how increasing $k$ and $c$ change the computational footprint of $(k, c)(\leq)$-SETGNN$^*$ in practice. Fig. 2 shows that increasing these parameters expectedly increases both memory consumption (in MB in (a)) as well as runtime (in seconds per epoch in (b)). Notably, since larger $k$ and $c$ increases our model's expressivity, we observe that suitable choices for $k$ and $c$ allow us to practically realize these increasingly expressive models on commodity hardware. With conservative values of $c$ (e.g. $c{=}1$), we are able to consider passing messages between sets of $k$ (e.g. 10) nodes far larger than $k$-WL-style higher order models can achieve ($\leq 3$).

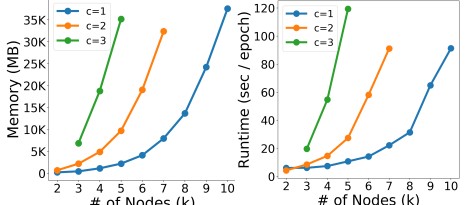

(a) Memory usage  (b) Training time

Figure 2: $(k, c)(\leq)$-SETGNN$^*$'s footprint scales practically with both $k$ and $c$ in memory (a) and running time (b) – results on ZINC-12K.

## 6 Conclusion

Our work is motivated by the impracticality of higher-order GNN models based on the $k$-WL hierarchy, which make it challenging to study how much expressiveness real-world tasks truly necessitate. To this end, we proposed $(k, c)(\leq)$-SetWL, a more practical and progressively-expressive hierarchy with theoretical connections to $k$-WL and drastically lowered complexity. We also designed and implemented a practical model $(k, c)(\leq)$-SetGNN$(^*)$, expressiveness of which is gradually increased by larger $k$ and $c$. Our model achieves strong performance, including several new best results on graph-level tasks like ZINC-12K and expressiveness-relevant tasks like substructure counting, while being practically trainable on commodity hardware.

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
