# A Appendix

## A.1 Visualization Supergraph Connection of 2-SetWL

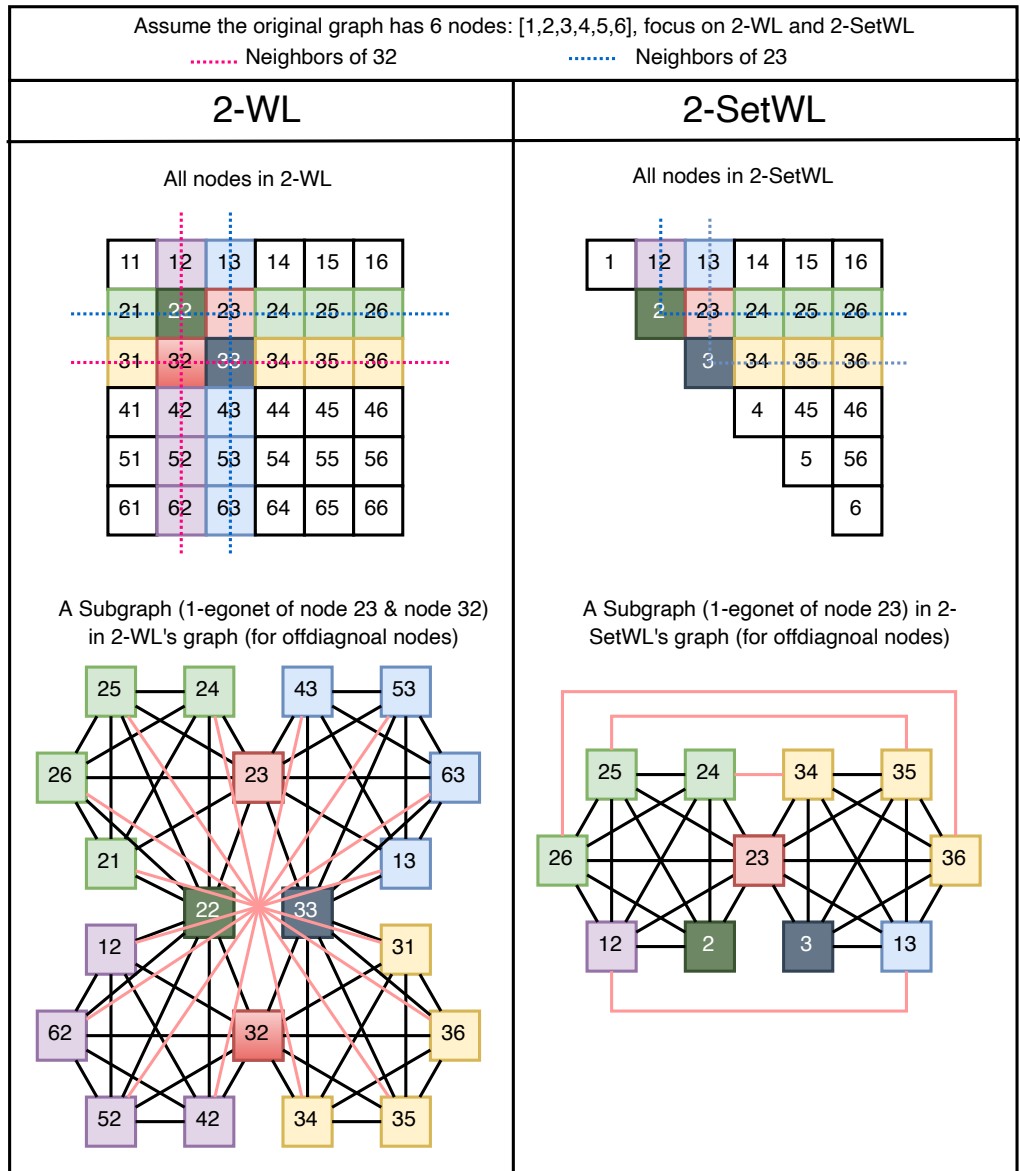

Figure 3: Comparison between 2-SetWL and 2-WL in their supergraph connection

## A.2 Bijective $k$-Pebble Game for $k$-WL

The pebble game characterization of k-FWL appeared in [11]. We use the pebble game defined in [28] for k-WL. Let $G, G'$ be graphs and $\overrightarrow{\boldsymbol{v}}_0, \overrightarrow{\boldsymbol{v}}'_0$ be $k$-tuple. Then the bijective $k$-pebble game $\mathrm{BP}_k(G, \overrightarrow{\boldsymbol{v}}_0, G', \overrightarrow{\boldsymbol{v}}'_0)$ is defined as the follows. The game is played by two players Player I and Player II. The game proceeds in rounds starting from initial position $(\overrightarrow{\boldsymbol{v}}_0, \overrightarrow{\boldsymbol{v}}'_0)$ and continues to new position $(\overrightarrow{\boldsymbol{v}}_t, \overrightarrow{\boldsymbol{v}}'_t)$ as long as after $t$ rounds $\overrightarrow{\boldsymbol{v}}_t[i] \mapsto \overrightarrow{\boldsymbol{v}}'_t[i]$ defines an isomorphism between $G[\overrightarrow{\boldsymbol{v}}_t]$ and $G'[\overrightarrow{\boldsymbol{v}}'_t]$. If the game has not ended after $t$ rounds, Player II wins the $t$-step $\mathrm{BP}_k(G, \overrightarrow{\boldsymbol{v}}_0, G', \overrightarrow{\boldsymbol{v}}'_0)$, otherwise Player I wins.

The $t$-th round is played as follows.

1. Player I picks up the $i$-th pair of pebbles with $i \in [k]$.
2. Player II chooses bijection $f : V(G) \to V(G')$.
3. Player I chooses $x \in V(G)$.
4. The new position is $(\overrightarrow{\boldsymbol{v}}_t[x/i], \overrightarrow{\boldsymbol{v}}'_t[f(x)/i])$. If $G[\overrightarrow{\boldsymbol{v}}_t[x/i]]$ and $G[\overrightarrow{\boldsymbol{v}}'_t[f(x)/i]]$ are still isomorphic, the game continues. Otherwise, the game ends and Player II loses.

The game $\mathrm{BP}_k(G, \overrightarrow{\boldsymbol{v}}_0, G', \overrightarrow{\boldsymbol{v}}'_0)$ has the same expressivity as $k$-WL in distinguishing $(G, \overrightarrow{\boldsymbol{v}}_0)$ and $(G', \overrightarrow{\boldsymbol{v}}'_0)$.

**Theorem 9.** *$k$-WL cannot distinguish $(G, \overrightarrow{\boldsymbol{v}}_0)$ and $(G', \overrightarrow{\boldsymbol{v}}'_0)$ at step $t$, i.e. $\boldsymbol{wl}_k^{(t)}(G, \overrightarrow{\boldsymbol{v}}_0) = \boldsymbol{wl}_k^{(t)}(G', \overrightarrow{\boldsymbol{v}}'_0)$, if and only if Player II has a winning strategy for $t$-step $\mathrm{BP}_k(G, \overrightarrow{\boldsymbol{v}}_0, G', \overrightarrow{\boldsymbol{v}}'_0)$.*

There is an extension of the pebble game, $\mathrm{BP}_k(G, G')$, without specifying the starting position. Specifically, at the beginning of the game, Player II is first asked to provide a bijection $g : V(G)^k \to V(G')^k$. Then Player I chooses $\overrightarrow{\boldsymbol{v}}_0 \in V(G)^k$. Then Player I and Player II start to play $\mathrm{BP}_k(G, \overrightarrow{\boldsymbol{v}}_0, G', g(\overrightarrow{\boldsymbol{v}}_0))$.

**Theorem 10.** *$k$-WL cannot distinguish $G$ and $G'$ at step $t$, i.e. $\boldsymbol{gwl}_k^{(t)}(G) = \boldsymbol{gwl}_k^{(t)}(G')$, if and only if Player II has a winning strategy for $t$-step $\mathrm{BP}_k(G, G')$.*

The proof of Theorem 9 and Theorem 10 can be found in Hella's work [30].

### A.3 Doubly Bijective $k$-Pebble Game for $k$-MultisetWL

Let $G, G'$ be graphs and $\tilde{v}_0, \tilde{v}'_0$ be $k$-multisets. We adapt the $\mathrm{BP}_k$ game for $k$-MULTISETWL and call it doubly bijective $k$-pebble game, i.e. $\mathrm{DBP}_k(G, \tilde{v}_0, G', \tilde{v}'_0)$. A similar version of the pebble game for $k$-MULTISETWL was suggested by Grohe in [27].

The $\mathrm{DBP}_k(G, \tilde{v}_0, G', \tilde{v}'_0)$ is defined as the follows. The game starts at the position $(\tilde{v}_0, \tilde{v}'_0)$.

Let the current position be $(\tilde{v}_t, \tilde{v}'_t)$. The $t$-th round is played as follows.

1. Player II chooses a bijection $h : \tilde{v}_t \to \tilde{v}'_t$
2. Player I chooses the $y \in \tilde{v}_t$
3. Player II chooses bijection $f : V(G) \to V(G')$
4. Player I chooses $x \in V(G)$
5. The new position is $(\tilde{v}_t[x/\mathrm{idx}_{\tilde{v}_t}(y)], \tilde{v}'_t[f(x)/\mathrm{idx}_{\tilde{v}'_t}(h(y))])$ and the game continues, if $G[\tilde{v}_t[x/\mathrm{idx}_{\tilde{v}_t}(y)]]$ and $G'[\tilde{v}'_t[f(x)/\mathrm{idx}_{\tilde{v}_t}(h(y))]]$ are isomorphic. Otherwise the game ends and Player II loses.

To extend the game to the setting of no starting position, we define the corresponding game $\mathrm{DBP}_k(G, G')$. Similiar to $\mathrm{BP}_k(G, G')$, at the beginning Player II is asked to pick up a bijection $g : \mathrm{Multiset}(V(G)^k) \to \mathrm{Multiset}(V(G')^k)$. Then Player I picks up a $\tilde{v}_0$ and the game $\mathrm{DBP}_k(G, \tilde{v}_0, G', g(\tilde{v}_0))$ starts.

**Theorem 3.** *$k$-MULTISETWL has the same expressivity as the doubly bijective $k$-pebble game.*

*Proof.* Specifically, given graph $G$ with $k$-multiset $\tilde{v}$ and graph $G'$ with $k$-multiset $\tilde{v}'$, we are going to prove that $\boldsymbol{mwl}_k^{(t)}(G, \tilde{v}) = \boldsymbol{mwl}_k^{(t)}(G', \tilde{v}') \iff$ Player II has a winning strategy for $t$-step game $\mathrm{DBP}_k(G, \tilde{v}, G', \tilde{v}')$.

We now prove it by induction on $t$. When $t = 0$, it's obvious that the statement is correct, as $\boldsymbol{mwl}_k^{(0)}(G, \tilde{v}) = \boldsymbol{mwl}_k^{(0)}(G', \tilde{v}')$ is equivalent to $G[\tilde{v}]$ and $G'[\tilde{v}']$ being isomorphic, which implies that Player II can start the game without losing. Now assume that for $t \leq l$ the statement is correct. For step $t = l + 1$, let's first prove left $\implies$ right.

$\boldsymbol{mwl}_k^{(l+1)}(G, \tilde{v}) = \boldsymbol{mwl}_k^{(l+1)}(G', \tilde{v}')$. By Eq.(3) we know this is equivalent to
(1) $\boldsymbol{mwl}_k^{(l)}(G, \tilde{v}) = \boldsymbol{mwl}_k^{(l)}(G', \tilde{v}')$
(2) $\exists$ bijective $h : \tilde{v} \to \tilde{v}'$, $\forall y \in \tilde{v}$, $\{\!\{\boldsymbol{mwl}_k^{(l)}(G, \tilde{v}[x/\mathrm{idx}_{\tilde{v}}(y)]) \mid x \in V(G)\}\!\} = \{\!\{\boldsymbol{mwl}_k^{(l)}(G', \tilde{v}'[x/\mathrm{idx}_{\tilde{v}'}(h(y))]) \mid x \in V(G')\}\!\}$.

Now let's start the $\text{DBP}_k$ game at position $(\tilde{\boldsymbol{v}}, \tilde{\boldsymbol{v}}')$. By (2) we know that there exist a $h$ satisfying (2). Then at the first round, we as Player II pick the $h$ as the bijection. Next Player I will choose a $y \in \tilde{\boldsymbol{v}}$. According to (2), for the $h$ and $y$ we have $\{\!\!\{\boldsymbol{mwl}_k^{(l)}(G, \tilde{\boldsymbol{v}}[x/\text{idx}_{\tilde{\boldsymbol{v}}}(y)]) \mid x \in V(G)\}\!\!\} = \{\!\!\{\boldsymbol{mwl}_k^{(l)}(G', \tilde{\boldsymbol{v}}'[x/\text{idx}_{\tilde{\boldsymbol{v}}'}(h(y))]) \mid x \in V(G')\}\!\!\}$. This implies that there exists a bijection $f : V(G) \to V(G')$ such that $\forall x \in V(G), \boldsymbol{mwl}_k^{(l)}(G, \tilde{\boldsymbol{v}}[x/\text{idx}_{\tilde{\boldsymbol{v}}}(y)]) = \boldsymbol{mwl}_k^{(l)}(G, \tilde{\boldsymbol{v}}'[f(x)/\text{idx}_{\tilde{\boldsymbol{v}}'}(h(y))])$. Hence let Player II pick the $f$. Now player I will choose a $x \in V(G)$. Then $\boldsymbol{mwl}_k^{(l)}(G, \tilde{\boldsymbol{v}}[x/\text{idx}_{\tilde{\boldsymbol{v}}}(y)]) = \boldsymbol{mwl}_k^{(l)}(G, \tilde{\boldsymbol{v}}'[f(x)/\text{idx}_{\tilde{\boldsymbol{v}}'}(h(y))])$ implies $\boldsymbol{mwl}_k^{(0)}(G, \tilde{\boldsymbol{v}}[x/\text{idx}_{\tilde{\boldsymbol{v}}}(y)]) = \boldsymbol{mwl}_k^{(0)}(G, \tilde{\boldsymbol{v}}'[f(x)/\text{idx}_{\tilde{\boldsymbol{v}}'}(h(y))])$, hence $G[\tilde{\boldsymbol{v}}_t[x/\text{idx}_{\tilde{\boldsymbol{v}}_t}(y)]]$ and $G'[\tilde{\boldsymbol{v}}'_t[f(x)/\text{idx}_{\tilde{\boldsymbol{v}}'_t}(h(y))]]$ are isomorphic. So the game doesn't end. At the next round, the game starts at the position $(\tilde{\boldsymbol{v}}[x/\text{idx}_{\tilde{\boldsymbol{v}}}(y)], \tilde{\boldsymbol{v}}'[f(x)/\text{idx}_{\tilde{\boldsymbol{v}}'}(h(y))])$, and we know $\boldsymbol{mwl}_k^{(l)}(G, \tilde{\boldsymbol{v}}[x/\text{idx}_{\tilde{\boldsymbol{v}}}(y)]) = \boldsymbol{mwl}_k^{(l)}(G, \tilde{\boldsymbol{v}}'[f(x)/\text{idx}_{\tilde{\boldsymbol{v}}'}(h(y))])$. By the inductive hypothesis, Player II has a strategy to play $\text{DBP}_k$ at the new position for $l$ rounds. Hence Player II has a winning strategy to play $\text{DBP}_k$ at original position $(\tilde{\boldsymbol{v}}, \tilde{\boldsymbol{v}}')$ for $l + 1$ rounds.

Next we prove right $\implies$ left by showing that $\boldsymbol{mwl}_k^{(l+1)}(G, \tilde{\boldsymbol{v}}) \neq \boldsymbol{mwl}_k^{(l+1)}(G', \tilde{\boldsymbol{v}}') \implies$ Player I has a winning strategy for $l + 1$-step game $\text{DBP}_k(G, \tilde{\boldsymbol{v}}, G', \tilde{\boldsymbol{v}'})$.

$\boldsymbol{mwl}_k^{(l+1)}(G, \tilde{\boldsymbol{v}}) \neq \boldsymbol{mwl}_k^{(l+1)}(G', \tilde{\boldsymbol{v}}')$ implies:
(1) $\boldsymbol{mwl}_k^{(l)}(G, \tilde{\boldsymbol{v}}) \neq \boldsymbol{mwl}_k^{(l)}(G', \tilde{\boldsymbol{v}}')$
or (2) for any bijection $h : \tilde{\boldsymbol{v}} \to \tilde{\boldsymbol{v}}', \exists y \in \tilde{\boldsymbol{v}}$, such that $\{\!\!\{\boldsymbol{mwl}_k^{(l)}(G, \tilde{\boldsymbol{v}}[x/\text{idx}_{\tilde{\boldsymbol{v}}}(y)]) \mid x \in V(G)\}\!\!\} \neq \{\!\!\{\boldsymbol{mwl}_k^{(l)}(G', \tilde{\boldsymbol{v}}'[x/\text{idx}_{\tilde{\boldsymbol{v}}'}(h(y))]) \mid x \in V(G')\}\!\!\}$

If (1) holds, then by induction we know that Player I has a winning strategy within $l$-steps hence Player I also has a winning strategy within $l + 1$ steps. If (2) holds, then at the first round after Player II picks up a bijection $h$, Player I can choose the specific $y \in \tilde{\boldsymbol{v}}$ with $\{\!\!\{\boldsymbol{mwl}_k^{(l)}(G, \tilde{\boldsymbol{v}}[x/\text{idx}_{\tilde{\boldsymbol{v}}}(y)]) \mid x \in V(G)\}\!\!\} \neq \{\!\!\{\boldsymbol{mwl}_k^{(l)}(G', \tilde{\boldsymbol{v}}'[x/\text{idx}_{\tilde{\boldsymbol{v}}'}(h(y))]) \mid x \in V(G')\}\!\!\}$. Then no matter which bijection $f : V(G) \to V(G')$ Player II chooses, Player I can always choose a $x \in V(G)$ such that $\boldsymbol{mwl}_k^{(l)}(G, \tilde{\boldsymbol{v}}[x/\text{idx}_{\tilde{\boldsymbol{v}}}(y)]) \neq \boldsymbol{mwl}_k^{(l)}(G', \tilde{\boldsymbol{v}}'[x/\text{idx}_{\tilde{\boldsymbol{v}}'}(h(y))])$. Then by induction, Player I has a winning strategy within $l$-steps for the $\text{DBP}_k$ starts at position $(\tilde{\boldsymbol{v}}[x/\text{idx}_{\tilde{\boldsymbol{v}}}(y)], \tilde{\boldsymbol{v}}'[x/\text{idx}_{\tilde{\boldsymbol{v}}'}(h(y))])$. Hence even if Player I doesn't win in the first round, he/she can still win in $l + 1$ rounds.

Combining both sides we know that the equivalence between $t$-step $\text{DBP}_k$ and $t$-step $k$-MULTISETWL holds for any $t$ and any $k$-multisets.

$\square$

**Theorem 11.** $k$-MULTISETWL *cannot distinguish $G$ and $G'$ at step $t$, i.e.* $\boldsymbol{gmwl}_k^{(t)}(G) = \boldsymbol{gmwl}_k^{(t)}(G')$ *, if and only if Player II has a winning strategy for $t$-step $DBP_k(G, G')$.*

*Proof.* The proof is strict forward with using the proof inside Theorem 3. We just need to show that the pooling of all $k$-multisets representations is equivalent to Player II finding a bijection $g : \text{Multiset}(V(G)^k) \to \text{Multiset}(V(G')^k)$ at the first step of the game. We omit that given its simplicity. $\square$

## A.4 Proofs of Theorems

### A.4.1 Bound of Summation of Binomial Coefficients

Derivation of:

$$\sum_{i=1}^{k} \binom{n}{i} \leq \binom{n}{k} \frac{n - k + 1}{n - 2k + 1} \tag{15}$$

*Proof.*

$$\frac{\sum_{i=1}^{k} \binom{n}{i}}{\binom{n}{k}} = \frac{\binom{n}{k} + \binom{n}{k-1} + \binom{n}{k-2} + \dots}{\binom{n}{k}} \tag{16}$$

$$= 1 + \frac{k}{n-k+1} + \frac{k(k-1)}{(n-k+1)(n-k+2)} + \dots \tag{17}$$

$$\leq 1 + \frac{k}{n-k+1} + \left(\frac{k}{n-k+1}\right)^2 + \dots \tag{18}$$

$$\leq \frac{n-k+1}{n-2k+1} \tag{19}$$

$\square$

### A.4.2 Proof of Theorem 1

**Theorem 1.** *Let $k \geq 1$ and $\boldsymbol{wl}_k^{(t)}(G, \tilde{\boldsymbol{v}}) := \{\!\{\boldsymbol{wl}_k^{(t)}(G, p(\tilde{\boldsymbol{v}})) | p \in perm[k]\}\!\}$. For all $t \in \mathbb{N}$ and all graphs $G, G'$: $k$-MULTISETWL is upper bounded by $k$-WL in distinguishing multisets $G, \tilde{\boldsymbol{v}}$ and $G', \tilde{\boldsymbol{v}}'$ at $t$-th iteration, i.e. $\boldsymbol{wl}_k^{(t)}(G, \tilde{\boldsymbol{v}}) = \boldsymbol{wl}_k^{(t)}(G', \tilde{\boldsymbol{v}}') \Longrightarrow \boldsymbol{mwl}_k^{(t)}(G, \tilde{\boldsymbol{v}}) = \boldsymbol{mwl}_k^{(t)}(G', \tilde{\boldsymbol{v}}')$.*

*Proof.* By induction on $t$. It's obvious that when $t = 0$ the above statement hold, as both side are equivalent to $G[\tilde{\boldsymbol{v}}]$ and $G'[\tilde{\boldsymbol{v}}']$ being isomorphic to each other. Assume $\leq t$ the above statement is true. For $t + 1$ case, by definition the left side is equivalent to $\{\!\{\boldsymbol{wl}_k^{(t+1)}(G, p(\tilde{\boldsymbol{v}})) | p \in perm[k]\}\!\} = \{\!\{\boldsymbol{wl}_k^{(t+1)}(G', p(\tilde{\boldsymbol{v}}')) | p \in perm[k]\}\!\}$. Let $\overrightarrow{\boldsymbol{v}}$ be the ordered version of $\tilde{\boldsymbol{v}}$ following canonical ordering over $G$, then there exists a bijective mapping $f$ between $\tilde{\boldsymbol{v}}$ and $\tilde{\boldsymbol{v}}'$, such that $\boldsymbol{wl}_k^{(t+1)}(G, \overrightarrow{\boldsymbol{v}}) = \boldsymbol{wl}_k^{(t+1)}(G', f(\overrightarrow{\boldsymbol{v}}))$, where $f(\overrightarrow{\boldsymbol{v}}) := (f(\overrightarrow{\boldsymbol{v}}_1), ..., f(\overrightarrow{\boldsymbol{v}}_k))$. By [11]'s Theorem 5.2, for any $t$, $\boldsymbol{wl}_k^{(t)}(G, \overrightarrow{\boldsymbol{v}}) = \boldsymbol{wl}_k^{(t)}(G', f(\overrightarrow{\boldsymbol{v}}))$ is equivalent to that player II has a winner strategy for $t$-step pebble game with initial configuration $G, \overrightarrow{\boldsymbol{v}}$ and $G', f(\overrightarrow{\boldsymbol{v}})$ (please refer to [11] for the description of pebble game). Notice that applying any permutation to the pebble game's initial configuration won't change having winner strategy for player II, hence we know that $\forall p \in perm[k]$, $\boldsymbol{wl}_k^{(t+1)}(G, p(\overrightarrow{\boldsymbol{v}})) = \boldsymbol{wl}_k^{(t+1)}(G', p(f(\overrightarrow{\boldsymbol{v}})))$. Now applying Eq.(1), we know that (1) $\boldsymbol{wl}_k^{(t)}(G, p(\overrightarrow{\boldsymbol{v}})) = \boldsymbol{wl}_k^{(t)}(G', p(f(\overrightarrow{\boldsymbol{v}})))$, and (2)$\forall i \in [k]$, $\{\!\{\boldsymbol{wl}_k^{(t)}(G, p(\overrightarrow{\boldsymbol{v}}[x/i])) \mid x \in V(G)\}\!\} = \{\!\{\boldsymbol{wl}_k^{(t)}(G', p(f(\overrightarrow{\boldsymbol{v}})[x/i])) \mid x \in V(G')\}\!\}$. We rewrite (2) as, $\forall i \in [k]$, $\exists$ bijective $h_i : V(G) \to V(G')$, $\forall x \in V(G)$, $\boldsymbol{wl}_k^{(t)}(G, p(\overrightarrow{\boldsymbol{v}}[x/i])) = \boldsymbol{wl}_k^{(t)}(G', p(f(\overrightarrow{\boldsymbol{v}})[h_i(x)/i]))$. As (1) and (2) hold for any $p \in perm[k]$, now applying induction hypothesis to both (1) and (2), we can get (a) $\boldsymbol{mwl}_k^{(t)}(G, \tilde{\boldsymbol{v}}) = \boldsymbol{mwl}_k^{(t)}(G', \tilde{\boldsymbol{v}}')$, and (b) $\forall i \in [k]$, $\exists$ bijective $h_i : V(G) \to V(G')$, $\forall x \in V(G)$, $\boldsymbol{mwl}_k^{(t)}(G, \tilde{\boldsymbol{v}}[x/i]) = \boldsymbol{mwl}_k^{(t)}(G', \tilde{\boldsymbol{v}}'[h_i(x)/g(i)])$, where $g : [k] \to [k]$ is the index mapping function corresponding to $f$. Now we rewrite (b) as $\exists g : [k] \to [k]$, $\forall i \in [k]$, $\{\!\{\boldsymbol{mwl}_k^{(t)}(G, \tilde{\boldsymbol{v}}[x/i]) \mid x \in V(G)\}\!\} = \{\!\{\boldsymbol{mwl}_k^{(t)}(G', \tilde{\boldsymbol{v}}'[x/g(i)]) \mid x \in V(G')\}\!\}$. Combining (a) and (b), using Eq.(3) we can get $\boldsymbol{mwl}_k^{(t+1)}(G, \tilde{\boldsymbol{v}}) = \boldsymbol{mwl}_k^{(t+1)}(G', \tilde{\boldsymbol{v}}')$. Thus for any $t$ the above statement is correct. $\square$

### A.4.3 Proof of Theorem 2

#### CFI Graphs and Their Properties

Cai, Furer and Immerman [11] designed a construction of a series of pairs of graphs CFI($k$) such that for any $k$, $k$-WL cannot distinguish the pair of graphs CFI($k$). We will use the CFI graph construction to prove the theorem. Here we use a variant of CFI construction that is proposed in Grohe et al.'s work [28]. Let $\mathcal{K}$ he a complete graph with $k$ nodes. We first construct an enlarged graph $\mathcal{X}(\mathcal{K})$ from the base graph $\mathcal{K}$ by mapping each node and edge to a group of vertices and connecting all ($|V(\mathcal{K})| + |E(\mathcal{K})|$) groups of vertices following certain rules. Notice that we use node for base graph and use vertex for the enlarged graph for a distinction. Let $vw$ denotes the edge connecting node $v$ and node $w$. For a node $v \in V(\mathcal{K})$, let $E(v) := \{vw | vw \in E(\mathcal{K})\}$ denotes the set of all adjacent edges of $v$ in the graph $\mathcal{K}$.

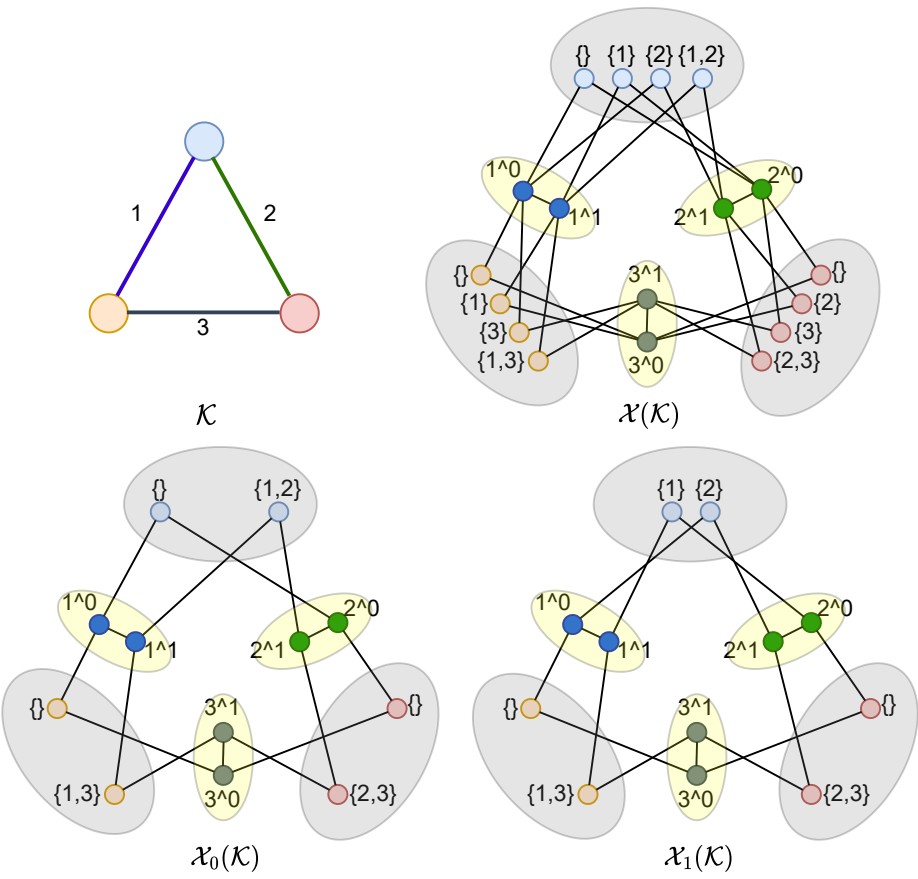

Figure 4: CFI($k$) construction visualization for $k = 3$

For every node $v \in V(\mathcal{K})$, we map node $v$ to a group of vertices $S_v := \{v^X | X \subseteq E(v)\}$ with size $|S_v| = 2^{deg(v)} = 2^{k-1}$. For every edge $e \in E(\mathcal{K})$, the construction maps $e$ to two vertices $S_e := \{e^0, e^1\}$. Hence there are $2 * |E(\mathcal{K})| + |V(\mathcal{K})| * 2^{k-1} = k(k-1) + k(2^{k-1})$ number of vertices in the enlarged graph $\mathcal{X}(\mathcal{K})$ with $V(\mathcal{X}(\mathcal{K})) = (\cup_{v \in V(\mathcal{K})} S_v) \cup (\cup_{e \in E(\mathcal{K})} S_e)$. Let $V^* := \cup_{v \in V(\mathcal{K})} S_v$ and $E^* := \cup_{e \in E(\mathcal{K})} S_e$.

Now edges inside $\mathcal{X}(\mathcal{K})$ are defined as follows

$$E(\mathcal{X}(\mathcal{K})) := \{v^X e^1 \mid v \in V(\mathcal{K}), X \subseteq E(v), \text{and } e \in X\} \cup \tag{20}$$

$$\{v^X e^0 \mid v \in V(\mathcal{K}), X \subseteq E(v), \text{and } e \notin X\} \cup \{e^0 e^1 \mid e \in E(\mathcal{K})\} \tag{21}$$

What's more, we also color the vertices such that all vertices inside $S_v$ have color $v$ for every $v \in V(\mathcal{K})$ and all vertices inside $S_e$ have color $e$ for every $e \in E(\mathcal{K})$. See Figure.4 top right for the visualization of transforming from base graph $\mathcal{K}$ to $\mathcal{X}(\mathcal{K})$ with $k = 3$.

There are several important properties about the automorphisms of $\mathcal{X}(\mathcal{K})$. Let $h \in \text{Aut}(\mathcal{X}(\mathcal{K}))$ be an automorphism, then

1. $h(S_v) = S_v$ and $h(S_e) = S_e$ for all $v \in V(\mathcal{K})$ and $e \in E(\mathcal{K})$.

2. For every subset $F \subseteq E(\mathcal{K})$, there is exactly one automorphism $h_F$ that flips precisely all edges in F, i.e. $h_F(e^0) = e^1$ and $h_F(e^1) = e^0$ if and only if $e \in F$. More specifically,
   - $h_F(e^i) = e^{1-i}, \forall e \in F$
   - $h_F(e^i) = e^i, \forall e \notin F$
   - $h_F(v^X) = v^Y, \forall v \in E(\mathcal{K}), X \subseteq E(v)$
     where $Y := X \triangle (F \cap E(v)) = \left(X \setminus (F \cap E(v))\right) \cup \left((F \cap E(v)) \setminus X\right)$

*Proof.* These properties are not hard to prove. First for property 1, it is true based on the coloring rules of vertices in $\mathcal{X}(\mathcal{K})$. Now for the second property. As $h_F$ flips precisely all edges in $F$, we have $h_F(e^i) = e^{1-i}, \forall e \in F$ and $h_F(e^i) = e^i, \forall e \notin F$. Now let $h_F(v^X) = v^Y \; \forall v \in E(\mathcal{K}), X \subseteq E(v)$, we need to figure out $Y$'s formulation. Let's focus on node $v$ without losing generality, and we can partition the set $E(v)$ to two parts: $E(v) \cap F$ and $E(v) \setminus F$. Then for any $e \in E(v) \cap F$, we know that $h_F(e^i) = e^{1-i}$. Let $v \leftrightarrow w$ denote that $v$ and $w$ are connected in $\mathcal{X}(\mathcal{K})$. Then $\forall X \subseteq E(v)$, $e \in X \iff e^0 \leftrightarrow v^X \iff h_F(e^0) \leftrightarrow h_F(v^X) \iff e^1 \leftrightarrow v^Y \iff e \notin Y$. And similarly we have $\forall X \subseteq E(v), e \notin X \iff e \in Y$. Hence $\forall e \in E(v) \cap F, \forall X \subseteq E(v), e \in X \triangle Y$. This implies that $E(v) \cap F \subseteq X \triangle Y$. Following the same logic we can also get $\forall e \in E(v) \setminus F, \forall X \subseteq E(v)$, $e \in E(v) \setminus (X \triangle Y)$, which is equivalent to $E(v) \setminus F \subseteq E(v) \setminus (X \triangle Y)$, which further implies that $E(v) \cap F \supseteq X \triangle Y$ as $X \triangle Y \subseteq E(v)$. Then combining both side we know that $\forall v \in E(\mathcal{K}), \forall X \subseteq E(v), E(v) \cap F = X \triangle Y$. Hence we get $Y = X \triangle (X \triangle Y) = X \triangle (F \cap E(v))$. $\qquad\square$

In the proof we can also know another important property. That is $\forall v, X \subseteq E(v), X \triangle h_F(X) = E(v) \cap F$ is constant for any input $X \in E(v)$.

Now we are ready to construct variants of graphs that are not isomorphic from the enlarged graph $\mathcal{X}(\mathcal{K})$. Now let T be a subset of $V(\mathcal{K})$. Now we define an induced subgraph $\mathcal{X}_T(\mathcal{K})$ of the enlarged graph $\mathcal{X}(\mathcal{K})$. Specifically, we define the new node group as follows

$$S_v^T := \begin{cases} \{v^X \in S_v \mid |X| \equiv 0 \mod 2\} \text{ if } v \notin T \\ \{v^X \in S_v \mid |X| \equiv 1 \mod 2\} \text{ if } v \in T \end{cases} \tag{22}$$

Then the induced subgraph is defined as $\mathcal{X}_T(\mathcal{K}) := \mathcal{X}(\mathcal{K})[(\cup_{v \in V(\mathcal{K})} S_v^T) \cup E^*]$. In Figure 4 we show $\mathcal{X}_\emptyset(\mathcal{K})$ in bottom left (labeled with $\mathcal{X}_0(\mathcal{K})$) and $\mathcal{X}_{\{v_1\}}(\mathcal{K})$ in the bottom right (labeled with $\mathcal{X}_1(\mathcal{K})$).

**Lemma 1.** *For all $T, U \subseteq V(\mathcal{K})$, $\mathcal{X}_T(\mathcal{K}) \cong \mathcal{X}_U(\mathcal{K})$ if and only if $|T| \equiv |U| \mod 2$. And if they are isomorphic, one isomorphism between $\mathcal{X}_T(\mathcal{K})$ and $\mathcal{X}_U(\mathcal{K})$ is $h_F$ with $F = E(\mathcal{K}[T \triangle U])$, where $E(\mathcal{K}[T \triangle U])$ denotes the set of all edges $\{v_i, v_j\} \subseteq T \triangle U$.*

Notice that $h_F$ is an automorphism for $\mathcal{X}(\mathcal{K})$, but with restricting its domain to $(\cup_{v \in V(\mathcal{K})} S_v^T) \cup E^*$ it becomes the isomorphism between $\mathcal{X}_T(\mathcal{K})$ and $\mathcal{X}_U(\mathcal{K})$.

The proof of the lemma can be found in [11] and [28]. With this lemma we know that $\mathcal{X}_{\{v_1\}}(\mathcal{K})$ and $\mathcal{X}_\emptyset(\mathcal{K})$ are not isomorphic. In next part we show that $\mathcal{X}_\emptyset(\mathcal{K})$ and $\mathcal{X}_{\{v_1\}}(\mathcal{K})$ cannot be distinguished by $(k-1)$-WL but can be distinguished by $k$-MULTISETWL, thus proving Theorem 2.

### Main Proof

**Theorem 2.** $k$-MULTISETWL *is no less powerful than (k-1)-WL in distinguishing graphs: for $k \geq 3$ there is a pair of graphs that can be distinguished by $k$-MULTISETWL but not by (k-1)-WL.*

*Proof.* To prove the theorem, we show that for any $k \geq 3$, $\mathcal{X}_{\{v_1\}}(\mathcal{K})$ and $\mathcal{X}_\emptyset(\mathcal{K})$, defined previously, are two nonisomorphic graphs that can be distinguished by $k$-MULTISETWL but not by $(k-1)$-WL. It's well known that these two graphs cannot be distinguished by $(k-1)$-WL, and one can refer to Theorem 5.17 in [28] for its proof. Now to prove these two graphs can be distinguished by $k$-MULTISETWL, using Theorem 11 we know it's equivalent to show that Player I has a winning strategy for the doubly bijective $k$-pebble game $\text{DBP}_k(\mathcal{X}_\emptyset(\mathcal{K}), \mathcal{X}_{\{v_1\}}(\mathcal{K}))$.

At the start of the pebble game $\text{DBP}_k(\mathcal{X}_\emptyset(\mathcal{K}), \mathcal{X}_{\{v_1\}}(\mathcal{K}))$, Player II is asked to provide a bijection between all $k$-multisets of $\mathcal{X}_\emptyset(\mathcal{K})$ to all $k$-multisets of $\mathcal{X}_{\{v_1\}}(\mathcal{K})$. For any bijection the Player II chosen, the Player I can pickup $\tilde{\boldsymbol{v}}_0 = \{\!\{v_1^\emptyset, v_2^\emptyset, ..., v_k^\emptyset\}\!\}$, with corresponding position $\tilde{\boldsymbol{v}}_0' = \{\!\{v_1^{X_1}, v_2^{X_2}, ..., v_k^{X_k}\}\!\}$. Notice that for a position $(\tilde{\boldsymbol{x}}, \tilde{\boldsymbol{y}}) := (\{\!\{x_1, ..., x_k\}\!\}, \{\!\{y_1, ..., y_k\}\!\})$, the position is called *consistent* if there exists a $F \subseteq E(\mathcal{K})$, such that $h_F(\tilde{\boldsymbol{x}}) = \{\!\{h_F(x_1), ..., h_F(x_k)\}\!\} = \tilde{\boldsymbol{y}}$. One can show that Player I can easily win the game $\text{DBP}_k(\mathcal{X}_\emptyset(\mathcal{K}), \mathcal{X}_{\{v_1\}}(\mathcal{K}))$ after one additional step if the current position $(\tilde{\boldsymbol{x}}, \tilde{\boldsymbol{y}})$ is not consistent, even if $\mathcal{X}_\emptyset(\mathcal{K})[\tilde{\boldsymbol{x}}]$ and $\mathcal{X}_{\{v_1\}}(\mathcal{K})[\tilde{\boldsymbol{y}}]$ are isomorphic [28].

We claim that for any $k$-multiset $\tilde{\boldsymbol{v}}_0' = \{\!\{v_1^{X_1}, v_2^{X_2}, ..., v_k^{X_k}\}\!\}$ the position $(\tilde{\boldsymbol{v}}_0, \tilde{\boldsymbol{v}}_0')$ cannot be consistent, and thus Player II loses $\text{DBP}_k(\mathcal{X}_\emptyset(\mathcal{K}), \mathcal{X}_{\{v_1\}}(\mathcal{K}))$ after the first round. Assume that there

exists a subset $F \subseteq E(\mathcal{K})$, such that $h_F(\tilde{\boldsymbol{v}}_0) = \tilde{\boldsymbol{v}}_0'$. By property 1 of automorphisms of $\mathcal{X}(\mathcal{K})$ we have that $h_F(S_{v_1}) = S_{v_1}$, and since $\tilde{\boldsymbol{v}}_0$ contains exactly one vertex of each color, it also holds that $h_F(v_1^{\emptyset}) = v_1^{X_1}$. It follows from property 2 that $X_1 = \emptyset \triangle \big(F \cap E(v_1)\big) = F \cap E(v_1)$. Based on the definition of $\mathcal{X}_{\{v_1\}}(\mathcal{K})$, we know that $|X_1| \equiv 1 \mod 2$, and thus $F$ flips an odd number of neighbors of $v_1$, i.e. $|F \cap E(v_1)| \equiv 1 \mod 2$. Similarly, we have that $h_F(S_{v_i}) = S_{v_i}$ and $|X_i| \equiv 0 \mod 2, \forall i \geq 2$, hence $|F \cap E(v_i)| \equiv 0 \mod 2$. It follows from a simple handshake argument that there exists an $m \geq 0$ such that

$$2m = \sum_{v \in V} |(F \cap E)(v)| = \sum_{i \in [k] \setminus \{1\}} |F \cap E(v_i)| + |F \cap E(v_1)| \equiv 1 \mod 2,$$

which is a contradiction. Hence Player I has a winning strategy for $\mathrm{DBP}_k(\mathcal{X}_{\emptyset}(\mathcal{K}), \mathcal{X}_{\{v_1\}}(\mathcal{K}))$.

$\square$

### A.4.4 Proof of Theorem 3

See Appendix.

### A.4.5 Proof of Theorem 4

**Theorem 4.** *Let $k \geq 1$, then $\forall t \in \mathbb{N}$ and all graphs $G, G'$: $\boldsymbol{mwl}_k^{(t)}(G, \hat{\boldsymbol{v}}) = \boldsymbol{mwl}_k^{(t)}(G', \hat{\boldsymbol{v}}') \implies \boldsymbol{swl}_k^{(t)}(G, \hat{\boldsymbol{v}}) = \boldsymbol{swl}_k^{(t)}(G', \hat{\boldsymbol{v}}').$*

*Proof.* **Notation:** Let $s(\cdot)$ transform a multiset to set by removing repeats, and let $r(\cdot)$ return a tuple with the number of repeats for each distinct element in a multiset. Let $f$ be the inverse mapping such that $\tilde{\boldsymbol{v}} = f(s(\tilde{\boldsymbol{v}}), r(\tilde{\boldsymbol{v}}))$.

Define $F^{(t+1)}(G, G', \tilde{\boldsymbol{v}}, \tilde{\boldsymbol{v}}') := \{$ injective $f : \tilde{\boldsymbol{v}} \to \tilde{\boldsymbol{v}}' \mid f \in F^{(t)}(G, G', \tilde{\boldsymbol{v}}, \tilde{\boldsymbol{v}}')$, AND , $\forall y \in \tilde{\boldsymbol{v}}, \exists h_y : V(G) \to V(G'), \forall x, f \in F^{(t)}(G, G', \tilde{\boldsymbol{v}}[x/\mathrm{idx}_{\tilde{\boldsymbol{v}}}(y)], \tilde{\boldsymbol{v}}'[h_y(x)/\mathrm{idx}_{\tilde{\boldsymbol{v}}'}(f(y))])\}$. Let $F^{(0)}(G, G', \tilde{\boldsymbol{v}}, \tilde{\boldsymbol{v}}') := \{f \mid f \text{ is an isomorphism from } G[\tilde{\boldsymbol{v}}] \text{ to } G'[\tilde{\boldsymbol{v}}']\}$.

**Lemma 2.** $\forall t$, $\boldsymbol{mwl}_k^{(t)}(G, \tilde{\boldsymbol{v}}) = \boldsymbol{mwl}_k^{(t)}(G', \tilde{\boldsymbol{v}}') \iff \forall h \in F^{(t)}(G, G', \tilde{\boldsymbol{v}}, \tilde{\boldsymbol{v}}')$, $\forall \hat{\boldsymbol{n}}$ with $(\sum_{i=1}^m \hat{\boldsymbol{n}}_i = k, \forall i \; \hat{\boldsymbol{n}}_i \geq 1)$, $\boldsymbol{mwl}_k^{(t)}(G, f(s(\tilde{\boldsymbol{v}}), \hat{\boldsymbol{n}})) = \boldsymbol{mwl}_k^{(t)}(G', f(h(s(\tilde{\boldsymbol{v}})), \hat{\boldsymbol{n}}))$.

When $t = 0$, by the definition of $\boldsymbol{swl}_k^{(0)}$, the statement is true. Now hypothesize that the statement is true for $\leq t$ case. For $t + 1$ case, the left side implies that existing a $\tilde{\boldsymbol{v}}$ with $s(\tilde{\boldsymbol{v}}) = \hat{\boldsymbol{v}}$ and a $\tilde{\boldsymbol{v}}'$ with $s(\tilde{\boldsymbol{v}}') = \hat{\boldsymbol{v}}'$, such that $\boldsymbol{mwl}_k^{(t+1)}(G, \tilde{\boldsymbol{v}}) = \boldsymbol{mwl}_k^{(t+1)}(G', \tilde{\boldsymbol{v}}')$. And for a mapping $h \in F^{(t+1)}(G, G', \tilde{\boldsymbol{v}}, \tilde{\boldsymbol{v}}')$, we have $\forall \hat{\boldsymbol{n}}$ with $(\sum_{i=1}^m \hat{\boldsymbol{n}}_i = k, \forall i \; \hat{\boldsymbol{n}}_i \geq 1)$, $\boldsymbol{mwl}_k^{(t+1)}(G, f(s(\tilde{\boldsymbol{v}}), \hat{\boldsymbol{n}})) = \boldsymbol{mwl}_k^{(t+1)}(G', f(h(s(\tilde{\boldsymbol{v}})), \hat{\boldsymbol{n}}))$.

For a specific $\hat{\boldsymbol{n}}$, define $\tilde{\boldsymbol{u}} = f(s(\tilde{\boldsymbol{v}}), \hat{\boldsymbol{n}})$ and $\tilde{\boldsymbol{u}}' = f(h(s(\tilde{\boldsymbol{v}})), \hat{\boldsymbol{n}}) = h(\tilde{\boldsymbol{u}})$. By Eq.(3):
(1) $\boldsymbol{mwl}_k^{(t)}(G, \tilde{\boldsymbol{u}}) = \boldsymbol{mwl}_k^{(t)}(G', \tilde{\boldsymbol{u}}')$;
(2) $\forall f \in F^{(t+1)}(G, G', \tilde{\boldsymbol{u}}, \tilde{\boldsymbol{u}}')$, $\forall y \in \tilde{\boldsymbol{u}}$, $\{\!\{\boldsymbol{mwl}_k^{(t)}(G, \tilde{\boldsymbol{u}}[x/\mathrm{idx}_{\tilde{\boldsymbol{v}}}(y)]) \big| x \in V(G)\}\!\} = \{\!\{\boldsymbol{mwl}_k^{(t)}(G', \tilde{\boldsymbol{u}}'[x/\mathrm{idx}_{\tilde{\boldsymbol{v}}'}(f(y))]) \big| x \in V(G')\}\!\}$.

As $h \in F^{(t+1)}(G, G', \tilde{\boldsymbol{v}}, \tilde{\boldsymbol{v}}') = F^{(t+1)}(G, G', \tilde{\boldsymbol{u}}, \tilde{\boldsymbol{u}}')$, we can change (2) by choosing $f$ as $h$. Hence we update it as:
(2) $\forall y \in \tilde{\boldsymbol{u}}$, $\{\!\{\boldsymbol{mwl}_k^{(t)}(G, \tilde{\boldsymbol{u}}[x/\mathrm{idx}_{\tilde{\boldsymbol{v}}}(y)]) \big| x \in V(G)\}\!\} = \{\!\{\boldsymbol{mwl}_k^{(t)}(G', h(\tilde{\boldsymbol{u}})[x/\mathrm{idx}_{\tilde{\boldsymbol{v}}'}(h(y))]) \big| x \in V(G')\}\!\}$
And (2) can be split into two parts:
(2) $\forall y \in \tilde{\boldsymbol{u}}$, $\{\!\{\boldsymbol{mwl}_k^{(t)}(G, \tilde{\boldsymbol{u}}[x/\mathrm{idx}_{\tilde{\boldsymbol{v}}}(y)]) \big| x \in s(\tilde{\boldsymbol{u}})\}\!\} = \{\!\{\boldsymbol{mwl}_k^{(t)}(G', h(\tilde{\boldsymbol{u}})[x/\mathrm{idx}_{\tilde{\boldsymbol{v}}'}(h(y))]) \big| x \in s(h(\tilde{\boldsymbol{u}}))\}\!\}$ and
$\{\!\{\boldsymbol{mwl}_k^{(t)}(G, \tilde{\boldsymbol{u}}[x/\mathrm{idx}_{\tilde{\boldsymbol{v}}}(y)]) \big| x \in V(G) \setminus s(\tilde{\boldsymbol{u}})\}\!\} = \{\!\{\boldsymbol{mwl}_k^{(t)}(G', h(\tilde{\boldsymbol{u}})[x/\mathrm{idx}_{\tilde{\boldsymbol{v}}'}(h(y))]) \big| x \in V(G') \setminus s(h(\tilde{\boldsymbol{u}}))\}\!\}$

Combining the Lemma.2 and induction hypothesis, also knowing that $s(\tilde{u}) = \hat{v}$ and $s(\tilde{u}') = \hat{v}'$, we get:

(a) $swl_k^{(t)}(G, \hat{v}) = swl_k^{(t)}(G', \hat{v}')$;

(b) $\forall y \in \hat{v}$, $\{\!\{swl_k^{(t)}(G, \hat{v} \setminus y) | x \in \hat{v}\}\!\} = \{\!\{swl_k^{(t)}(G', \hat{v}' \setminus h(y)) | x \in \hat{v}'\}\!\}$, this is derived with picking $\hat{n}$ with $\hat{n}[y] = 1$, such that $\tilde{u}$ only has 1 $y$ and $\tilde{u}'$ only has 1 $h(y)$.

(c) $\forall y \in \hat{v}$, $\{\!\{swl_k^{(t)}(G, \hat{v} \cup \{x\}) | x \in V(G) \setminus \hat{v}\}\!\} = \{\!\{swl_k^{(t)}(G', \hat{v}' \cup \{x\}) | x \in V(G') \setminus \hat{v}'\}\!\}$, this is derived with picking $\hat{n}$ with $\hat{n}[y] > 1$.

(d) $\forall y \in \hat{v}$, $\{\!\{swl_k^{(t)}(G, \hat{v} \setminus y \cup \{x\}) | x \in V(G) \setminus \hat{v}\}\!\} = \{\!\{swl_k^{(t)}(G', \hat{v}' \setminus h(y) \cup \{x\}) | x \in V(G') \setminus \hat{v}'\}\!\}$, this is derived with picking $\hat{n}$ with $\hat{n}[y] = 1$, such that $\tilde{u}$ only has 1 $y$ and $\tilde{u}'$ only has 1 $h(y)$.

Now combining (a) (b) (c) (d) and using Eq.(5), we can get that $swl_k^{(t+1)}(G, \hat{v}) = swl_k^{(t+1)}(G', \hat{v}')$, and we proved the statement.

$\square$

### A.4.6 Proof of Theorem 5

**Theorem 5.** *Let $k \geq 1$, then $\forall t \in \mathbb{N}$ and all graphs $G, G'$:*
  *(1) when $1 \leq c_1 < c_2 \leq k$, if $G, G'$ cannot be distinguished by $(k, c_2)(\leq)$-SETWL, they cannot be distinguished by $(k, c_1)(\leq)$-SETWL*
  *(2) when $k_1 < k_2$, $\forall c \leq k_1$, if $G, G'$ cannot be distinguished by $(k_2, c)(\leq)$-SETWL, they cannot be distinguished by $(k_1, c)(\leq)$-SETWL*

*Proof.* We will prove (2) first, and the proof for (1) follows the same argument. To help understand, we present the formulation for $(k, c)(\leq)$-SETWL first.

$$swl_{k,c}^{(t+\frac{1}{2})}(G, \hat{v}) = \{\!\{swl_{k,c}^{(t)}(G, \hat{u}) \mid \hat{u} \in \mathcal{N}_{k,c,\text{right}}^G(\hat{v})\}\!\} \tag{23}$$

$$swl_{k,c}^{(t+1)}(G, \hat{v}) = \left(swl_{k,c}^{(t)}(G, \hat{v}), swl_{k,c}^{(t+\frac{1}{2})}(G, \hat{v}), \{\!\{swl_{k,c}^{(t)}(G, \hat{u}) \mid \hat{u} \in \mathcal{N}_{k,c,\text{left}}^G(\hat{v})\}\!\}, \tag{24}$$

$$\{\!\{swl_{k,c}^{(t+\frac{1}{2})}(G, \hat{u}) \mid \hat{u} \in \mathcal{N}_{k,c,\text{left}}^G(\hat{v})\}\!\}\right) \tag{25}$$

**Lemma 3.** *For $k_1 < k_2$, for any $t$, for any $\hat{v}$ and $\hat{v}'$ with $\leq k_1$ nodes inside, $swl_{k_2,c}^{(t)}(G, \hat{v}) = swl_{k_2,c}^{(t)}(G', \hat{v}') \Longrightarrow swl_{k_1,c}^{(t)}(G, \hat{v}) = swl_{k_1,c}^{(t)}(G', \hat{v}')$.*

*Proof of Lemma.3:*

*Proof.* We prove it by induction. As the color initialization stage of $(k_1, c)(\leq)$-SETWL and $(k_2, c)(\leq)$-SETWL are the same, when $t = 0$ the statement of the Lemma is correct. Assume it holds correct for $\leq t$. When $t + 1$, $swl_{k_2,c}^{(t+1)}(G, \hat{v}) = swl_{k_2,c}^{(t+1)}(G', \hat{v}')$ implies:

(1) $swl_{k_2,c}^{(t)}(G, \hat{v}) = swl_{k_2,c}^{(t)}(G', \hat{v}')$
(2) $swl_{k_2,c}^{(t+1/2)}(G, \hat{v}) = swl_{k_2,c}^{(t+1/2)}(G', \hat{v}')$
(3) $\{\!\{swl_{k_2,c}^{(t)}(G, \hat{u}) \mid \hat{u} \in \mathcal{N}_{k_2,c,\text{left}}^G(\hat{v})\}\!\} = \{\!\{swl_{k_2,c}^{(t)}(G', \hat{u}') \mid \hat{u}' \in \mathcal{N}_{k_2,c,\text{left}}^{G'}(\hat{v}')\}\!\}$
(4) $\{\!\{swl_{k_2,c}^{(t+1/2)}(G, \hat{u}) \mid \hat{u} \in \mathcal{N}_{k_2,c,\text{left}}^G(\hat{v})\}\!\} = \{\!\{swl_{k_2,c}^{(t+1/2)}(G', \hat{u}') \mid \hat{u}' \in \mathcal{N}_{k_2,c,\text{left}}^{G'}(\hat{v}')\}\!\}$

(1) + induction hypothesis $\Longrightarrow$ (a) $swl_{k_1,c}^{(t)}(G, \hat{v}) = swl_{k_1,c}^{(t)}(G', \hat{v}')$

(2) has two situations: $|\hat{v}| = |\hat{v}'| < k_1$ and $|\hat{v}| = |\hat{v}'| = k_1$. For the first situation, $\mathcal{N}_{k_2,c,\text{right}}^G(\hat{v}) = \mathcal{N}_{k_1,c,\text{right}}^G(\hat{v})$ and $\mathcal{N}_{k_2,c,\text{right}}^{G'}(\hat{v}') = \mathcal{N}_{k_1,c,\text{right}}^{G'}(\hat{v}')$, $\exists$ a bijective mapping $b$ between $\mathcal{N}_{k_2,c,\text{right}}^{G'}(\hat{v}')$ and $\mathcal{N}_{k_2,c,\text{right}}^G(\hat{v})$ such that $\forall \hat{u} \in \mathcal{N}_{k_2,c,\text{right}}^G(\hat{v})$, $swl_{k_2,c}^{(t)}(G, \hat{u}) = swl_{k_2,c}^{(t)}(G', b(\hat{u}))$ and by induction $swl_{k_1,c}^{(t)}(G, \hat{u}) = swl_{k_1,c}^{(t)}(G', b(\hat{u}))$, hence $swl_{k_1,c}^{(t+1/2)}(G, \hat{v}) =$

$swl_{k_1,c}^{(t+1/2)}(G',\hat{v}')$. For the second situation, $\mathcal{N}_{k_2,c,\text{right}}^{G'}(\hat{v}')$ elements while $\mathcal{N}_{k_1,c,\text{right}}^{G'}(\hat{v}') = \emptyset$, then clearly (b) $swl_{k_1,c}^{(t+1/2)}(G,\hat{v}) = swl_{k_1,c}^{(t+1/2)}(G',\hat{v}')$ (both are empty multisets).

(3) follows the same argument step in (2)'s first situation, with induction hypothesis we can get (c) $\{\!\{swl_{k_1,c}^{(t)}(G,\hat{u}) \mid \hat{u} \in \mathcal{N}_{k_1,c,\text{left}}^{G}(\hat{v})\}\!\} = \{\!\{swl_{k_1,c}^{(t)}(G',\hat{u}') \mid \hat{u}' \in \mathcal{N}_{k_1,c,\text{left}}^{G'}(\hat{v}')\}\!\}$

(4) implies that $\exists$ a mapping $b$ between $\mathcal{N}_{k_2,c,\text{left}}^{G'}(\hat{v}')$ and $\mathcal{N}_{k_2,c,\text{left}}^{G}(\hat{v})$ such that for any $\hat{u} \in \mathcal{N}_{k_2,c,\text{left}}^{G'}(\hat{v}')$, $swl_{k_2,c}^{(t+1/2)}(G,\hat{u}) = swl_{k_2,c}^{(t+1/2)}(G',b(\hat{u}'))$. Using the same argument in (2) and induction hypothesis, this implies that for any $\hat{u} \in \mathcal{N}_{k_1,c,\text{left}}^{G'}(\hat{v}')$, $swl_{k_1,c}^{(t+1/2)}(G,\hat{u})$ $= swl_{k_1,c}^{(t+1/2)}(G',b(\hat{u}'))$. Hence we get (d) $\{\!\{swl_{k_1,c}^{(t+1/2)}(G,\hat{u}) \mid \hat{u} \in \mathcal{N}_{k_1,c,\text{left}}^{G}(\hat{v})\}\!\} = \{\!\{swl_{k_1,c}^{(t+1/2)}(G',\hat{u}') \mid \hat{u}' \in \mathcal{N}_{k_1,c,\text{left}}^{G'}(\hat{v}')\}\!\}$.

Combining (a) (b) (c) (d) we get $swl_{k_1,c}^{(t+1)}(G,\hat{v}) = swl_{k_1,c}^{(t+1)}(G',\hat{v}')$. $\qquad\square$

With Lemma.3 we are ready to prove (2) in Theorem 3. When two graphs $G$ and $G'$ cannot be distinguished by $(k_2,c)(\leq)$-SETWL, we have $\{\!\{swl_{k_2,c}^{(t)}(G,\hat{v}) \mid \hat{v} \in V(S_{k_2,c\text{-swl}}(G))\}\!\} = \{\!\{swl_{k_2,c}^{(t)}(G',\hat{v}') \mid \hat{v}' \in V(S_{k_2,c\text{-swl}}(G'))\}\!\}$. When $\hat{v}$ and $\hat{u}$ have different $k$ (number of nodes) and $c$ (number of components of its induced subgraph), their color cannot be the same (as $t=0$ already be different). Hence $\{\!\{swl_{k_2,c}^{(t)}(G,\hat{v}) \mid \hat{v} \in V(S_{k_2,c\text{-swl}}(G))\}\!\} = \{\!\{swl_{k_2,c}^{(t)}(G',\hat{v}') \mid \hat{v}' \in V(S_{k_2,c\text{-swl}}(G'))\}\!\}$ is equivalent to $\forall k \leq k_2, cc \leq c$, $\{\!\{swl_{k_2,c}^{(t)}(G,\hat{v}) \mid \hat{v} \in V_{k,cc}(S_{k_2,c\text{-swl}}(G))\}\!\} = \{\!\{swl_{k_2,c}^{(t)}(G',\hat{v}') \mid \hat{v}' \in V_{k,cc}(S_{k_2,c\text{-swl}}(G'))\}\!\}$. With Lemma.3 we can get that $\forall k \leq k_1, cc \leq c$, $\{\!\{swl_{k_1,c}^{(t)}(G,\hat{v}) \mid \hat{v} \in V_{k,cc}(S_{k_1,c\text{-swl}}(G))\}\!\} = \{\!\{swl_{k_1,c}^{(t)}(G',\hat{v}') \mid \hat{v}' \in V_{k,cc}(S_{k_1,c\text{-swl}}(G'))\}\!\}$. Hence two graphs $G$ and $G'$ cannot be distinguished by $(k_1,c)(\leq)$-SETWL.

$\qquad\square$

### A.4.7  Proof of Theorem 6

**Theorem 6.** *When (i) BaseGNN can distinguish any non-isomorhpic graphs with at most $k$ nodes, (ii) all MLPs have sufficient depth and width, and (iii) POOL is an injective function, then for any $t \in \mathbb{N}$, $t$-layer $(k,c)(\leq)$-SETGNN is as expressive as $(k,c)(\leq)$-SETWL at the $t$-th iteration.*

*Proof.* For any $G,\hat{v}$, let $swl_{k,c}^{(t)}(G,\hat{v})$ denotes the color of $\hat{v}$ at $t$-iteration $(k,c)(\leq)$-SETWL and $h_{k,c}^{(t)}(G,\hat{v})$ denotes the embedding of $\hat{v}$ at $t$-th layer $(k,c)(\leq)$-SETGNN. We prove the above theorem by showing that $swl_{k,c}^{(t)}(G,\hat{v}) = swl_{k,c}^{(t)}(G',\hat{v}') \iff h_{k,c}^{(t)}(G,\hat{v}) = h_{k,c}^{(t)}(G',\hat{v}')$. We remove the subscript $k,c$ when possible without introducing confusion.

For easier reference, recall the updating formulation for $t$-iteration $(k,c)(\leq)$-SETWL is

$$swl^{(t+\frac{1}{2})}(G,\hat{v}) = \{\!\{swl_{k,c}^{(t)}(G,\hat{u}) \mid \hat{u} \in \mathcal{N}_{\text{right}}^{G}(\hat{v})\}\!\} \tag{26}$$

$$swl^{(t+1)}(G,\hat{v}) = \Big(swl^{(t)}(G,\hat{v}), swl^{(t+\frac{1}{2})}(G,\hat{v}), \tag{27}$$

$$\{\!\{swl^{(t)}(G,\hat{u}) \mid \hat{u} \in \mathcal{N}_{\text{left}}^{G}(\hat{v})\}\!\}, \{\!\{swl^{(t+\frac{1}{2})}(G,\hat{u}) \mid \hat{u} \in \mathcal{N}_{\text{left}}^{G}(\hat{v})\}\!\}\Big) \tag{28}$$

And the updating formulation for $(k,c)(\leq)$-SETGNN is

$$h^{(t+\frac{1}{2})}(\hat{v}) = \sum_{\hat{u} \in \mathcal{N}_{\text{right}}^{G}(\hat{v})} \text{MLP}^{(t+\frac{1}{2})}(h^{(t)}(\hat{u})) \tag{29}$$

$$h^{(t+1)}(\hat{v}) = \text{MLP}^{(t)}\Big(h^{(t)}(\hat{v}), h^{(t+\frac{1}{2})}(\hat{v}), \tag{30}$$

$$\sum_{\hat{u} \in \mathcal{N}_{\text{left}}^{G}(\hat{v})} \text{MLP}_A^{(t)}(h^{(t)}(\hat{u})), \sum_{\hat{u} \in \mathcal{N}_{\text{left}}^{G}(\hat{v})} \text{MLP}_B^{(t)}(h^{(t+\frac{1}{2})}(\hat{u}))\Big) \tag{31}$$

At $t = 0$, given powerful enough BaseGNN with condition (i) in the theorem, $h^{(0)}(G, \hat{v}) = h^{(0)}(G', \hat{v}') \iff G[\hat{v}]$ and $G'[\hat{v}']$ are isomorphic $\iff \boldsymbol{swl}^{(0)}(G, \hat{v}) = \boldsymbol{swl}^{(0)}(G', \hat{v}')$.

Now assume for $\leq t$ iterations the claim $\boldsymbol{swl}^{(t)}(G, \hat{v}) = \boldsymbol{swl}^{(t)}(G', \hat{v}') \iff h^{(t)}(G, \hat{v}) = h^{(t)}(G', \hat{v}')$ holds (for any $\hat{v}$ and $\hat{v}'$). We prove it holds for $t + 1$ iteration. We first prove the forward direction. $\boldsymbol{swl}^{(t+1)}(G, \hat{v}) = \boldsymbol{swl}^{(t+1)}(G', \hat{v}')$ imples that

(1) $\boldsymbol{swl}^{(t)}(G, \hat{v}) = \boldsymbol{swl}^{(t)}(G', \hat{v}')$
(2) $\boldsymbol{swl}^{(t+1/2)}(G, \hat{v}) = \boldsymbol{swl}^{(t+1/2)}(G', \hat{v}')$
(3) $\{\!\{ \boldsymbol{swl}^{(t)}(G, \hat{u}) \mid \hat{u} \in \mathcal{N}_{\text{left}}^{G}(\hat{v}) \}\!\} = \{\!\{ \boldsymbol{swl}^{(t)}(G', \hat{u}') \mid \hat{u}' \in \mathcal{N}_{\text{left}}^{G'}(\hat{v}') \}\!\}$
(4) $\{\!\{ \boldsymbol{swl}^{(t+1/2)}(G, \hat{u}) \mid \hat{u} \in \mathcal{N}_{\text{left}}^{G}(\hat{v}) \}\!\} = \{\!\{ \boldsymbol{swl}^{(t+1/2)}(G', \hat{u}') \mid \hat{u}' \in \mathcal{N}_{\text{left}}^{G'}(\hat{v}') \}\!\}$

(1) and (3) can be directly transformed to relationship between $h$ using the inductive hypothesis. Formally, we have
(a) $h^{(t)}(G, \hat{v}) = h^{(t)}(G', \hat{v}')$ and
(c) $\{\!\{ h^{(t)}(G, \hat{u}) \mid \hat{u} \in \mathcal{N}_{\text{left}}^{G}(\hat{v}) \}\!\} = \{\!\{ h^{(t)}(G', \hat{u}') \mid \hat{u}' \in \mathcal{N}_{\text{left}}^{G'}(\hat{v}') \}\!\}$

(2) and (4) need one additional process. Notice that (2) is equivalent to $\{\!\{ \boldsymbol{swl}^{(t)}(G, \hat{u}) \mid \hat{u} \in \mathcal{N}_{\text{right}}^{G}(\hat{v}) \}\!\} = \{\!\{ \boldsymbol{swl}^{(t)}(G', \hat{u}') \mid \hat{u} \in \mathcal{N}_{\text{right}}^{G'}(\hat{v}') \}\!\}$ and by inductive hypothesis we know $\{\!\{ h^{(t)}(G, \hat{u}) \mid \hat{u} \in \mathcal{N}_{\text{right}}^{G}(\hat{v}) \}\!\} = \{\!\{ h^{(t)}(G', \hat{u}') \mid \hat{u} \in \mathcal{N}_{\text{right}}^{G'}(\hat{v}') \}\!\}$. As the formulation Eq.(29) applies a MLP with summation, which is permutation invariant to ordering, to the multiset $\{\!\{ h^{(t)}(G, \hat{u}) \mid \hat{u} \in \mathcal{N}_{\text{right}}^{G}(\hat{v}) \}\!\}$, the same multiset leads to the same output. Hence we know
(b) $h^{(t+1/2)}(G, \hat{v}) = h^{(t+1/2)}(G', \hat{v}')$.

For (4) we know that there is a bijective mapping $g$ between $\mathcal{N}_{\text{left}}^{G}(\hat{v})$ and $\mathcal{N}_{\text{left}}^{G}(\hat{v}')$ such that $\boldsymbol{swl}^{(t+1/2)}(G, \hat{u}) = \boldsymbol{swl}^{(t+1/2)}(G', g(\hat{u}))$. Then using the same argument as (2) =>(b) we can get $h^{(t+1/2)}(G, \hat{u}) = h^{(t+1/2)}(G', g(\hat{u}))$ for any $\hat{u} \in \mathcal{N}_{\text{left}}^{G}(\hat{v})$, which is equivalent to
(d) $\{\!\{ h^{(t+1/2)}(G, \hat{u}) \mid \hat{u} \in \mathcal{N}_{\text{left}}^{G}(\hat{v}) \}\!\} = \{\!\{ h^{(t+1/2)}(G', \hat{u}') \mid \hat{u}' \in \mathcal{N}_{\text{left}}^{G'}(\hat{v}') \}\!\}$.

Combining (a)(b)(c)(d) and using the permutation invariant property of MLP with summation, we can derive that $h^{(t+1)}(G, \hat{v}) = h^{(t+1)}(G', \hat{v}')$.

Now for the backward direction. We first characterize the property of MLP with summation from DeepSet [64] and GIN [61]'s Lemma 5.

**Lemma 4** (Lemma 5 of [61]). *Assume $\mathcal{X}$ is countable. There exists a function $f : \mathcal{X} \to \mathbb{R}^n$ so that $h(X) = \sum_{x \in X} f(x)$ is unique for each multiset $X \in \mathcal{X}$ of bounded size.*

Using this Lemma, we know that given enough depth and width of a MLP, there exist a MLP that $\sum_{x \in X} \text{MLP}(x) = \sum_{y \in Y} \text{MLP}(y) \iff X = Y$ or in other words two multisets $X$ and $Y$ are equivalent. Now by $h^{(t+1)}(G, \hat{v}) = h^{(t+1)}(G', \hat{v}')$ and using the Eq.(31), we know there exists MLPs inside Eq.(29) and Eq.(31), such that

(1) $h^{(t)}(G, \hat{v}) = h^{(t)}(G', \hat{v}')$
(2) $h^{(t+1/2)}(G, \hat{v}) = h^{(t+1/2)}(G', \hat{v}')$
(3) $\{\!\{ h^{(t)}(G, \hat{u}) \mid \hat{u} \in \mathcal{N}_{\text{left}}^{G}(\hat{v}) \}\!\} = \{\!\{ h^{(t)}(G', \hat{u}') \mid \hat{u}' \in \mathcal{N}_{\text{left}}^{G'}(\hat{v}') \}\!\}$
(4) $\{\!\{ h^{(t+1/2)}(G, \hat{u}) \mid \hat{u} \in \mathcal{N}_{\text{left}}^{G}(\hat{v}) \}\!\} = \{\!\{ h^{(t+1/2)}(G', \hat{u}') \mid \hat{u}' \in \mathcal{N}_{\text{left}}^{G'}(\hat{v}') \}\!\}$

Where (3) and (4) are derived using the provided lemma. Hence following the same argument as the forward process, we can prove that $\boldsymbol{swl}^{(t+1)}(G, \hat{v}) = \boldsymbol{swl}^{(t+1)}(G', \hat{v}')$.

Combining the forward and backward direction, we have $\boldsymbol{swl}^{(t+1)}(G, \hat{v}) = \boldsymbol{swl}^{(t+1)}(G', \hat{v}') \iff h^{(t+1)}(G, \hat{v}) = h^{(t+1)}(G', \hat{v}')$. Hence by induction we proved for any step $t$ and any $\hat{v}, \hat{v}'$, the above statement is true. This shows that $(k, c)(\leq)$-SETGNN and $(k, c)(\leq)$-SETWL have same expressivity.

$\square$

### A.4.8 Proof of Theorem 7

**Theorem 7.** *For any $t \in \mathbb{N}$, the $t$-layer $(k, c)(\leq)$-SETGNN$^*$ is more expressive than the $t$-layer $(k, c)(\leq)$-SETGNN. As $\lim_{t \to \infty}$, $(k, c)(\leq)$-SETGNN is as expressive as $(k, c)(\leq)$-SETGNN$^*$.*

*Proof.* We first prove that $t$-layer $(k, c)(\leq)$-SETGNN$^*$ is more expressive than $t$-layer $(k, c)(\leq)$-SETGNN, by showing that if a $m$-set $\hat{\boldsymbol{u}}$ has the same representation to another $m$-set $\hat{\boldsymbol{v}}$ in $t$-layer $(k, c)(\leq)$-SETGNN$^*$, then they also have the same representation in $t$-layer $(k, c)(\leq)$-SETGNN. Let $h$ be the representation inside $(k, c)(\leq)$-SETGNN$^*$, and $g$ be the representation inside $(k, c)(\leq)$-SETGNN.

To simplify the proof, we first simplify the formulations of $(k, c)(\leq)$-SETGNN$^*$ Eq.13 and Eq.14 by removing unnecessary super and under script of MLP without introducing ambiguity. We then add another superscript to embeddings to indicate which step inside the one-layer bidirectional propagation. Notice that for one-layer bidirectional propagation, there are $2k - 2$ intermediate sequential steps. The proof assumes that all MLPs are injective. We rewrite Eq.13 and Eq.14 as follows

$$\forall m\text{-set } \hat{\boldsymbol{v}}, h^{(t+\frac{1}{2})}(\hat{\boldsymbol{v}}) := h^{(t+\frac{1}{2}, k-m)}(\hat{\boldsymbol{v}}) = \text{MLP}\Big(h^{(t)}(\hat{\boldsymbol{v}}), \sum_{\hat{\boldsymbol{w}} \in \mathcal{N}_{\text{right}}^G(\hat{\boldsymbol{v}})} \text{MLP}(h^{(t+\frac{1}{2}, k-m-1)}(\hat{\boldsymbol{w}}))\Big) \tag{32}$$

$$\forall m\text{-set } \hat{\boldsymbol{v}}, h^{(t+1)}(\hat{\boldsymbol{v}}) := h^{(t+1, m-1)}(\hat{\boldsymbol{v}}) = \text{MLP}\Big(h^{(t+\frac{1}{2}, k-m)}(\hat{\boldsymbol{v}}), \sum_{\hat{\boldsymbol{w}} \in \mathcal{N}_{\text{left}}^G(\hat{\boldsymbol{v}})} \text{MLP}(h^{(t+1, m-2)}(\hat{\boldsymbol{w}}))\Big) \tag{33}$$

Where we have boundary case with $h^{(t+\frac{1}{2}, 0)}(\hat{\boldsymbol{u}}) = h^{(t)}(\hat{\boldsymbol{u}})$ and $h^{(t+1, 0)}(\hat{\boldsymbol{u}}) = h^{(t+\frac{1}{2})}(\hat{\boldsymbol{u}})$. $h^{(t+\frac{1}{2})}(\hat{\boldsymbol{v}}) := h^{(t+\frac{1}{2}, k-m)}(\hat{\boldsymbol{v}})$ represents that the representation is calculated at $k - m$ step for $t + \frac{1}{2}$ layer.

We prove the theorem by induction on $t$. Specifically, we want to prove that for any $t$, $\hat{\boldsymbol{v}}$, $\hat{\boldsymbol{u}}$, $h^{(t)}(\hat{\boldsymbol{u}}) = h^{(t)}(\hat{\boldsymbol{v}}) \implies g^{(t)}(\hat{\boldsymbol{u}}) = g^{(t)}(\hat{\boldsymbol{v}})$ and $h^{(t-\frac{1}{2})}(\hat{\boldsymbol{u}}) = h^{(t-\frac{1}{2})}(\hat{\boldsymbol{v}}) \implies g^{(t-\frac{1}{2})}(\hat{\boldsymbol{u}}) = g^{(t-\frac{1}{2})}(\hat{\boldsymbol{v}})$. The base case is easy to verify as the definition of initialization step is the same.

Now assume that for $t \leq l$ we have for any $t$, $\hat{\boldsymbol{v}}$, $\hat{\boldsymbol{u}}$, $h^{(t)}(\hat{\boldsymbol{u}}) = h^{(t)}(\hat{\boldsymbol{v}}) \implies g^{(t)}(\hat{\boldsymbol{u}}) = g^{(t)}(\hat{\boldsymbol{v}})$ and $h^{(t-\frac{1}{2})}(\hat{\boldsymbol{u}}) = h^{(t-\frac{1}{2})}(\hat{\boldsymbol{v}}) \implies g^{(t-\frac{1}{2})}(\hat{\boldsymbol{u}}) = g^{(t-\frac{1}{2})}(\hat{\boldsymbol{v}})$. We first prove that for $t = l + 1$, $h^{(l+\frac{1}{2})}(\hat{\boldsymbol{u}}) = h^{(l+\frac{1}{2})}(\hat{\boldsymbol{v}}) \implies g^{(l+\frac{1}{2})}(\hat{\boldsymbol{u}}) = g^{(l+\frac{1}{2})}(\hat{\boldsymbol{v}})$.

First, Eq.32 can be rewrite as

$$h^{(t+\frac{1}{2})}(\hat{\boldsymbol{v}}) : h^{(t+\frac{1}{2}, k-m)}(\hat{\boldsymbol{v}}) = \text{MLP}\Big(h^{(t)}(\hat{\boldsymbol{v}}), \sum_{\hat{\boldsymbol{w}} \in \mathcal{N}_{\text{right}}^G(\hat{\boldsymbol{v}})} \text{MLP}(h^{(t+\frac{1}{2}, k-m-1)}(\hat{\boldsymbol{w}}))\Big) \tag{34}$$

$$= \text{MLP}\Big(h^{(t)}(\hat{\boldsymbol{v}}), \sum_{\hat{\boldsymbol{w}} \in \mathcal{N}_{\text{right}}^G(\hat{\boldsymbol{v}})} \text{MLP}(\text{MLP}(h^{(t)}(\hat{\boldsymbol{w}})), \sum_{\hat{\boldsymbol{p}} \in \mathcal{N}_{\text{right}}^G(\hat{\boldsymbol{w}})} \text{MLP}(h^{(t+\frac{1}{2}, k-m-2)}(\hat{\boldsymbol{p}})))\Big) \tag{35}$$

Then $h^{(l+\frac{1}{2})}(\hat{\boldsymbol{v}}) = h^{(l+\frac{1}{2})}(\hat{\boldsymbol{u}}) \implies \sum_{\hat{\boldsymbol{w}} \in \mathcal{N}_{\text{right}}^G(\hat{\boldsymbol{v}})} \text{MLP}(h^{(l)}(\hat{\boldsymbol{w}})) = \sum_{\hat{\boldsymbol{w}} \in \mathcal{N}_{\text{right}}^G(\hat{\boldsymbol{u}})} \text{MLP}(h^{(l)}(\hat{\boldsymbol{w}}))$. Hence we can find a bijective mapping $f$ between $\mathcal{N}_{\text{right}}^G(\hat{\boldsymbol{v}})$ and $\mathcal{N}_{\text{right}}^G(\hat{\boldsymbol{u}})$ such that $\text{MLP}(h^{(l)}(\hat{\boldsymbol{w}})) = \text{MLP}(h^{(l)}(f(\hat{\boldsymbol{w}})))$. Then by inductive hyphothesis, $\text{MLP}(g^{(l)}(\hat{\boldsymbol{w}})) = \text{MLP}(g^{(l)}(f(\hat{\boldsymbol{w}})))$ for all $\hat{\boldsymbol{w}} \in \mathcal{N}_{\text{right}}^G(\hat{\boldsymbol{v}})$. This implies that $\sum_{\hat{\boldsymbol{w}} \in \mathcal{N}_{\text{right}}^G(\hat{\boldsymbol{v}})} \text{MLP}(g^{(l)}(\hat{\boldsymbol{w}})) = \sum_{\hat{\boldsymbol{w}} \in \mathcal{N}_{\text{right}}^G(\hat{\boldsymbol{u}})} \text{MLP}(g^{(l)}(\hat{\boldsymbol{w}}))$ or equivalently $g^{(l+\frac{1}{2})}(\hat{\boldsymbol{v}}) = g^{(l+\frac{1}{2})}(\hat{\boldsymbol{u}})$.

Now we can assume that for $t \leq m$ we have for any $t$, $\hat{\boldsymbol{v}}$, $\hat{\boldsymbol{u}}$, $h^{(t)}(\hat{\boldsymbol{u}}) = h^{(t)}(\hat{\boldsymbol{v}}) \implies g^{(t)}(\hat{\boldsymbol{u}}) = g^{(t)}(\hat{\boldsymbol{v}})$ and for $t \leq l + 1$ $h^{(t-\frac{1}{2})}(\hat{\boldsymbol{u}}) = h^{(t-\frac{1}{2})}(\hat{\boldsymbol{v}}) \implies g^{(t-\frac{1}{2})}(\hat{\boldsymbol{u}}) = g^{(t-\frac{1}{2})}(\hat{\boldsymbol{v}})$. We prove that for $t = l + 1$, $h^{(l+1)}(\hat{\boldsymbol{u}}) = h^{(l+1)}(\hat{\boldsymbol{v}}) \implies g^{(l+1)}(\hat{\boldsymbol{u}}) = g^{(l+1)}(\hat{\boldsymbol{v}})$.

We first rewrite Eq.33 as

$$h^{(t+1)}(\hat{\boldsymbol{v}}) := h^{(t+1,m-1)}(\hat{\boldsymbol{v}}) = \mathrm{MLP}\Big(h^{(t+\frac{1}{2},k-m)}(\hat{\boldsymbol{v}}), \sum_{\hat{\boldsymbol{w}}\in\mathcal{N}^G_{\mathrm{left}}(\hat{\boldsymbol{v}})} \mathrm{MLP}(h^{(t+1,m-2)}(\hat{\boldsymbol{w}}))\Big) \qquad (36)$$

$$= \mathrm{MLP}\Big(h^{(t+\frac{1}{2},k-m)}(\hat{\boldsymbol{v}}), \sum_{\hat{\boldsymbol{w}}\in\mathcal{N}^G_{\mathrm{left}}(\hat{\boldsymbol{v}})} \mathrm{MLP}\Big(\mathrm{MLP}\big(h^{(t+\frac{1}{2},k-m+1)}(\hat{\boldsymbol{w}}), \sum_{\hat{\boldsymbol{p}}\in\mathcal{N}^G_{\mathrm{left}}(\hat{\boldsymbol{w}})} \mathrm{MLP}(h^{(t+1,m-3)}(\hat{\boldsymbol{p}}))\big)\Big)\Big)$$
$$(37)$$

Then $h^{(l+1)}(\hat{\boldsymbol{u}}) = h^{(l+1)}(\hat{\boldsymbol{v}})$ implies

1) $h^{(t+\frac{1}{2},k-m)}(\hat{\boldsymbol{v}}) = h^{(t+\frac{1}{2},k-m)}(\hat{\boldsymbol{u}})$ or equivalently $h^{(t+\frac{1}{2})}(\hat{\boldsymbol{v}}) = h^{(t+\frac{1}{2})}(\hat{\boldsymbol{u}})$
2) $\{\!\{h^{(t+\frac{1}{2})}(\hat{\boldsymbol{w}}) \mid \hat{\boldsymbol{w}} \in \mathcal{N}^G_{\mathrm{left}}(\hat{\boldsymbol{v}})\}\!\} = \{\!\{h^{(t+\frac{1}{2})}(\hat{\boldsymbol{w}}) \mid \hat{\boldsymbol{w}} \in \mathcal{N}^G_{\mathrm{left}}(\hat{\boldsymbol{u}})\}\!\}$

Also by Eq.32 we know that $h^{(t+\frac{1}{2})}(\hat{\boldsymbol{v}}) = h^{(t+\frac{1}{2})}(\hat{\boldsymbol{u}})$ implies

3) $h^{(t)}(\hat{\boldsymbol{v}}) = h^{(t)}(\hat{\boldsymbol{u}})$.

Combining with 2) and 3) we know

4) $\{\!\{h^{(t)}(\hat{\boldsymbol{w}}) \mid \hat{\boldsymbol{w}} \in \mathcal{N}^G_{\mathrm{left}}(\hat{\boldsymbol{v}})\}\!\} = \{\!\{h^{(t)}(\hat{\boldsymbol{w}}) \mid \hat{\boldsymbol{w}} \in \mathcal{N}^G_{\mathrm{left}}(\hat{\boldsymbol{u}})\}\!\}$

Now using our inductive hypothesis, we can transform 1) 2) 3) 4) by replace $h$ with $g$. Then based on the equation in 11, we know $g^{(l+1)}(\hat{\boldsymbol{u}}) = g^{(l+1)}(\hat{\boldsymbol{v}})$.

Combining the two proved inductive hypothesis and applying them alternately, we know that for any $t, \hat{\boldsymbol{v}}, \hat{\boldsymbol{u}}$, $h^{(t)}(\hat{\boldsymbol{u}}) = h^{(t)}(\hat{\boldsymbol{v}}) \implies g^{(t)}(\hat{\boldsymbol{u}}) = g^{(t)}(\hat{\boldsymbol{v}})$ and $h^{(t-\frac{1}{2})}(\hat{\boldsymbol{u}}) = h^{(t-\frac{1}{2})}(\hat{\boldsymbol{v}}) \implies g^{(t-\frac{1}{2})}(\hat{\boldsymbol{u}}) = g^{(t-\frac{1}{2})}(\hat{\boldsymbol{v}})$. Hence $t$-layer $(k,c)(\leq)$-SETGNN* is more expressive than $t$-layer $(k,c)(\leq)$-SETGNN.

Now let's considering $t$ to infinity. Then all representations will become stable with $h^{(t)}(\hat{\boldsymbol{v}}) = h^{(t)}(\hat{\boldsymbol{u}}) \iff h^{(t+1)}(\hat{\boldsymbol{v}}) = h^{(t+1)}(\hat{\boldsymbol{u}})$ and $h^{(t-1/2)}(\hat{\boldsymbol{v}}) = h^{(t-1/2)}(\hat{\boldsymbol{u}}) \iff h^{(t+1/2)}(\hat{\boldsymbol{v}}) = h^{(t+1/2)}(\hat{\boldsymbol{u}})$. Notice that for a single set $\hat{\boldsymbol{v}}$, the information used from its neighbors are the same in both $(k,c)(\leq)$-SETGNN and $(k,c)(\leq)$-SETGNN*. Hence the equilibrium equations for set $\hat{\boldsymbol{v}}$ should be the same in both $(k,c)(\leq)$-SETGNN and $(k,c)(\leq)$-SETGNN*. Then they will have the same stable representations at the end.

$\square$

### A.4.9 Proof of Theorem 8

**Theorem 8.** *Let $G$ be a graph with $c$ connected components $C_1, ..., C_c$, and $G'$ be a graph also with $c$ connected components $C'_1, ..., C'_c$, then $G$ and $G'$ are isomorphic if and only if $\exists p : [c] \to [c]$, s.t. $\forall i \in [c]$, $C_i$ and $C'_{p(i)}$ are isomorphic.*

*Proof.* **Right $\implies$ Left**. Let $h_i : V(C_i) \to V(C'_{p(i)})$ be one isomorphism from $C_i$ to $C'_{p(i)}$ for $i \in [c]$. Then we can create a new mapping $h : V(G) \to V(G')$, such that for any $x \in V(C)$, it first locates which component $x$ inside, for example $i$, then apply $h_i$ to $x$. Clearly $h$ is a isomorphism between $G$ and $G'$, hence $G$ and $G'$ are isomorphic.

**Left $\implies$ Right**. $G$ and $G'$ are isomorphic, then there exists an isomorphism $h$ from $G$ to $G'$. Now for a specific component $C_i$ in $G$, $h$ maps $V(C_i)$ to a nodes set inside $V(G')$, and name it as $B_i$. For $i \neq j$, $B_i \cap B_j = \emptyset$ and there is not edges $e \in E(G')$ between $B_i$ and $B_j$ otherwise there must be an corresponding edge (apply $h^{-1}$) between $V(C_i)$ and $V(C_j)$ which is impossible. Hence $\{G'[B_i]\}_{i=1}^c$ are disconnected subgraphs. As $h$ also preserves connections, for any $i$, $C_i$ and $G'[B_i]$ are isomorphic. Hence we proved the right side.

$\square$

### A.4.10 Conjecture that k-MWL is equvialent to k-WL

**Proof of** $\forall t, \boldsymbol{wl}^{(t)}_k(G, \tilde{\boldsymbol{v}}) = \boldsymbol{wl}^{(t)}_k(G', \tilde{\boldsymbol{v}}') \impliedby \boldsymbol{mwl}^{(t)}_k(G, \tilde{\boldsymbol{v}}) = \boldsymbol{mwl}^{(t)}_k(G', \tilde{\boldsymbol{v}}')$

*Proof.* We first take a closer look at the condition of $\boldsymbol{mwl}_k^{(t+1)}(G,\tilde{\boldsymbol{v}}) = \boldsymbol{mwl}_k^{(t+1)}(G',\tilde{\boldsymbol{v}}')$. By Eq.(3) we know this is equivalent to

(1) $\boldsymbol{mwl}_k^{(t)}(G,\tilde{\boldsymbol{v}}) = \boldsymbol{mwl}_k^{(t)}(G',\tilde{\boldsymbol{v}}')$ and (2) $\left\{\!\!\left\{ \{\boldsymbol{mwl}_k^{(t)}(G,\tilde{\boldsymbol{v}}[x/o_G^{-1}(\tilde{\boldsymbol{v}},i)])| x \in V(G)\} \mid i = 1,...,k \right\}\!\!\right\} = \left\{\!\!\left\{ \{\boldsymbol{mwl}_k^{(t)}(G',\tilde{\boldsymbol{v}}'[x/o_G^{-1}(\tilde{\boldsymbol{v}}',i)])| x \in V(G')\} \mid i = 1,...,k \right\}\!\!\right\}$. (2) can be rewrite as $\left\{\!\!\left\{ \{\boldsymbol{mwl}_k^{(t)}(G,\tilde{\boldsymbol{v}}[x/\mathrm{idx}_{\tilde{\boldsymbol{v}}}(y)])| x \in V(G)\} \mid y \in \tilde{\boldsymbol{v}} \right\}\!\!\right\} = \left\{\!\!\left\{ \{\boldsymbol{mwl}_k^{(t)}(G',\tilde{\boldsymbol{v}}'[x/\mathrm{idx}_{\tilde{\boldsymbol{v}}'}(y)])| x \in V(G')\} \mid y \in \tilde{\boldsymbol{v}}' \right\}\!\!\right\}$, which is equivalent to (3) $\exists$ bijective $f : V(G) \to V(G')$, $\forall y \in \tilde{\boldsymbol{v}}$, $\{\boldsymbol{mwl}_k^{(t)}(G,\tilde{\boldsymbol{v}}[x/\mathrm{idx}_{\tilde{\boldsymbol{v}}}(y)])| x \in V(G)\} = \{\boldsymbol{mwl}_k^{(t)}(G',\tilde{\boldsymbol{v}}'[x/\mathrm{idx}_{\tilde{\boldsymbol{v}}'}(f(y))])| x \in V(G')\}$.

Define $F^{(t+1)}(G,G',\tilde{\boldsymbol{v}},\tilde{\boldsymbol{v}}') := \{$ injective $f : \tilde{\boldsymbol{v}} \to \tilde{\boldsymbol{v}}' \mid f \in F^{(t)}(G,G',\tilde{\boldsymbol{v}},\tilde{\boldsymbol{v}}')$, AND , $\forall y \in \tilde{\boldsymbol{v}}, \exists h_y : V(G) \to V(G'), \forall x, f \in F^{(t)}(G,G',\tilde{\boldsymbol{v}}[x/\mathrm{idx}_{\tilde{\boldsymbol{v}}}(y)], \tilde{\boldsymbol{v}}'[h_y(x)/\mathrm{idx}_{\tilde{\boldsymbol{v}}'}(f(y))])\}$. Let $F^{(0)}(G,G',\tilde{\boldsymbol{v}},\tilde{\boldsymbol{v}}') := \{f \mid f$ is an isomorphism from $G[\tilde{\boldsymbol{v}}]$ to $G'[\tilde{\boldsymbol{v}}']\}$.

**Lemma 5.** $\forall t, \boldsymbol{mwl}_k^{(t)}(G,\tilde{\boldsymbol{v}}) = \boldsymbol{mwl}_k^{(t)}(G',\tilde{\boldsymbol{v}}')$ if and only if $F^{(t)}(G,G',\tilde{\boldsymbol{v}},\tilde{\boldsymbol{v}}') \neq \emptyset$.

*Notice that this Lemma is conjectured to be true. This needs additional proof.*

**Lemma 6.** $\forall t, \boldsymbol{mwl}_k^{(t)}(G,\tilde{\boldsymbol{v}}) = \boldsymbol{mwl}_k^{(t)}(G',\tilde{\boldsymbol{v}}') \implies \forall f \in F^{(t)}(G,G',\tilde{\boldsymbol{v}},\tilde{\boldsymbol{v}}'), \boldsymbol{wl}_k^{(t)}(G,\overrightarrow{\boldsymbol{v}}) = \boldsymbol{wl}_k^{(t)}(G,f(\overrightarrow{\boldsymbol{v}}))$

*Proof of Lemma.6:* We prove it by induction on $t$. When $t = 0$, $F^{(0)}(G,G',\tilde{\boldsymbol{v}},\tilde{\boldsymbol{v}}')$ contains all isomorphisms between $G[\tilde{\boldsymbol{v}}]$ and $G'[\tilde{\boldsymbol{v}}']$, hence the right side is correct. Assume the statement is correct for $\leq t$. For $t + 1$ case, the left side implies (1) $\boldsymbol{mwl}_k^{(t)}(G,\tilde{\boldsymbol{v}}) = \boldsymbol{mwl}_k^{(t)}(G',\tilde{\boldsymbol{v}}')$ and (2)$F^{(t+1)}(G,G',\tilde{\boldsymbol{v}},\tilde{\boldsymbol{v}}') \neq \emptyset, \forall f \in F^{(t+1)}(G,G',\tilde{\boldsymbol{v}},\tilde{\boldsymbol{v}}')$, $\forall y \in \tilde{\boldsymbol{v}}, \exists h_y : V(G) \to V(G'), \forall x \in V(G), \boldsymbol{mwl}_k^{(t)}(G,\tilde{\boldsymbol{v}}[x/\mathrm{idx}_{\tilde{\boldsymbol{v}}}(y)]) = \boldsymbol{mwl}_k^{(t)}(G',\tilde{\boldsymbol{v}}'[h_y(x)/\mathrm{idx}_{\tilde{\boldsymbol{v}}'}(f(y))])$. By induction hypothesis, (1) and $F^{(t+1)}(G,G',\tilde{\boldsymbol{v}},\tilde{\boldsymbol{v}}') \subseteq F^{(t)}(G,G',\tilde{\boldsymbol{v}},\tilde{\boldsymbol{v}}')$ imply (a) $\forall f \in F^{(t+1)}(G,G',\tilde{\boldsymbol{v}},\tilde{\boldsymbol{v}}'), \boldsymbol{wl}_k^{(t)}(G,\overrightarrow{\boldsymbol{v}}) = \boldsymbol{wl}_k^{(t)}(G',f(\overrightarrow{\boldsymbol{v}}))$; (2) and $\forall y \in \tilde{\boldsymbol{v}}, f \in F^{(t)}(G,G',\tilde{\boldsymbol{v}}[x/\mathrm{idx}_{\tilde{\boldsymbol{v}}}(y)], \tilde{\boldsymbol{v}}'[h_y(x)/\mathrm{idx}_{\tilde{\boldsymbol{v}}'}(f(y))])$ imply (b) $\exists h, \forall x \in V(G), \boldsymbol{wl}_k^{(t)}(G,\overrightarrow{\boldsymbol{v}}[x/\mathrm{idx}_{\overrightarrow{\boldsymbol{v}}}(y)]) = \boldsymbol{wl}_k^{(t)}(G',f(\overrightarrow{\boldsymbol{v}})[h(x)/\mathrm{idx}_{f(\overrightarrow{\boldsymbol{v}})}(f(y))])$, which can be rewritten as $\forall y \in \tilde{\boldsymbol{v}}, \{\!\{\boldsymbol{wl}_k^{(t)}(G,\overrightarrow{\boldsymbol{v}}[x/\mathrm{idx}_{\overrightarrow{\boldsymbol{v}}}(y)] \mid x \in V(G))\}\!\} = \{\!\{\boldsymbol{wl}_k^{(t)}(G',f(\overrightarrow{\boldsymbol{v}})[x/\mathrm{idx}_{f(\overrightarrow{\boldsymbol{v}})}(f(y))] \mid x \in V(G))\}\!\}$. Now applying Eq.(1) with (a) and (b), we can derive that $\forall f \in F^{(t+1)}(G,G',\tilde{\boldsymbol{v}},\tilde{\boldsymbol{v}}')$, $\boldsymbol{wl}_k^{(t+1)}(G,\overrightarrow{\boldsymbol{v}}) = \boldsymbol{wl}_k^{(t+1)}(G,f(\overrightarrow{\boldsymbol{v}}))$. Hence the statement is correct for any $t$.

Using Lemma.6 and the conclusion ($\forall p \in \mathrm{perm}[k], \boldsymbol{wl}_k^{(t+1)}(G,p(\overrightarrow{\boldsymbol{v}})) = \boldsymbol{wl}_k^{(t+1)}(G',p(f(\overrightarrow{\boldsymbol{v}})))$) proved in the first part of the proof of Theorem.1, we proved $\forall t, \boldsymbol{mwl}_k^{(t)}(G,\tilde{\boldsymbol{v}}) = \boldsymbol{mwl}_k^{(t)}(G',\tilde{\boldsymbol{v}}') \implies \boldsymbol{wl}_k^{(t)}(G,\tilde{\boldsymbol{v}}) = \boldsymbol{wl}_k^{(t)}(G',\tilde{\boldsymbol{v}}')$. $\qquad\square$

## A.5 Algorithm of extracting $(k,c)(\leq)$ sets and constructing the super-graph of $(k,c)(\leq)$-SETWL

Here we present the algorithm of constructing supernodes $(t+1,c)(\leq)$ sets from supernodes $(t,c)(\leq)$ sets, and constructing the bipartite graph between $(t,c)(\leq)$ sets and $(t+1,c)(\leq)$ sets. Notice the algorithm we presented is just the pseudo code and we use for-loops to help presentation. The algorithm can easily be parallelized (removing for-loops) with using matrix operations.

To get the full supergraph $S_{k,c\text{-swl}}$, we need to apply Algorithm.1 $k-1$ times sequentially.

## A.6 Algorithm of connecting supernodes to their connected components

In Sec.4.3 we proved that for a supernode with multi-component induced subgraph (call it multi-component supernode), the color initialization can be greatly sped up given knowing its each component's corresponding supernode. Hence we need to build the bipartite graph (call it components

---

**Algorithm 1** Constructing supernodes $(t+1, c)(\leq)$ sets and the bipartite graph between $t$ and $t+1$

---

**Input:** Input graph $G$, the list $O_{t, \leq c}$ containing all supernodes with $t$ nodes and $\leq c$ number of components, the list $N_{t, \leq c}$ with $\bar{N}_{t, \leq c}[i]$ be the number of components of $G[O_{t, \leq c}[i]]$.
**Output:** $O_{t+1, \leq c}, N_{t+1, \leq c}, B_t$ containing edges between $O_{t, \leq c}$ and $O_{t+1, \leq c}$
    $O_{t+1, \leq c}, N_{t+1, \leq c}, B_t \leftarrow [], [], []$
    **for all** $\hat{s}, n$ in zip( $O_{t, \leq c}, N_{t, \leq c}$ ) **do**
        $\mathcal{N}_1 \leftarrow$ 1-st hop neighbors of $G[\hat{s}]$
        $\mathcal{N}_{>1} \leftarrow (V(G) \setminus \hat{s}) \setminus \mathcal{N}_1$
        **for all** $m$ in $\mathcal{N}_1$ **do**                            ▷ Number of components doesn't change.
            $\hat{q} \leftarrow \hat{s} + \{m\}$
            Append $\hat{q}$ into $O_{t+1, \leq c}$, $n$ into $N_{t+1, \leq c}$
            Append edge $(\hat{s}, \hat{q})$ into $B_t$          ▷ We change the edge to $(\hat{s}, \hat{q}, m)$ for Algorithm.2
        **end for**
        **if** n < c **then**                              ▷ Creating an additional component.
            **for all** $m$ in $\mathcal{N}_{>1}$ **do**
                $\hat{q} \leftarrow \hat{s} + \{m\}$
                Append $\hat{q}$ into $O_{t+1, \leq c}$, $n + 1$ into $N_{t+1, \leq c}$
                Append edge $(\hat{s}, \hat{q})$ into $B_t$          ▷ We change the edge to $(\hat{s}, \hat{q}, m)$ for Algorithm.2
            **end for**
        **end if**
    **end for**
    Remove repeated elements inside $O_{t+1, \leq c}$, with corresponding mask $M$
    Apply mask $M$ to $N_{t+1, \leq c}$
    **return** $O_{t+1, \leq c}, N_{t+1, \leq c}, B_t$

---

graph) between single-component supernodes and multi-component supernodes. We present the algorithm of constructing these kind of connections sequentially. That is, given knowing the connections between single-component supernodes and all multi-component supernodes with $\leq t$ number of nodes, to build the connections between single-component supernodes and all multi-component supernodes with $\leq t+1$ number of nodes. To build the full bipartite graph for $(k, c)(\leq)$ sets, we need to conduct the algorithm $k-1$ times sequentially.

For the clear of presentation we use for-loops inside Algorithm.2, while in practice we use matrix operations to eliminate for-loops.

### A.7 Dataset Details

Table 6: Dataset statistics.

| Dataset | Task | # Cls./Tasks | # Graphs | Avg. # Nodes | Avg. # Edges |
|---|---|---|---|---|---|
| EXP [1] | Distinguish 1-WL failed graphs | 2 | 1200 | 44.4 | 110.2 |
| SR25 [6] | Distinguish 3-WL failed graphs | 15 | 15 | 25 | 300 |
| CountingSub. [15] | Regress num. of substructures | 4 | 1500 / 1000 / 2500 | 18.8 | 62.6 |
| GraphProp. [17] | Regress global graph properties | 3 | 5120 / 640 / 1280 | 19.5 | 101.1 |
| ZINC-12K [19] | Regress molecular property | 1 | 10000 / 1000 / 1000 | 23.1 | 49.8 |
| QM9 [59] | Regress molecular properties | 19[2] | 130831 | 18.0 | 37.3 |

### A.8 Experimental Setup

Due to limited time and resource, we highly restrict the hyperparameters and fix most of hyperparameters the same across all models and baselines to ensure a fair comparison. This means the performance of $(k, c)(\leq)$-SETGNN$^*$ reported in the paper is not the best performance given that we didn't tune much hyperparameters. Nevertheless the performance still reflects the theory and designs proposed in the paper, and we postpone studying the SOTA performance of $(k, c)(\leq)$-SETGNN$^*$ to future work. To be clear, we fix batch size to 128, the hidden size to 128, the number of layers

---

[2] We use the version of the dataset from PyG [20], and it contains 19 tasks.

**Algorithm 2** Constructing the bipartite graph between all single-component supernodes and all multi-component supernodes with $\leq t+1$ number of nodes)

---

**Input:** $\{B_i\}_{i=1}^{t}$, $\{O_{i,\leq c}\}_{i=1}^{t+1}$, $\{N_{i,\leq c}\}_{i=1}^{t+1}$ from Algorithm.1 (with blue lines applied), dictionary $D_{\leq t}$ with each key being a multi-component supernode with $\leq t$ nodes, and value being a list of its corresponding single-component supernodes.
**Output:** $D_{\leq t+1}$
$\quad D_{\leq t+1} \leftarrow D_{\leq t}$
$\quad$**for all** $\hat{q}$ in $O_{t+1,\leq c}$ **do**
$\quad\quad$ Get all edges $\bar{T} = \{(\hat{s}_i, \hat{q}), m_i\}$ in $B_t$ with $\hat{q}$ inside, and let $j \leftarrow \arg\max_i N_{t,\leq c}[\hat{s}_i]$
$\quad\quad$**if** $N_{t,\leq c}[\hat{s}_j] == N_{t+1,\leq c}[\hat{q}]$ **then** $\qquad\qquad\qquad$ ▷ No increasing of components
$\quad\quad\quad D_{\leq t+1}[\hat{q}] \leftarrow []$
$\quad\quad\quad$**for all** $\hat{p}$ in $D_{\leq t}[\hat{s}_j]$ **do**
$\quad\quad\quad\quad$ Assume $k = |\hat{p}|$, i.e. the number of nodes inside.
$\quad\quad\quad\quad$ Traverse $B_k$ to find the supernode $(\hat{p}, m_j)$
$\quad\quad\quad\quad$**if** Found it and $N_{k+1,\leq c}[(\hat{p}, m_j)] == 1$ **then**
$\quad\quad\quad\quad\quad$ Append $(\hat{p}, m_j)$ to $D_{\leq t+1}[\hat{q}]$
$\quad\quad\quad\quad$**else**
$\quad\quad\quad\quad\quad$ Append $\hat{p}$ to $D_{\leq t+1}[\hat{q}]$
$\quad\quad\quad\quad$**end if**
$\quad\quad\quad$**end for**
$\quad\quad$**else** $\qquad\qquad\qquad\qquad\qquad\qquad\qquad\qquad\qquad$ ▷ $N_{t,\leq c}[\hat{s}_j] = N_{t+1,\leq c}[\hat{q}] - 1$
$\quad\quad\quad D_{\leq t+1}[\hat{q}] \leftarrow D_{\leq t}[\hat{s}_j] + [m_j]$
$\quad\quad$**end if**
$\quad$**end for**
$\quad$**return** $D_{\leq t+1}$

---

of Base GNN to 4, and the number of layers of $(k,c)(\leq)$-SETGNN$^*$ (the number of iterations of $(k,c)(\leq)$-SETWL) to be 2 (we will do ablation study over it later). This hyperparameters configuration is used for all datasets. We have run the baseline GINE over many datasets, and we tune the number of layers from [4,6] and keep other hyperparameters the same as $(k,c)(\leq)$-SETGNN$^*$. For all other baselines, we took the performance reported in [68].

For all datasets except QM9, we follow the same configuration used in [68]. For QM9, we use the dataset from PyG and conduct regression over all 19 targets simultaneously. To balance the scale of each target, we preprocess the dataset by standardizing every target to a Gaussian distribution with mean 0 and standard derivation 1. The dataset is randomly split with ratio 80%/10%/10% to train/validation/test sets (with a fixed random state so that all runs and models use the same split). For every graph in QM9, it contains 3d positional coordinates for all nodes, and we use them to augment edge features by using the absolute difference between the coordinates of two nodes on an edge for all models. Notice that our goal is not to achieve SOTA performance on QM9 but mainly to verify our theory and effectiveness of designs.

We use Batch Normalization and ReLU activation in all models. We use Adam optimizer with learning rate 0.001 in all experiments for optimization. We repeat all experiments three times (for random initialization) to calculate mean and standard derivation. All experiments are conducted on V100 and RTX-A6000 GPUs.

## A.9 Graph Property Dataset Results

Table 7: Train and Test performances of $(k, c)(\leq)$-SETGNN* on regressing graph properties by varying $k$ and $c$.

| | | Regressing Graph Properties ($\log_{10}$(MSE)) | | | | | |
|---|---|---|---|---|---|---|---|
| | | Is Connected | | Diameter | | Radius | |
| k | c | Train | Test | Train | Test | Train | Test |
| 2 | 1 | -4.2266 ± 0.1222 | -2.9577 ± 0.1295 | -4.0347 ± 0.0468 | -3.6322 ± 0.0458 | -4.4690 ± 0.0348 | -4.9436 ± 0.0277 |
| 3 | 1 | -4.2360 ± 0.1854 | -3.4631 ± 0.6392 | -4.0228 ± 0.1256 | -3.7885 ± 0.0589 | -4.4762 ± 0.1176 | -5.0245 ± 0.0881 |
| 4 | 1 | -4.7776 ± 0.0386 | -4.9941 ± 0.0913 | -4.1396 ± 0.0442 | -4.0122 ± 0.0071 | -4.2837 ± 0.5880 | -4.1528 ± 0.9383 |
| 2 | 2 | -4.6623 ± 0.3170 | -4.7848 ± 0.3150 | -4.0802 ± 0.1654 | -3.8962 ± 0.0124 | -4.5362 ± 0.2012 | -5.1603 ± 0.0610 |
| 3 | 2 | -4.2601 ± 0.3192 | -4.4547 ± 1.1715 | -4.3235 ± 0.3050 | -3.9905 ± 0.0799 | -4.6766 ± 0.1797 | -4.9836 ± 0.0658 |
| 4 | 2 | -4.8489 ± 0.1354 | -5.4667 ± 0.2125 | -4.5033 ± 0.1610 | -3.9495 ± 0.3202 | -4.4130 ± 0.2686 | -4.1432 ± 0.4405 |

## A.10 Ablation Study over Number of Layers

Table 8: Ablation study of $(k, c)(\leq)$-SETGNN* over number of bidirectional message passing layers (L) on ZINC dataset.

| L | | 2 | | 4 | | 6 | |
|---|---|---|---|---|---|---|---|
| k | c | Train | Test | Train | Test | Train | Test |
| 2 | 1 | 0.1381 ± 0.0240 | 0.2345 ± 0.0131 | 0.1135 ± 0.0418 | 0.1921 ± 0.0133 | 0.0712 ± 0.0015 | 0.1729 ± 0.0134 |
| 3 | 1 | 0.1172 ± 0.0063 | 0.2252 ± 0.0030 | 0.0792 ± 0.0190 | 0.1657 ± 0.0035 | 0.0692 ± 0.0118 | 0.1679 ± 0.0061 |
| 4 | 1 | 0.0693 ± 0.0111 | 0.1636 ± 0.0052 | 0.0700 ± 0.0085 | 0.1566 ± 0.0101 | 0.0768 ± 0.0116 | 0.1572 ± 0.0051 |

The table shows that increasing the number of bidirectional message passing ($t$ in Eq.(13) and Eq.(14) ) always increase the performance, which is aligning with the fact that increasing number of layers always increases expressivity.

## A.11 Computational Footprint on QM9

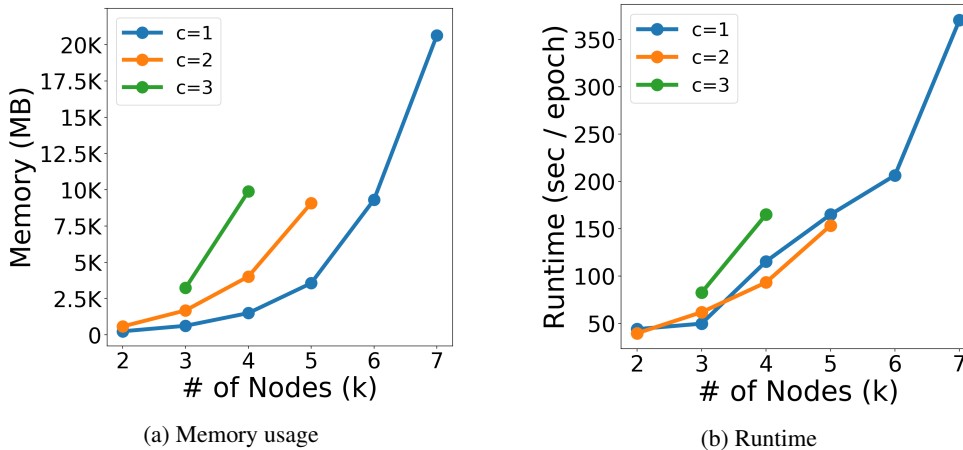

(a) Memory usage

(b) Runtime

Figure 5: $(k, c)(\leq)$-SETGNN's computational footprint scales with both $k$ and $c$ in terms of memory (a) and runtime (b). Solid blue, orange and green lines track scaling as $k$ increases, when running $(k, c)(\leq)$-SETGNN on the QM9 dataset with $c = 1$, 2 and 3 respectively.

## A.12 Discussion of k-Bipartite Message Passing

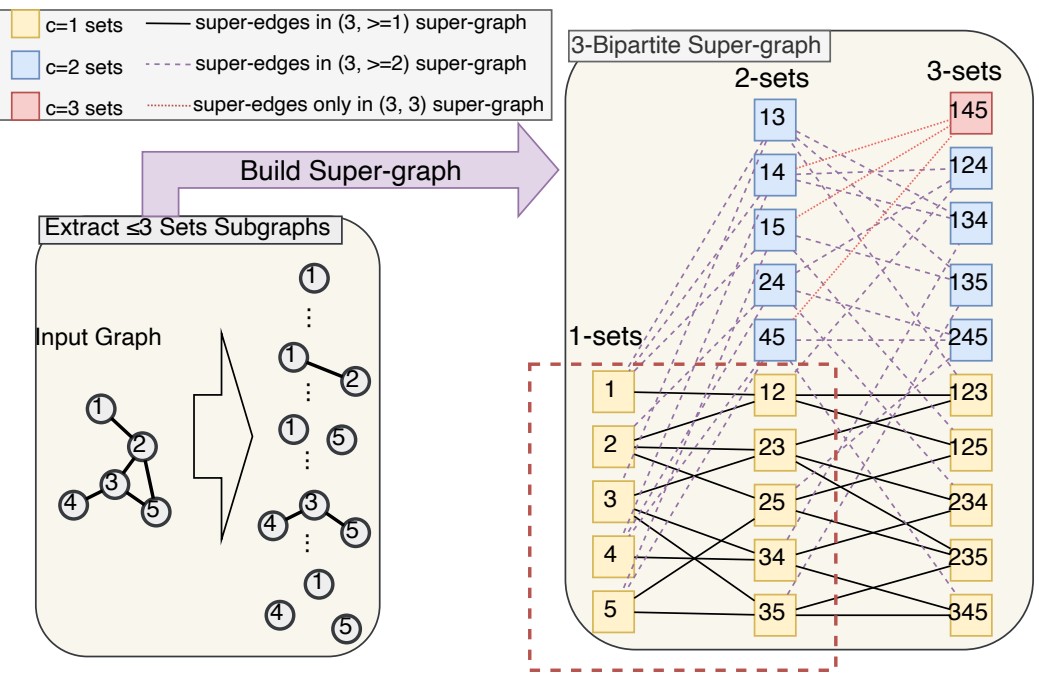

Figure 6: k-bipartite message passing recovers many well-known GNNs: message passing based GNNs [25], edge-enhanced GNNs [7], and line-graph GNNs [14, 16]. The line graph is marked with red dash frame.

Interestingly, the k-bipartite bidirectional message passing is very general and its modification covers many well known GNNs. Considering only using 1-sets and 2-sets with single connected components (the red frame inside Figure 6), and initializing their embeddings with their original nodes and edges representations, then it covers the follows

- Message passing based GNNs [25]. By using sequential message passing defined in Eq.(13) and Eq.(14), and performing forward step first then backward step for all 1-sets.

- Line graph based GNNs [14, 16]. By using sequential message passing defined in Eq.(13) and Eq.(14), and performing backward step first then forward step for all 2-sets.

- Relational Graph Networks [7] or edge-enhanced GNN [16]. By using performing bidirectional sequential message passing on all 1-sets and 2-sets.