# OpenReview forum: "A Practical, Progressively-Expressive GNN"
_NeurIPS.cc/2022/Conference — NeurIPS 2022 Accept_

### Official Review · Reviewer_KTkq · 2022-07-09

**Rating:** 4
**Confidence:** 4
**Soundness:** 2 fair
**Presentation:** 1 poor
**Contribution:** 1 poor

**Summary:**

The paper introduces a family of models, namely (k, c)(≤)-SETGNN, whose expressiveness increases with k and c.
The expressiveness is measured in terms of the newly proposed hierarchy (k, c)(≤)-SETWL which is more "granular" than the standard k-WL hierarchy.
The presentation guides the reader to the derivation of the (k, c)(≤)-SETWL by iteratively refining the k-WL hierarchy.
The paper comes with proofs relating the various refinements and experiments that investigate the progressive expressiveness of the proposed architecture for varying k and c.


**Questions:**

- Is it possible to analyse the expressiveness of existing approaches in term on the new hierarchy?
- Did you use the parameter budget (100K or 500K) as prescribed by ZINC-12K?
- The linked repository is empty, please update it.

**Ethics Review Area:**

["I don’t know"]

**Strengths And Weaknesses:**

**Strengths**
- The empirical evaluation shows increased performance wrt existing methods
- The proposed architecture's expressiveness can be modulated easily through the k and c parameters.

**Weaknesses**
- It's unclear whether one can use the new proposed "ruler" (the (k, c)(≤)-SETWL hierarchy) to compare existing models and whether new models other than (k, c)(≤)-SETGNN can be built around it. Consequently, the significance appears rather limited.


Minor comments:
- $s$ is not defined in line 135 but introduced much later in line 175
- Fig. 1 needs a few more iterations. Many edges are disconnected from their endpoints and the labels (1, 5 etc.) of the nodes are not centered.

---

> ### Author Response · Authors · 2022-08-02
> **Response to KTkq**
>
> We thank the reviewer for their questions and feedback. We address questions one by one.
>
> >**It's unclear whether one can use the new proposed "ruler" (the (k, c)(≤)-SETWL hierarchy) to compare existing models and whether new models other than (k, c)(≤)-SETGNN can be built around it.**
>
> We would like to clarify the motivation and goal of the design of (k,c)(≤)-SETWL. The main goal is to replace the current k-WL for **practically** (while keeping connection to k-WL theoretically) measuring expressivity as the k-WL is only a good tool in theory but is not usable in practice for k larger than 3. However there is the gap between theoretical expressivity and real-world performance which can only be investigated with “practical k-WL”.
> The proposed model is to help the community to study the effect of expressivity experimentally so that the community can answer the very important question: “Do we need an extremely expressive model in practice for all kinds of datasets? When do we need extremely powerful GNNs?” With the help of the proposed model, researchers can now start to study the relationship between expressivity and dataset/task hardness, and also the relationship between expressivity and generalization. To summarize, the GNN community that is focusing on developing more and more expressive GNNs is at a stage that really needs to study the effect of expressivity experimentally before putting more effort on “chasing” highly expressive models, with the help of our proposed model.
>
> >**s Is not defined in line 135 but introduced much later in line 175**
>
> Thank you for pointing out that, there is in fact a typo. We have updated the paper to remove s in line 135.
>
> >**Is it possible to analyze the expressiveness of existing approaches in terms of the new hierarchy?**
>
> Most of the existing approaches analyze their model in the hierarchy of k-WL. The proposed model is closely related to k-WL and is possibly equivalent to k-WL if the conjecture is true. Based on the close relationship, we think that the analysis is possible but is nontrivial.
> What’s more, the goal of our proposed model is mainly to be able to study expressivity experimentally while keeping the theoretical connection to k-WL.
>
> >**Did you use the parameter budget (100K or 500K) as prescribed by ZINC-12K?**
>
> We don’t control the experiment based on the parameter budget, instead we control the experiment by fixing hidden size (128) and number of layers (4 for BaseGNN and 2 for bidirectional propagation), so that changing k and c will keep all other hyperparameters fixed.
> The 100K and 500K parameter control involves tuning the model by reducing hidden size and improving the number of layers (tradeoff between depth and width) which needs more computing resources to find the balance. We plan to work on a rigorous study for all kinds of hyperparameters like depth and width in another project in which we are studying the effect of expressivity using the proposed model. Note that this means the current reported performance still has room to be improved with hyperparameter search.
>
>
> >**The linked repository is empty, please update it.**
>
> Thank you for showing interest in our code. We will update it after the paper decision. The current code is used in another project we are working on, and needs the extra time for packaging and documenting.

---

> > ### Comment · Reviewer_KTkq · 2022-08-04
> > **Response to the authors**
> >
> > I thank the authors for the extra efforts in their reply. I will address the above points below.
> >
> > **Usage of the (k, c)(≤)-SETWL hierarchy to compare existing models and to propose new architectures other than (k, c)(≤)-SETGNN.**
> >
> > I understand that in the reply the authors point out that, while it is not currently possible to compare existing methods with the newly proposed ruler, the goal is to practically measure how increased expressivity impacts the performance and the generalization abilities in real world tasks. However:
> > 1. This is not exactly what is studied in the experimental section, since Table 1 and Table 3 do not show how changing the expressive power impacts the results.
> > 2. (k, c)(≤)-SETWL does not become _strictly_ more expressive with increasing k and c.
> > Indeed, (k, c)(≤)-SETWL is only _at least as_ expressive with increasing k and c (Theorem 3). Therefore, one cannot use (k, c)(≤)-SETWL to study how increased expressivity impacts performances since increasing k or c is not proven to increase the expressivity.
> >
> > **Parameter budget as prescribed by ZINC-12K**
> >
> > I will rephrase my question: how many parameters do you use for your model in Table 3? The models you are comparing with in that table are compliant with the 100K parameter budget. A comparison not accounting for the number of parameters (as prescribed by the dataset) is not fair, and cannot be deemed SoTA.
> >
> > **Related work.**
> >
> > It looks like this paper has a very strong overlap with the ICLR 22 GTRL workshop paper "SpeqNets: Sparsity-aware Permutation-equivariant Graph Networks". Please compare and contrast thoroughly the two works.
> >
> > https://openreview.net/forum?id=rc8x9VZJpe5

---

> > > ### Author Response · Authors · 2022-08-07
> > > **Response to further questions [1/2]**
> > >
> > > >”This is not exactly what is studied in the experimental section, since Table 1 and Table 3 do not show how changing the expressive power impacts the results.”
> > >
> > > We have stated that Table 1 and Table 3 are used to compare with SoTA, while Table 4 and Table 5 are used to show the performance with respect to different expressivity (changing k and c), with keeping all settings the same (number of layers and hidden size).
> > >
> > > >“(k, c)(≤)-SETWL does not become strictly more expressive with increasing k and c.”
> > >
> > > We disagree that this model is not strictly more powerful with increasing k and c, as the results in table 4 and table 5 show its strength. However, we acknowledge that giving exact proof is not trivial. To theoretically prove that (k, c)(≤)-SETWL is strictly more powerful with increasing k and c, we have to construct a series of nonisomorphic pairs (G_k, H_k) such that they can be distinguished by (k+1, c)(≤)-SETWL but not by (k+1, c)(≤)-SETWL.
> > > We think the CFI graphs (G_k, H_k) constructed in [4] could be the one, which are graphs that can be distinguished by (k+1)-WL but not k-WL. If the conjecture of the expressivity equivalence between MultisetWL and SetWL is correct, this immediately implies that the strictness is correct. If we don’t rely on the relationship between SetWL and MultisetWL and to formally prove that, we need to transform the current SetWL to a modified pebble game and then use the pebble game to prove that two graphs (G_k, H_k) can be distinguished by (k+1, k+1)(≤)-SETWL but not by (k, k)(≤)-SETWL. Indeed this is a good suggestion to further strengthen the theoretical contribution, and we leave it to future work.
> > >
> > > >”how many parameters do you use for your model in Table 3? The models you are comparing with in that table are compliant with the 100K parameter budget.”
> > >
> > > We thank the reviewer for raising the question about parameter budget, and we are willing to report that. To reduce the experimental time, we focus on k=4 and c=1 and 2. Notice that the number of parameters is fixed for variable c, that is, (k=4,c=1) has the same number of parameters as (k=4,c=2). We chose the setting because the reviewer argued that the performance improvement is possible due to an increasing number of parameters, and k=4 is faster to run. In the following table we report the number of parameters of the original setting, and the one required as a 500K budget.
> > >
> > > |  Setting    | 1200K Parameters | 500K Parameters|
> > > |---|---|---|
> > > | Note | Original, #hidden=128 | New, #hidden=80|
> > > | k=4, c=1   |    0.1636 ± 0.0052   |  0.1742 ± 0.0090 |
> > > | k=4, c=2   |    0.0836 ± 0.0010   |  0.0862 ± 0.0050  |
> > >
> > > As shown in the table, there is not much performance reduction by reducing the hidden size and parameter budget. On the contrary, with the same number of parameters, there is a large performance gain by changing c from 1 to 2. We hope this can make the source of improvement much clearer.
> > >
> > > >”SpeqNets: Sparsity-aware Permutation-equivariant Graph Networks". Please compare and contrast thoroughly the two works.”
> > >
> > > We thank the reviewer for pointing it out. We have revised the related work section to include Morris’s new paper. By looking carefully at the paper, we find that the idea of using different numbers of connected components is concurrently explored in the SpeqNet, but with a very different approach:
> > >
> > > 1. SpeqNet focuses on improving the delta-k-LWL [1] methods which is a **tuple** based WL algorithm that improves original WL by reducing the number of connections among supernodes. Notice that our main contribution that greatly reduces the complexity is by moving from tuple to multiset and to set. Because of this, in the experimental section SpeqNet can only run with (k=3,c=2) which is still limited by 3-WL (notice that 3-WL-powerful GNN can be implemented with PPGN [2] directly), while we can run with (k=10, c=1) or (k=7,c=2) or (k=4, c=3).
> > >
> > > 2. SpeqNet designs the neural version GNN directly based on the supergraph defined by k-WL, while we greatly sparsify the supergraph with the carefully designed k-biparate supergraph and corresponding bipartite GNN without losing any expressivity. Even more, our model is a pure neural network without relying any isomorphism algorithm to process, while SpeqNet is actually running with a two-stage processing: using isomorphism algorithm to extract the atomic type of every subgraph, construct the supergraph, and then run
> > > delta-k-LWL [1] based GNN.
> > >
> > > [See part 2 for more comparisons]

---

> > > > ### Author Response · Authors · 2022-08-07
> > > > **Response to further questions [2/2]**
> > > >
> > > > We have to emphasize that 1) using the different number of connected components is an independent idea but is concurrently explored in SpeqNet; 2) this part is NOT our main contribution, and it only takes one subsection 3.4 (less than half page). We would like to summarize our contributions formally:
> > > >
> > > > 1. We theoretically design different WL algorithms by moving from tuple to multiset, and from multiset to set. And we studied their expressiveness and scalability. Notice that the sets are not sets with fixed $k$ number of elements which are explored in [3] (without any theoretical analysis), but are sets with $\leq k$ number of elements which are first proposed and analyzed.   [Reduce number of supernodes]
> > > >
> > > >
> > > > 2. We further reduce the number of sets for sparse graphs by restricting the number of connected components, with theoretical analysis.  [Reduce number of supernodes]
> > > >
> > > >
> > > > 3. We transform the original supergraph defined in the k-SetWL to a k-bipartites supergraph equivalently in Eq. (6), which greatly reduces the number of superedges from $\sum_{2=1}^{k} \frac{n+i}{2} \binom{n}{i}$ to $\sum_{2=1}^{k} i \binom{n}{i}$, without any reduction in expressivity.  [Reduce number of superedges]
> > > >
> > > >
> > > > 4. We design a sophisticated, fully end-to-end GNN for the $(k,c)(\leq)-SetWL$, based on the k-biparatites supergraph with bidirectional propagation. Furthermore, the sequential bidirectional propagation and the lazy  supernode initialization are designed to reduce the memory cost and improve the runtime without any expressivity reduction.
> > > >
> > > >
> > > > 5. With all introduced designs and their theoretical analysis, we achieved SoTA performance on ZINC by a large margin, and we showed the effect of progressively changing expressivity with $k$ and $c$ in Table 4 and Table 5. For the first time, this is the first implementable model with theoretical analysis that is not limited by k=3.
> > > >
> > > > We kindly ask the reviewer to consider revising the score if we have addressed the reviewer’s concerns.
> > > >
> > > >
> > > > [1] Morris, Christopher, Gaurav Rattan, and Petra Mutzel. "Weisfeiler and Leman go sparse: Towards scalable higher-order graph embeddings." Advances in Neural Information Processing Systems 33 (2020): 21824-21840.
> > > >
> > > > [2] Maron, H., Ben-Hamu, H., Serviansky, H., & Lipman, Y. (2019). Provably powerful graph networks. Advances in neural information processing systems, 32.
> > > >
> > > > [3] Morris, Christopher, et al. "Weisfeiler and leman go neural: Higher-order graph neural networks." Proceedings of the AAAI conference on artificial intelligence. Vol. 33. No. 01. 2019.
> > > >
> > > > [4] Cai, Jin-Yi, Martin Fürer, and Neil Immerman. "An optimal lower bound on the number of variables for graph identification." Combinatorica 12.4 (1992): 389-410.

---

> > > > > ### Comment · Reviewer_KTkq · 2022-08-08
> > > > > **Response to the authors**
> > > > >
> > > > > I thank the authors for their reply. However,
> > > > >
> > > > > 1. I still believe that a proof of (k, c)(≤)-SETWL becoming strictly more expressive with increasing k and c is needed to achieve the aforementioned goal, which is to study how increasing expressivity improves performances. The fact that you can run with (k=10, c=1) or (k=7,c=2) or (k=4, c=3) is not impressive if those are not strictly more expressive than (k=3,c=2) (which is the one considered in SpeqNet). Indeed, if strictness is not true, they are equally expressive and it is unclear why one would need higher k and c.
> > > > >
> > > > > 2. Since your Test MAE within the 500K parameter budget is 0.0862 ± 0.0050, you did not obtain SOTA results (CIN gets 0.079 ± 0.006 while staying in the same budget). As I was saying, ZINC-12K requires the methods to be compliant to one of the two budgets, and to compare new methods to existing one abiding to that budget. Therefore, I would suggest removing your statement about SOTA and updating your Table with the new results (which are still very good). Moreover, I strongly encourage the authors to revisit Table 3, since you are reporting some methods that are compliant to the 100K budget, and some others to the 500K budget.

---

> > > > > > ### Author Response · Authors · 2022-08-09
> > > > > > **Thank you for additional suggestions**
> > > > > >
> > > > > > We thank the reviewer's question to help improving the paper. We would like to provide additional experimental results and explanations to answer these questions.
> > > > > >
> > > > > > 1. Same as the reviewer, we are interested and attracted on proving the strict increasing of expressiveness with changing k and c. We have referenced several papers for the proof of showing that (k+1)-WL is strictly more powerful than k-WL. However, even for this well-known result, the proof is hard to find. Many papers like [1,2,3] cited [4] for the claim that (k+1)-WL is strictly more powerful than k-WL. However [4] only provides a series of graphs (G_k, H_k) that k-WL cannot distinguish, without claiming that (G_k, H_k) can be distinguished by (k+1)-WL. Hence we believe that theoretical proof of the strict expressiveness is non-trivial and worth a separate paper to target for future work. However, empirically, Table 1's results on EXP (2-WL indistinguishable graphs) and SR25 (3-WL indistinguishable graphs) already shows that the strictness hold: only with k>=3 and c>=2 the (k,c)(<=)-SetWL can distinguish 2-WL indistinguishable graphs, and only with k>=4 and c>=1 the (k,c)(<=)-SetWL can distinguish 3-WL indistinguishable graphs. These empirical results support the strict increasing of expressiveness --- at least equal expressiveness with changing k and c is impossible.
> > > > > >
> > > > > > 2. We appologize for only showing k=4 and c=2 case for 500k parameter budget in the last reply. To answer the reviewer's question with budget limitation for fair evaluation, we again run k=5 and c=2 case for 500k parameter budget, which is exactly the setting for the result in Table 1. The below table shows that the **SoTA performance still holds**
> > > > > >
> > > > > > | Setting| Original, Hidden=128,1600K params| New, Hidden=72, 500K params|
> > > > > > | ----------- | ----------- | ----------- |
> > > > > > | k=5, c=2 |0.0750 ± 0.0027 |**0.0765 ± 0.0034**|
> > > > > >
> > > > > > Moreover, we would like provide the reason of choosing fixed hidden size 128 for table 4 and table 5. Table 4 and table 5 is intended to show the effect of changing expressivity through changing k and c. Based on Theorem 4, the expressivity of any GNN is only achieved with sufficient wide MLPs. To let proposed GNN to achieve the claimed expressivity, we want hidden size to be wide enough to isolate the effect of expressivity of MLPs. However with limited GPU memory we can only choose fixed hidden size. Hence we isolate the effect of expressivity of MLPs for different k and c by choosing fixed hidden size for across all settings. To be more clear, choosing fixed number of parameters is not the correct setting for studying expressivity.
> > > > > >
> > > > > >
> > > > > >
> > > > > > We thank sincerely for the reviewer's questions to further improve the paper. We are happy to address the reviewer's any additional concerns. We also kindly ask the reviewer considering whether all five contributions of the paper mentioned in last reply are meaningless, as the current score suggests the paper makes no contributions.
> > > > > >
> > > > > > [1] Grohe, Martin. "The logic of graph neural networks." 2021 36th Annual ACM/IEEE Symposium on Logic in Computer Science (LICS). IEEE, 2021.
> > > > > > [2] Maron, Haggai, et al. "Provably powerful graph networks." Advances in neural information processing systems 32 (2019).
> > > > > > [3] Jegelka, Stefanie. "Theory of Graph Neural Networks: Representation and Learning." arXiv preprint arXiv:2204.07697 (2022).
> > > > > > [4] Cai, Jin-Yi, Martin Fürer, and Neil Immerman. "An optimal lower bound on the number of variables for graph identification." Combinatorica 12.4 (1992): 389-410.

---

> > > > > > > ### Comment · Reviewer_KTkq · 2022-08-09
> > > > > > > **Final Answer**
> > > > > > >
> > > > > > > I encourage the authors to use the empirical results to prove strictness for some k and c.
> > > > > > > That is, use some pairs in EXP and SR to show they become distinguishable only with greater k and c.
> > > > > > > Also, please note that SpeqNet proves strictness.
> > > > > > >
> > > > > > > I have increased my score to a Borderline reject.

---

### Official Review · Reviewer_ZZnY · 2022-07-11

**Rating:** 5
**Confidence:** 4
**Soundness:** 3 good
**Presentation:** 4 excellent
**Contribution:** 3 good

**Summary:**

The goal of this paper is to develop more expressive GNNs using the k-WL hierarchy. While the one-dimensional Weisfeiler Leman (1-WL) helps to distinguish isomorphic graphs, the use of k-WL hierarchy helps build more expressive GNNs.  The authors propose (k,c) <= SETWL which reduces the complexity of the k-WL hierarchy. Tweaking k and c in this model, helps develop practical and progressively expressive GNNs.


**Questions:**

- What is the basic definition of "expressiveness" of a GNN model? Does increased expressiveness of the super-graph imply better generalization performance?

- Is expressiveness related to interpretability of the model? If not, what marks the fundamental difference between them?

- Why is the work consistently using GNNs when the literature is replete with many other kinds of neural networks whose expressiveness can be studied and interpreted?

- The motivation for (k,c) (<=)-SETGNN has not been well discussed.

 - Section 5: What is a SOTA expressive GNN? The acronym has not been defined.
 -For ease of readability, consider having a notation table instead of the section 3
 -Table 1,2,3,4 and 5 should simply use two places of decimals for ease of readability

 The authors claim "Such a hierarchy could enable us to gradually build more expressive models which admit improved scaling, and avoid unnecessary leaps in model complexity for tasks which do not require them, guarding against overfitting."
 What is the relation between expressivity of a graph and graph isomorphism?

 In Table 2, with k=3 or 4, and more than one connected components, the performance on the test set is much worse than on the train set. So for these highly expressive graphs, how is generalization performance demonstrated to be better?



**Limitations:**

Complexity of the proposed technique in addition to scalability are clearly limitations which the authors try to solve and are successful to a good extent.

**Strengths And Weaknesses:**

Strengths
- The paper is well written with both theoretical and empirical work.
- An interesting problem on expressiveness of GNNs is explored using set theoretic tools along with message passing algorithms

Weakness
- The problem needs to be better motivated -- how is expressiveness related to generalization performance? Why use (k,c) (<=)-SETGNN at all?
- The paper assumes an audience well versed with graph isomorphism, Weisfeiler Leman hierarchy, and other kinds of isomorphism test. A reader unfamiliar with these concepts would wonder why even this is relevant. Motivating with concrete real world examples would perhaps make it easier to see why this is an important problem to solve.

---

> ### Author Response · Authors · 2022-08-02
> **Response to ZZnY**
>
> We thank the reviewer for giving detailed feedback. We answer the reviewer’s questions one-by-one.
>
>
>
> >**What is the basic definition of "expressiveness" of a GNN model? Does increased expressiveness of the super-graph imply better generalization performance?**
>
> The expressiveness of GNN mainly refers to the ability of separating two non-isomorphic graphs to different representations, as Chen et al. (2019) proved the equivalence
> between perfect separation and learning universal permutation invariant functions on graphs. Unlike MLP, which is a universal function approximator on continuous vector space, GNN is not universal for approximating functions on graphs. Researchers use the separation ability to measure expressiveness, which reflects the “closeness” to the universal graph approximator.
>
> >**Is expressiveness related to interpretability of the model? If not, what marks the fundamental difference between them?**
>
> They are two different properties of a model. But in general the more expressive a model is, the more complex it gets, and hence the harder it becomes to interpret. The interpretability of GNNs is a relatively new area, given the current studies are focusing on expressiveness and generalization ability.
>
> >**Why is the work consistently using GNNs when the literature is replete with many other kinds of neural networks whose expressiveness can be studied and interpreted?**
>
> As mentioned above, the expressiveness of GNN is a very different problem from other neural networks like CNN and MLP, as they are universal function approximators already. The reason is that the target function space is different. All other widely used neural networks are working over vector function space, while GNNs are working over combinatorial graph function space which is significantly more complicated and does not exhibit a universal approximator.
>
> >**The problem needs to be better motivated -- how is expressiveness related to generalization performance? Why use (k,c) (<=)-SETGNN at all?**
>
> We apologize for focusing on expressiveness directly without giving enough background. We would like to clarify here to show the importance of the problem studied in the paper. The story comes from the fact that GNN is not a universal function approximator on graphs and so many works target at designing more powerful GNNs in 2020 and 2021, as it DOES help to improve real-world performance. However designing a more powerful GNN is extremely hard and it often makes the model not scalable (one needs to increase complexity to trade expressivity). In 2022, increasing model (i.e. GNN) expressivity even further became a hot topic, however important questions remain to be answered in this field: “Should we chase expressivity? Will adding more expressivity further benefit real-world tasks? Or will increased expressivity hurt generalization performance, deeming high expressivity unnecessary in practice?” To answer these questions and to measure the effectiveness of expressivity, the field needs a scalable model which has the ability of varying its expressivity in a fine-grained fashion (i.e. gradually increasing expressivity). Currently, there exists no work that is able to answer these aforementioned questions because there is no such implementable powerful model with varying expressivity. That is exactly the motivation of our paper: with the proposed relatively scalable and implementable model with varying expressivity, our model can be used to formally study the important question in the field about expressivity. Systematically answering questions around expressivity in practice can greatly guide future research in the field.
>
> >**Table 1,2,3,4 and 5 should simply use two places of decimals for ease of readability.**
>
> Thank you for suggesting, we followed previous work of 4 decimals. We will reduce it in the final version.
>
> >**What is the relation between expressivity of a graph and graph isomorphism?**
>
> Graph isomorphism is used to characterize the separation ability among graphs, which is used to measure expressivity of a model.
>
> >**In Table 2, with k=3 or 4, and more than one connected component, the performance on the test set is much worse than on the train set. So for these highly expressive graphs, how is generalization performance demonstrated to be better?**
>
> Usually the test performance is worse than the training performance which is expected. The generalization question is really a good one, but there is not enough study about GNN’s generalization ability in the field yet. Also to study the relationship between expressivity and generalization, one needs the powerful and implementable model proposed in the paper. We are actually working on studying the effect of expressivity and generalization in a different project using the proposed model in this paper.
>
> Reference:
> [Chen et al. 2019] On the equivalence between graph isomorphism testing and function approximation with GNNs. NIPS 2019.

---

> > ### Author Response · Authors · 2022-08-09
> > **Asking for additional feedback**
> >
> > Thank you for providing suggestions for the paper. As the current discussing stage is about to end, we are willing to ask if the reviewer still have any additional concerns for the paper. If all concerns are addressed, we kindly ask the reviewer considering raise the score to support.

---

### Official Review · Reviewer_g44M · 2022-07-12

**Rating:** 7
**Confidence:** 4
**Soundness:** 3 good
**Presentation:** 4 excellent
**Contribution:** 4 excellent

**Summary:**

This work proposes a more fine-grained graph isomorphism test hierarchy, namely (k,c)(<=)-SetWL, by further considering removing order, removing repetition, and graph sparsity over the original k-WL algorithm. Based on the obtained graph isomorphism test algorithm, it further develops a progressively expressive GNN accordingly. Experiments on synthetic and real datasets are performed to evaluate the effectiveness of the proposed GNN.

**Questions:**

(1) I am not fully understand the derivation of Theorem 4 ((k,c)(<=)-SetGNN is as expressive as (k,c)(<=)-SetWL). In the given proof, it is mainly based on the results of DeepSet. In other words, MLP with summation is universal so that it can learn the desired injective function. However, this may not prove that Eq. (9) is injective directly, since the input of the outer MLP in Eq. (9) has four parts. It is unclear and non-trivial that how to ensure that this MLP can learn an injective function.

+++minor points
Line 182, a should be an


**Limitations:**

Not applicable.

**Strengths And Weaknesses:**

+++ Pros

(1) It is technically sound to consider removing order, removing repetition, and graph sparsity over the original k-WL algorithm. The step-by-step derivation is sound and clearly present.

(2) This work also characterizes the expressiveness of the obtained k-WL variants clearly. It is impressive to me to know that k-MultisetWL is as expressive as the original k-WL. This result might have impact to the community.

(3) The presentation of this work is great. Although it has a lot of math, the flow is generally easy to follow.

(4) The empirical results are strong to demonstrate the effectiveness of the proposed GNN.

+++ Cons

(1) Although the presentation is good enough, there are several places which are hard to understand, given that this is a math-heavy work. For example, the result given in Theorem 1 is amazing, it would be great to provide some intuitive explanation of the proof in the main text. This can also help readers to understand why we consider removing orders.

(2) Similarly, the derivation of Theorem 4 is also unclear to me. Please see the following question for details.

---

> ### Author Response · Authors · 2022-08-02
> **Response to g44M**
>
> We thank the reviewer for valuable comments and supportive feedback. We answer the reviewer’s questions in the following.
>
> >**The result given in Theorem 1 is amazing, it would be great to provide some intuitive explanation of the proof in the main text.**
>
> There are mainly two intuitions for the proof, one for proving that two same colored supernodes in k-MWL leads to same colored supernodes in k-MWL, and the other for the reverse direction. Taking k=3 for example. For k-WL, when the color of (v1, v2, v3) in G is the same as the color of (u1, u2, u3) in G’, so is the color of (v1, v3, v2) and (u1, u3, u2) or any two tuples from applying same permutation to (v1, v2, v3) and (u1, u2, u3). And when two colors are different, any different permutations to (v1, v2, v3) and (u1, u2, u3) will have different colors. This implies that in k-WL, the ordering doesn’t carry additional information. For the reverse direction it’s a bit harder, but the main strategy is by looking at the set of isomorphisms making two supernodes having the same color. And the set of isomorphisms will shrink with increasing iterations of k-MWL.
>
>
> >**However, this may not prove that Eq. (9) is injective directly, since the input of the outer MLP in Eq. (9) has four parts. It is unclear and non-trivial how to ensure that this MLP can learn an injective function.**
>
> We apologize for the informal proof of theorem 4 and we will update it in the final version. We clarify the reviewer’s concern here. Our goal is to prove that Eq.(9) has the same expressiveness as Eq.(6) which is a rewritten version of k-SetWL. As MLP can approximate any functions under enough depth and width, for sure it can approximate an injective function. When the outer MLP approximates the injective function, Two outputs are the same if and only if each part (there are four parts) of the outer MLP input is the same. And for each part, the universal ability of DeepSet helps to connect each part inside Eq.(9) to the corresponding part inside Eq.(6). We hope this informal intuition can clarify the reviewer’s question and we will update it formally in the final version.

---

> > ### Comment · Reviewer_g44M · 2022-08-05
> > **Please complete the formal proof, and clarify the difference with SpeqNets**
> >
> > Thank the authors for the response. I still have the following two concerns.
> >
> > (1) Since Eq. (9) is the key result of this paper. It would be more convincing to provide a formal proof for reviewing. The current proof is not enough to support the theorem 4.
> >
> > (2) As Reviewer KTkq, I also noticed that this work is largely related and overlapping with SpeqNets [1] (its Section 3 and 4). They also use set-based tuple with considering connected component and sparsity of the graph. An extensive and clear comparison/discussion with this work should be added.
> >
> > Thank you.
> >
> > [1] Morris et al. Sparsity-aware Permutation-equivariant Graph Networks. ICML 2022.

---

> > > ### Author Response · Authors · 2022-08-07
> > > **Response to further questions**
> > >
> > > We thank the reviewer for further questions. We have worked on two concerns raised.
> > >
> > > >"Formal proof of Theorem 4."
> > >
> > > We have now included the formal proof in appendix, marked with blue. Please have a check.
> > >
> > > >"Difference to SpeqNet"
> > >
> > > We thank the reviewer for pointing it out. We have revised the related work section to include Morris’s new paper. By looking carefully at the paper, we find that the idea of using different numbers of connected components is concurrently explored in the SpeqNet, but with a very different approach:
> > >
> > >
> > > 1. SpeqNet focuses on improving the delta-k-LWL [1] methods which is a **tuple** based WL algorithm that improves original WL by reducing the number of connections among supernodes. Notice that our main contribution that greatly reduces the complexity is by moving from tuple to multiset and to set. Because of this, in the experimental section SpeqNet can only run with (k=3,c=2) which is still limited by 3-WL (notice that 3-WL-powerful GNN can be implemented with PPGN [2] directly), while we can run with (k=10, c=1) or (k=7,c=2) or (k=4, c=3).
> > >
> > >
> > > 2. SpeqNet designs the neural version GNN directly based on the supergraph defined by k-WL, while we greatly sparsify the supergraph with the carefully designed k-biparate supergraph and corresponding bipartite GNN without losing any expressivity. Even more, our model is a pure neural network without relying any isomorphism algorithm to process, while SpeqNet is actually running with a two-stage processing: using isomorphism algorithm to extract the atomic type of every subgraph, construct the supergraph, and then run
> > > \delta-k-LWL [1] based GNN.
> > >
> > > We have to emphasize that 1) using the different number of connected components is an independent idea but is concurrently explored in SpeqNet; 2) this part is NOT our main contribution, and it only takes one subsection 3.4 (less than half page). We would like to summarize our contributions formally:
> > >
> > > 1. We theoretically design different WL algorithms by moving from tuple to multiset, and from multiset to set. And we studied their expressiveness and scalability. Notice that the sets are not sets with fixed $k$ number of elements which are explored in [3] (without any theoretical analysis), but are sets with $\leq k$ number of elements which are first proposed and analyzed.   [Reduce number of supernodes]
> > >
> > >
> > > 2. We further reduce the number of sets for sparse graphs by restricting the number of connected components, with theoretical analysis.  [Reduce number of supernodes]
> > >
> > >
> > > 3. We transform the original supergraph defined in the k-SetWL to a k-bipartites supergraph equivalently in Eq. (6), which greatly reduces the number of superedges from $\sum_{2=1}^{k} \frac{n+i}{2} \binom{n}{i}$ to $\sum_{2=1}^{k} i \binom{n}{i}$, without any reduction in expressivity.  [Reduce number of superedges]
> > >
> > >
> > > 4. We design a sophisticated, fully end-to-end GNN for the $(k,c)(\leq)-SetWL$, based on the k-biparatites supergraph with bidirectional propagation. Furthermore, the sequential bidirectional propagation and the lazy  supernode initialization are designed to reduce the memory cost and improve the runtime without any expressivity reduction.
> > >
> > >
> > > 5. With all introduced designs and their theoretical analysis, we achieved SoTA performance on ZINC by a large margin, and we showed the effect of progressively changing expressivity with $k$ and $c$ in Table 4 and Table 5. For the first time, this is the first implementable model with theoretical analysis that is not limited by k=3.
> > >
> > >
> > > We hope you find our response helpful! If your concern is fully addressed, we kindly ask you can consider raise the score.
> > >
> > >
> > > [1] Morris, Christopher, Gaurav Rattan, and Petra Mutzel. "Weisfeiler and Leman go sparse: Towards scalable higher-order graph embeddings." Advances in Neural Information Processing Systems 33 (2020): 21824-21840.
> > >
> > > [2] Maron, H., Ben-Hamu, H., Serviansky, H., & Lipman, Y. (2019). Provably powerful graph networks. Advances in neural information processing systems, 32.
> > >
> > > [3] Morris, Christopher, et al. "Weisfeiler and leman go neural: Higher-order graph neural networks." Proceedings of the AAAI conference on artificial intelligence. Vol. 33. No. 01. 2019.

---

> > > > ### Comment · Reviewer_g44M · 2022-08-08
> > > > **Thanks**
> > > >
> > > > Thanks for the reply. I would like to keep my score as accept. I strongly recommend clarifying the difference with SpeqNets more in the main text.

---

### Official Review · Reviewer_UZY3 · 2022-07-12

**Rating:** 7
**Confidence:** 3
**Soundness:** 3 good
**Presentation:** 3 good
**Contribution:** 4 excellent

**Summary:**

The paper deals with supervised learning on graph structured data for which the most common framework of GNN is limited in expressivity by 1-WL test in terms of its identifying capacity of graphs.
Higher order $K$-WL hierarchy of algorithms are well-known to improve the expressivity of GNN models with increasing $K$. However, the complexity of the $K$-WL increases exponentially with increasing $K$, making it infeasible to apply in practice. The authors propose $(k,c)(\leq)$-SetWL hierarchy in place of WL-hierarchy. a It is a “finer-grained ruler” than WL-hierarchy and mainly depends on sets (size $\leq k$) instead of tuples of nodes and connected components ($\leq c$). The authors develop the method moving from tuples in WL to multisets, then to sets and then to sets with connected components and design its neural counterpart $(k,c)(\leq)$-SETGNN. Experimental results on both synthetic and real datasets show that the method is promising.


**Questions:**

Please address the weakness mentioned above.


**Limitations:**

The authors have discussed some limitations. There is no potential negative societal impact.


**Strengths And Weaknesses:**

### Strengths:
1. The paper addresses an important problem of going beyond 1-WL, however, in such a manner such that expressivity is improved in a progressive way while runtime is still manageable
1. The idea naturally follows from K-WL models of moving from tuples to multisets, then to sets. The paper rigorously explores this.
1. Experimental results on graph classification and substructure identification as well as regression Zinc12k are promising.
1. The paper is well-written with needed rigor. However, the presentation could be improved for better readability.

### Weaknesses/Comments:
+ The result in Theorem-1 of k-Multiset-WL being equal to K-WL is surprising. I think this should have been discussed with some intuitive figures to show where exactly is the redundancy in K-WL that is being exploited by K-Multiset-WL. I could not follow the proof properly, however it seems to me that with the perm[] function, you would spend same amount of time as K-WL in updating the node colors. Exact number of supernodes in K-Multiset-WL in comparison with K-WL is also missing.
+ Runtime analysis is not complete. Though it is nice to see the exact ratio of speedup of K-SETWL with respect to WL, it would be much clearer if we see the runtime of WL, Multiset-WL, SET-WL,  in BigO notation. Further, what is the cost with respect to $c$ in (k,c)(\leq)-SETWL is also not clear. Asymptotic runtime would also help in visualizing where exactly the speedup comes from.
+ The empirical evaluation should include additional recent expressive GNN models to properly understand the benefits of SET-GNN in comparison with other more expressive models like GraphSNN (Wijesinghe, et al. (2021)), PF-GNN (Dupty et al. (2021)), GNNML3 (Balcilar et al. (2021)) etc. This would also be important since these methods employ different techniques to achieve expressivity and SETGNN can be analysed in perspective.
+ The paper can be significantly improved in readability if some pictures are added to describe the key points where redundancy is removed from tuple to multisets, then multisets to sets.
+ Line 238: derivation not in Appendix
+ Line 183, it should n_u in place of n_n

**Overall**, I find the idea of the paper novel as well as very significant for improving the expressivity of GNNs. Also, the idea is systematically and rigorously developed towards providing a finer scale for measuring expressivity.

However, I could not fully grasp the proofs and hence I’m less confident, especially since the results are somewhat surprising. Furthermore, experimental results need to be properly placed with respect to the recent literature.

### References:

+ [1] Wijesinghe, et al. (2021).  A New Perspective on" How Graph Neural Networks Go Beyond Weisfeiler-Lehman?". In International Conference on Learning Representations.
+ [2] Dupty et al. (2021) "PF-GNN: Differentiable particle filtering based approximation of universal graph representations." International Conference on Learning Representations.
+ [3] Balcilar et al. (2021)  "Breaking the limits of message passing graph neural networks." International Conference on Machine Learning. PMLR.

---

> ### Author Response · Authors · 2022-08-02
> **Response to UZY3**
>
> We thank the reviewer for the thorough and detailed review on our submission. We respond to the reviewers’ concerns and questions one by one.
>
> >**I think this should have been discussed with some intuitive figures to show where exactly is the redundancy in K-WL that is being exploited by K-Multiset-WL. …  It seems to me that with the perm[] function, you would spend the same amount of time as K-WL in updating the node colors. Exact number of supernodes in K-Multiset-WL in comparison with K-WL is also missing.**
>
> We thank the reviewer’s suggestion of adding more visualizations for each step of moving from k-WL to k-SetWL, which we consider to work in the final version of our paper inside the appendix given the limited space of the main paper.
>
> To answer the reviewer’s question about where exactly is the redundancy in K-WL comparing with k-MWL, we give an example of a tuple (v1, v2, v3) in G and the other tuple (u1, u2, u3) in G’. The key observation is that at any step, for k-WL, when the color of (v1, v2, v3) in G is the same as the color of (u1, u2, u3) in G’, so is the color of (v1, v3, v2) and (u1, u3, u2) or any two tuples from applying same permutation to (v1, v2, v3) and (u1, u2, u3). And when two colors are different, any different permutations to (v1, v2, v3) and (u1, u2, u3) will have different colors. This indicates that the ordering does not contain additional information for distinguishing (v1, v2, v3) and (u1, u2, u3). This property is proved in the first part of the proof of theorem 1.
>
> The reviewer argues that with the perm function the k-MWL will cost the same time as k-WL, which is not true as the number of tuples is greatly reduced. For a graph with n number of nodes, k-WL has n^k number of supernodes while k-MWL has C(n+k-1, k) number of supernodes (the number of combinations with replacement). With n approaching infinity, the ratio is approaching k! As lim_{n->inf} n^k / C(n+k-1,k) = k!.
>
> >**Runtime analysis is not complete. It would be much clearer if we see the runtime of WL, Multiset-WL, SET-WL, in BigO notation.**
>
> In the paper we have given the analysis between k-WL and k-SetWL with the specific number of supernodes and number of superedges for variable n and k in Sec.3.5. Unfortunately, the exact formulation is challenging  to simplify even with the help of the bound of summation of binomial coefficients. Hence we cannot derive the complexity for k-SetWL in BigO notation. Besides, the speedup cannot be fully reflected inside BigO notation as it ignores the constant factor (with respect to k), which is the main part of the improvement. For k-MultisetWL we can give the similar analysis as k-SetWL which we will add into the appendix in the final version. Specifically, the k-MultisetWL has C(n+k-1, k) number of supernodes, and k*C(n+k-1, k)/2 number of superedges.
>
> >**The empirical evaluation should include additional recent expressive GNN models. Such as GraphSNN (Wijesinghe, et al. (2021)), PF-GNN (Dupty et al. (2021)), GNNML3 (Balcilar et al. (2021)) etc**
>
> We appreciate the reviewer’s suggestion of adding additional baselines. We have added PF-GNN’s result on ZINC to table 3. For GNNML3 the implementation doesn’t take edge attributes into consideration so we left it out. For GraphSNN, it shares a similar idea as GNN-AK and to our best knowledge GNN-AK is the one that achieves the highest performance among all recent subgraph based methods.
>
> >**Line 238: derivation not in Appendix**
>
> Thank you for pointing out, and we have added that into Appendix in A.2.1.
>
> >**Line 183, it should n_u in place of n_n**
>
> Thank you and we have updated accordingly.

---

### Meta-Review · Area_Chair_uHsG · 2022-08-27

**Recommendation:** Accept
**Confidence:** Less certain

**Metareview:**

The papers recognizes an important open question in GNNs: expressiveness-complexity tradeoff. Current more expressive GNNs (k-WL equivalent GNNs) are not practical for even small values of k, and several works have found great empirical success in graph learning tasks without highly expressive models. This paper puts forth a more “fine-grained ruler,” which can more gradually increase expressiveness to investigate the expressiveness-complexity tradeoff.

The paper addresses an important problem of going beyond 1-WL, however, in such a manner such that expressivity is improved in a progressive way while runtime is still manageable. Experimental results on graph classification and substructure identification as well as regression Zinc12k are promising.

However, the committee has concerns on the presentation of the paper. The authors are suggested to make the paper more accessible by providing more high-level descriptions and interpretations of the results.

**Award:**

No

---

### Decision · Program_Chairs · 2022-09-14

Accept